# Mean-Field Control based Approximation of Multi-Agent Reinforcement Learning in Presence of a Non-decomposable Shared Global State

**Washim Uddin Mondal**                                              *wmondal@purdue.edu*
*School of IE and CE, Purdue University*

**Vaneet Aggarwal**                                                 *vaneet@purdue.edu*
*School of IE and ECE, Purdue University*

**Satish V. Ukkusuri**                                              *sukkusur@purdue.edu*
*Lyles School of Civil Engineering, Purdue University*

**Reviewed on OpenReview:** *https://openreview.net/forum?id=ZME2nZMTvY*

## Abstract

Mean Field Control (MFC) is a powerful approximation tool to solve large-scale Multi-Agent Reinforcement Learning (MARL) problems. However, the success of MFC relies on the presumption that given the local states and actions of all the agents, the next (local) states of the agents evolve conditionally independent of each other. Here we demonstrate that even in a MARL setting where agents share a common global state in addition to their local states evolving conditionally independently (thus introducing a correlation between the state transition processes of individual agents), the MFC can still be applied as a good approximation tool. The global state is assumed to be non-decomposable i.e., it cannot be expressed as a collection of local states of the agents. We compute the approximation error as $\mathcal{O}(e)$ where $e = \frac{1}{\sqrt{N}}\left[\sqrt{|\mathcal{X}|} + \sqrt{|\mathcal{U}|}\right]$. The size of the agent population is denoted by the term $N$, and $|\mathcal{X}|, |\mathcal{U}|$ respectively indicate the sizes of (local) state and action spaces of individual agents. The approximation error is found to be independent of the size of the shared global state space. We further demonstrate that in a special case if the reward and state transition functions are independent of the action distribution of the population, then the error can be improved to $e = \frac{\sqrt{|\mathcal{X}|}}{\sqrt{N}}$. Finally, we devise a Natural Policy Gradient based algorithm that solves the MFC problem with $\mathcal{O}(\epsilon^{-3})$ sample complexity and obtains a policy that is within $\mathcal{O}(\max\{e, \epsilon\})$ error of the optimal MARL policy for any $\epsilon > 0$.

## 1 Introduction

Adaptive decision-making by a large number of cooperative autonomous entities in the presence of a changing environment is a frequently appearing theme in many areas of modern human endeavor such as transportation, telecommunications, and internet networks. For example, consider the ride-hailing service provided by a large fleet of vehicles. Not only those vehicles are needed to be strategically placed to allow the maximum number of passengers to be served but such a formidable task needs to be performed even in the presence of spatio-temporal variation of passenger demand. In this and many other similar scenarios, the central question is how a large number of agents can learn to cooperatively achieve a desired target. The framework of cooperative multi-agent reinforcement learning (MARL) has been developed to answer such questions. However, in comparison to the single-agent learning framework, the task of cooperative MARL is significantly more challenging since as the number of agents increases, the size of the joint state space increases exponentially.

Several heuristic approaches have been designed to circumvent the aforementioned *curse of dimensionality*. Based on how the agents are trained, these approaches can be primarily classified into two categories. In Independent Q Learning (IQL) (Tan, 1993), the agents are trained independently. In contrast, centralized training with decentralized execution (CTDE) based methods (Rashid et al., 2018) train the agents in a centralized fashion. Both approaches avoid the problem of state-space explosion by restricting the policies of each agent to depend only on its local state. Despite having empirical success, none of the above approaches can be shown to have an optimality guarantee. Another approach that has recently emerged as an excellent approximation tool for cooperative MARL with theoretical optimality guarantee is called mean-field control (MFC). It works on the premise that in an infinite collection of homogeneous and exchangeable agents, the behavior of an arbitrarily chosen representative accurately reflects the behavior of the entire population.

The guarantee of MFC as an approximation tool of MARL primarily relies on the law of large numbers which dictates that the empirical average of a large number of independent and identically distributed random variables, with high probability, is very close to their mean value. To utilize this property, it is commonly assumed in the mean-field literature, often implicitly, that the agents are each associated with a local state that evolves conditionally independently of each other (Gu et al., 2021). Unfortunately, in many of the practical scenarios, such an assumption might appear to be too restrictive. As an example, consider the ride-sharing problem discussed before. Assume that the local state perceived by a vehicle is the location of the potential riders in its immediate vicinity. If two vehicles are located far apart, their local states might be assumed to evolve conditionally independently. However, if they are so close such that their pickup areas intersect, then the presumption of independent evolution of states might not hold.

Does MFC-based approximation still hold if the local state evolution processes of different agents are correlated? This is one of the most crucial questions that need to be addressed if mean-field-based techniques are to be adopted in a wide array of real-world MARL problems. In this paper, we establish that if each agent, in addition to their conditionally independently evolving local states, also possesses a common global state, the mean-field approach can still be shown to be a good approximation of MARL. The global state is assumed to be non-decomposable i.e., in general, it cannot be expressed as a collection of some agent-specific local components. Note that, as the global state is common to all agents, the combined state perceived by each agent can now no longer be considered to be evolving independently. As a result, the techniques that are commonly applied to show mean-field approximations can no longer be directly used. We address this challenge by showing that instead of naively applying the previously developed methods, if the problems are transformed to an equivalent but slightly different representation, then we can find new variables that evolve conditionally independently of each other and help us break the correlation barrier.

## 1.1 Our Contribution

We consider a network comprising $N$ number of agents, each associated with a local state space of size $|\mathcal{X}|$, a non-decomposable global state space of size $|\mathcal{G}|$ and an action space of size $|\mathcal{U}|$ (all are assumed to be of finite size). Given the global state, local states, and actions of all agents at time $t$, local states at $t+1$ are presumed to evolve conditionally independently of each other. The combined (local, global) states of each agent, however, do not evolve conditionally independently. We prove that the mean-field control-based approximation results can be applied even in the presence of such a correlation. We quantify the approximation error to be $\mathcal{O}(e)$ where $e \triangleq \frac{1}{\sqrt{N}}\left[\sqrt{|\mathcal{X}|} + \sqrt{|\mathcal{U}|}\right]$. Note that the expression of $e$ is independent of $|\mathcal{G}|$, the size of the global state-space. In a special case where the reward function and both the local and global state transition functions are assumed to not depend on the action distribution of the agent population, the approximation error is shown to improve to $e = \frac{\sqrt{|\mathcal{X}|}}{\sqrt{N}}$ i.e., it becomes independent of the size of the action space.

In traditional MFC (with no non-decomposable global state), one of the crucial steps in proving the MARL-MFC approximation error is upper bounding the term $\mathbb{E}|\boldsymbol{\mu}_t^N - \boldsymbol{\mu}_t|_1$ as $\mathcal{O}(1/\sqrt{N})$ where $\boldsymbol{\mu}_t^N$, and $\boldsymbol{\mu}_t$ are the state-distributions of the $N$-agent and the infinite agent systems respectively at time $t$. The key intuition in proving this bound comes from the fact that $\boldsymbol{\mu}_t^N(x)$ can be written as the average of the random variables $\{\delta(x_t^i = x)\}_{i=1}^N$ where $x_t^i$ is the state of the $i$th agent at time $t$. Moreover, the above-mentioned random

variables are independent conditioned on $\boldsymbol{\mu}_{t-1}^N$. This allows one to use the law of large numbers and obtain the desired bound.

If we follow the same footsteps in our setting (with a non-decomposable global state), we will end up with the term $\sum_{x,g} \mathbb{E}|\boldsymbol{\mu}_t^N(x)\delta(g_t^N = g) - \boldsymbol{\mu}_t(x)\delta(g_t = g)|$ where $g_t^N$ and $g_t$ are the non-decomposable states in the $N$-agent and the infinite agent systems respectively. Note that, we can write $\boldsymbol{\mu}_t^N(x)\delta(g_t^N = g)$ as the average of the random variables $\{\delta(x_t^i = x, g_t^N = g)\}_{i=1}^N$. Due to the common factor $\delta(g_t^N = g)$, the above-mentioned random variables are now correlated, and hence the law of large numbers can no longer be applied. This is essentially the primary challenge addressed by our paper. We invent new techniques to prove approximation guarantees even in the presence of a non-decomposable state. The details are provided in section 6.

Finally, we develop a natural policy gradient (NPG) based algorithm and using the result of (Liu et al., 2020), and our own approximation guarantee, establish that the proposed algorithm generates a policy that is within $\mathcal{O}(\max\{e, \epsilon\})$ error of the optimal MARL policy for any $\epsilon > 0$. Moreover, the sample complexity bound for obtaining such a solution is shown to be $\mathcal{O}(\epsilon^{-3})$.

## 1.2 Related Works

**Single Agent Learning:** Pioneering works in the single agent learning setup includes various finite state/tabular algorithms such as Q-learning (Watkins and Dayan, 1992) and SARSA (Rummery and Niranjan, 1994). Despite having theoretical guarantees, these algorithms did not get adopted in large-scale applications due to the huge memory requirement. Recently, neural network (NN) based Deep Q Learning (DQL) (Mnih et al., 2015) and policy gradient-based algorithms (Mnih et al., 2016) have garnered popularity due to the large expressive powers of NNs. Nevertheless, due to the exponential increase in the size of the joint state space with the number of agents, these algorithms are far from being the panacea for large-scale MARL problems.

**CTDE-based Approaches for MARL:** As stated before, the idea behind centralized training and decentralized execution (CTDE) based approaches is to restrict the policies to solely take the local state of the associated agent as an input. Depending on how such *local* policies are trained, various algorithms have been constructed. For example, VDN (Sunehag et al., 2017), trains the local policies by minimizing the Bellman error corresponding to the sum of individual Q-functions of each agent. QMIX (Rashid et al., 2018), on the other hand, computes the Bellman error corresponding to the state-dependent weighted sum of the Q-functions of each agent. Various other CTDE-based algorithms are WQMIX (Rashid et al., 2020), QTRAN (Son et al., 2019) etc. Parallel to CTDE, IQL-based algorithms have also garnered popularity in large-scale MARL (Wei et al., 2019). Alongside these heuristics, there have been some recent efforts to theoretically characterize the efficacy of the local policies Qu et al. (2020); Lin et al. (2021); Mondal et al. (2022c). However, none of them include a common global state.

**Mean-Field Control (MFC):** MFC is a relatively recent development that solves MARL with theoretical optimality guarantees. (Gu et al., 2021) exhibited that if all the agents are homogeneous and exchangeable, MFC can be used as a good approximation of an $N$-agent problem. Later, similar approximation results were proved for $K$-class of heterogeneous agents (Mondal et al., 2022a) and non-exchangeable agents (Mondal et al., 2022b). Moreover, various model-based (Pasztor et al., 2021) and model-free (Angiuli et al., 2022) algorithms have been developed to solve the MFC problem. We would like to point out that our framework is closely aligned with the framework of MFC with common noise (Motte and Pham, 2022; Carmona et al., 2019). However, (Motte and Pham, 2022) only considers open-loop policies which are essentially sequences of actions, rather than state-to-action maps (also known as closed-loop policies). On the other hand, although (Carmona et al., 2019) do consider closed-loop policies, they do not show the convergence between MARL and MFC as a function of the number of agents. Empirically, MFC has found its application in a diverse range of scenarios, including epidemic management (Watkins et al., 2016), congestion control (Wang et al., 2020), and ride-sharing (Al-Abbasi et al., 2019).

**Beyond Mean-Field Control:** The presumption of homogeneity of the agents turns out to be too restrictive in many practical scenarios. Graphon mean-field control (Caines and Huang, 2019) is an emerging new

area that attempts to do away with the presumption of homogeneity. However, as explained in (Mondal et al., 2022b), such approaches also come with their own set of restrictions.

**Mean-Field Games:** Similar to MFC, mean-field games (MFG) attempt to characterize the behavior of an infinite number of agents in a non-cooperative setup. In contrast to MFC, the goal of MFG is to identify the Nash equilibrium of the system (Elie et al., 2020; Yang et al., 2017).

## 2 MARL with Shared Global State

We consider a collection of $N$ interacting agents each with a local state space $\mathcal{X}$ and an action space $\mathcal{U}$. At instant $t$, the local state and action of the $i$th agent are respectively indicated by $x_t^i, u_t^i$. In addition to the local state, at time $t$, each agent also observes a global state $g_t^N$ whose realizations are from the global state space, $\mathcal{G}$. The collection of local states and actions of all agents are expressed as $\boldsymbol{x}_t^N, \boldsymbol{u}_t^N$ respectively. Given the tuple $(\boldsymbol{x}_t^N, g_t^N, \boldsymbol{u}_t^N)$, the local state of $i$th agent at time $t+1$ is given by the following transition law, $x_{t+1}^i \sim P_i(\boldsymbol{x}_t^N, g_t^N, \boldsymbol{u}_t^N)$. Similarly, the transition law for the global state is given as $g_{t+1}^N \sim P_G(\boldsymbol{x}_t^N, g_t^N, \boldsymbol{u}_t^N)$. It is assumed that the random variables $\{\{x_{t+1}^i\}_{i=1}^N, g_{t+1}\}$ are independent, conditioned on $\{\boldsymbol{x}_t^N, g_t^N, \boldsymbol{u}_t^N\}$. At time $t$, the (expected) reward received by the $i$th agent is denoted as $r_i(\boldsymbol{x}_t^N, g_t^N, \boldsymbol{u}_t^N)$. Let $\boldsymbol{\mu}_t^N, \boldsymbol{\nu}_t^N$ indicate the empirical state and action distributions of $N$ agents at instant $t$ which are defined respectively as follows.

$$\boldsymbol{\mu}_t^N(x) \triangleq \frac{1}{N} \sum_{i=1}^N \delta\left(x_t^i = x\right), \forall x \in \mathcal{X} \tag{1}$$

$$\boldsymbol{\nu}_t^N(u) \triangleq \frac{1}{N} \sum_{i=1}^N \delta\left(u_t^i = u\right), \forall u \in \mathcal{U} \tag{2}$$

where $\delta(\cdot)$ is an indicator function. We assume the agents to be homogeneous and exchangeable. Hence, the reward and state transition functions can be written as follows.

$$r_i(\boldsymbol{x}_t^N, g_t^N, \boldsymbol{u}_t^N) = r(x_t^i, u_t^i, \boldsymbol{\mu}_t^N, g_t^N, \boldsymbol{\nu}_t^N) \tag{3}$$

$$P_i(\boldsymbol{x}_t^N, g_t^N, \boldsymbol{u}_t^N) = P(x_t^i, u_t^i, \boldsymbol{\mu}_t^N, g_t^N, \boldsymbol{\nu}_t^N) \tag{4}$$

$$P_G(\boldsymbol{x}_t^N, g_t^N, \boldsymbol{u}_t^N) = P_G(\boldsymbol{\mu}_t^N, g_t^N, \boldsymbol{\nu}_t^N) \tag{5}$$

where $r, P, P_G$ are given as follows: $r : \mathcal{X} \times \mathcal{U} \times \Delta(\mathcal{X}) \times \mathcal{G} \times \Delta(\mathcal{U}) \to \mathbb{R}$, $P : \mathcal{X} \times \mathcal{U} \times \Delta(\mathcal{X}) \times \mathcal{G} \times \Delta(\mathcal{U}) \to \Delta(\mathcal{X})$ and $P_G : \Delta(\mathcal{X}) \times \mathcal{G} \times \Delta(\mathcal{U}) \to \Delta(\mathcal{G})$. The symbol $\Delta(\mathcal{S})$ defines the probability simplex defined over the set $\mathcal{S}$. Note that (5) is written with a slight abuse of notations.

A policy $\pi_t$ is a function of the form, $\pi_t : \mathcal{X} \times \Delta(\mathcal{X}) \times \mathcal{G} \to \Delta(\mathcal{U})$. In simple terms, a policy is a recipe for the agents to (probabilistically) choose actions based on their current local states, the empirical local state distribution of all the agents, and the global state[1]. Let $\boldsymbol{\pi} \triangleq \{\pi_t\}_{t \in \{0,1,\cdots\}}$ be a sequence of policies. The value generated by the sequence $\boldsymbol{\pi}$ corresponding to the initial state $(\boldsymbol{x}_0^N, g_0^N)$ is defined as follows.

$$
\begin{aligned}
V_N(\boldsymbol{x}_0^N, g_0^N, \boldsymbol{\pi}) &\triangleq \frac{1}{N} \sum_{i=1}^N \mathbb{E}\left[\sum_{t=0}^\infty \gamma^t r_i(\boldsymbol{x}_t^N, g_t^N, \boldsymbol{u}_t^N)\right] \\
&= \frac{1}{N} \sum_{i=1}^N \mathbb{E}\left[\sum_{t=0}^\infty \gamma^t r(x_t^i, u_t^i, \boldsymbol{\mu}_t^N, g_t^N, \boldsymbol{\nu}_t^N)\right]
\end{aligned}
\tag{6}
$$

where the expectation is taken over all trajectories generated by the policy sequence $\boldsymbol{\pi}$ starting from the initial states $(\boldsymbol{x}_0^N, g_0^N)$ and $\gamma \in (0, 1)$ is the discount factor. The target of MARL is to compute a policy sequence that maximizes $V_N(\boldsymbol{x}_0^N, g_0^N, \cdot)$ over the set of admissible policy sequences $\Pi^\infty \triangleq \Pi \times \Pi \times \cdots$ where $\Pi$ is the set of admissible policies. In the next section, we shall discuss the mean-field control (MFC) framework that can be used to approximately solve the MARL problem.

---

[1] Here we implicitly assume that all agents execute the same policy. This is primarily because the agents are presumed to have the same reward function and state-transition function (Gu et al., 2021; Pasztor et al., 2021).

## 3 Mean-Field Control (MFC) Framework

In this setting, we consider the size of the agent population to be infinite. Due to homogeneity, we can arbitrarily select a representative agent whose local state and action at time $t$ are defined as $x_t$, and $u_t$ respectively. In addition, let $g_t$ indicate the global state at time $t$. Let $\boldsymbol{\mu}_t, \boldsymbol{\nu}_t$ be the distributions of local states and actions at time $t$ over the infinite agent population. For a given sequence $\boldsymbol{\pi} = \{\pi_t\}_{t \in \{0,1,\cdots\}}$, define the following.

$$\boldsymbol{\nu}_t = \nu^{\mathrm{MF}}(\boldsymbol{\mu}_t, g_t, \pi_t) \triangleq \sum_{x \in \mathcal{X}} \pi_t(x, \boldsymbol{\mu}_t, g_t) \boldsymbol{\mu}_t(x) \tag{7}$$

The above relation demonstrates how the action distribution $\boldsymbol{\nu}_t$ can be obtained from the local state distribution $\boldsymbol{\mu}_t$ and the global state $g_t$. Now we quantitatively describe how $\boldsymbol{\mu}_{t+1}$, the local state distribution at $t + 1$, can be obtained from $\boldsymbol{\mu}_t$ an $g_t$.

$$\boldsymbol{\mu}_{t+1} = \sum_{x \in \mathcal{X}} \sum_{u \in \mathcal{U}} P(x, u, \boldsymbol{\mu}_t, g_t, \nu^{\mathrm{MF}}(\boldsymbol{\mu}_t, g_t, \pi_t)) \times \pi_t(x, \boldsymbol{\mu}_t, g_t)(u) \boldsymbol{\mu}_t(x) \triangleq P^{\mathrm{MF}}(\boldsymbol{\mu}_t, g_t, \pi_t) \tag{8}$$

Define $\boldsymbol{\lambda}_t \triangleq P_G(\boldsymbol{\mu}_{t-1}, g_{t-1}, \boldsymbol{\nu}_{t-1})$, $t \geq 1$ and $\boldsymbol{\lambda}_0 \triangleq \mathbf{1}(g_0)$ where $\mathbf{1}(g_0)$ indicates a one-hot vector with a nonzero element at the position corresponding to $g_0$. Intuitively, $\boldsymbol{\lambda}_t$ denotes the conditional distribution of $g_t$ given $(\boldsymbol{\mu}_{t-1}, g_{t-1}, \boldsymbol{\nu}_{t-1})$. Note that the following relation holds $\forall t \geq 0$.

$$\boldsymbol{\lambda}_{t+1} = P_G^{\mathrm{MF}}(\boldsymbol{\mu}_t, g_t, \pi_t) \triangleq P_G(\boldsymbol{\mu}_t, g_t, \nu^{\mathrm{MF}}(\boldsymbol{\mu}_t, g_t, \pi_t)) \tag{9}$$

Finally, the average reward at time $t$ is expressed as follows.

$$r^{\mathrm{MF}}(\boldsymbol{\mu}_t, g_t, \pi_t) = \sum_{x \in \mathcal{X}} \sum_{u \in \mathcal{U}} \pi_t(x, \boldsymbol{\mu}_t, g_t)(u) \times \boldsymbol{\mu}_t(x) \times r(x, u, \boldsymbol{\mu}_t, g_t, \nu^{\mathrm{MF}}(\boldsymbol{\mu}_t, g_t, \pi_t)) \tag{10}$$

The mean-field value generated by $\boldsymbol{\pi} = \{\pi_t\}_{t \in \{0,1,\cdots\}}$ for a local state distribution $\boldsymbol{\mu}_0$ and global state $g_0$ is expressed as follows.

$$V_\infty(\boldsymbol{\mu}_0, g_0, \boldsymbol{\pi}) = \sum_{t=0}^{\infty} \gamma^t \mathbb{E}\left[r^{\mathrm{MF}}(\boldsymbol{\mu}_t, g_t, \pi_t)\right] \tag{11}$$

where the expectation is obtained over $\{g_t\}_{t \in \{1,2\cdots\}}$ where $g_{t+1} \sim P_G^{\mathrm{MF}}(\boldsymbol{\mu}_t, g_t, \pi_t)$, $\boldsymbol{\mu}_{t+1} = P^{\mathrm{MF}}(\boldsymbol{\mu}_t, g_t, \pi_t)$, $t \geq 0$. The goal of MFC is to maximize the function $V_\infty(\boldsymbol{\mu}_0, g_0, \cdot)$ over the set of admissible policy sequences, $\Pi^\infty$. In the following section, we show that the optimal value of MARL is approximately equal to its associated optimal MFC value.

## 4 Approximation Result

We shall first dictate some assumptions that are needed to establish the main result.

**Assumption 1.** *The functions $r$, $P$ and $P_G$ are assumed to follow the following relations $\forall \boldsymbol{\mu}_1, \boldsymbol{\mu}_2 \in \Delta(\mathcal{X})$, $\forall \boldsymbol{\nu}_1, \boldsymbol{\nu}_2 \in \Delta(\mathcal{U})$, $\forall x \in \mathcal{X}$, $\forall u \in \mathcal{U}$, and $\forall g \in \mathcal{G}$*

$$(a) \ |r(x, u, \boldsymbol{\mu}_1, g, \boldsymbol{\nu}_1)| \leq M_R$$
$$(b) \ |r(x, u, \boldsymbol{\mu}_1, g, \boldsymbol{\nu}_1) - r(x, u, \boldsymbol{\mu}_2, g, \boldsymbol{\nu}_2)| \leq L_R\{|\boldsymbol{\mu}_1 - \boldsymbol{\mu}_2|_1 + |\boldsymbol{\nu}_1 - \boldsymbol{\nu}_2|_1\}$$
$$(c) \ |P(x, u, \boldsymbol{\mu}_1, g, \boldsymbol{\nu}_1) - P(x, u, \boldsymbol{\mu}_2, g, \boldsymbol{\nu}_2)|_1 \leq L_P\{|\boldsymbol{\mu}_1 - \boldsymbol{\mu}_2|_1 + |\boldsymbol{\nu}_1 - \boldsymbol{\nu}_2|_1\}$$
$$(d) \ |P_G(\boldsymbol{\mu}_1, g, \boldsymbol{\nu}_1) - P_G(\boldsymbol{\mu}_2, g, \boldsymbol{\nu}_2)|_1 \leq L_G\{|\boldsymbol{\mu}_1 - \boldsymbol{\mu}_2|_1 + |\boldsymbol{\nu}_1 - \boldsymbol{\nu}_2|_1\}$$

*The constants $L_R$, $L_P$ are arbitrary positive numbers. The function $|\cdot|_1$ denotes the $L_1$-norm.*

Assumption 1(a) states that the reward function is bounded within the finite interval $[-M_R, M_R]$. On the other hand, assumption 1(b), 1(c), and 1(d) respectively dictates that the reward function, $r$, the local state transition function, $P$ and the global state transition function, $P_G$ are all Lipschitz continuous with respect to their local state distribution and action distribution arguments. Such assumptions are common in the literature (Pasztor et al., 2021; Hinderer, 2005; Gu et al., 2021). We would like to point out that although the state and action distributions are treated as continuous variables, in an $N$-agent problem, they can only take a finite number of values in their respective probability simplexes. In the MFC problem, however, these variables can be arbitrary. Therefore, while comparing the $N$-agent and the MFC problem, we are implicitly extending the domain of definition for the reward and the state transition functions.

**Assumption 2.** *The set of admissible policies, $\Pi$ is such that any $\pi \in \Pi$ satisfies the following inequality $\forall x \in \mathcal{X}$, $\forall \boldsymbol{\mu}_1, \boldsymbol{\mu}_2 \in \Delta(\mathcal{X})$, and $\forall g \in \mathcal{G}$.*

$$|\pi(x, \boldsymbol{\mu}_1, g) - \pi(x, \boldsymbol{\mu}_2, g)| \leq L_Q |\boldsymbol{\mu}_1 - \boldsymbol{\mu}_2|_1$$

Assumption 2 states that the set of admissible policies is selected such that each of its elements is Lipschitz continuous with respect to their local state distribution argument. Such assumption typically holds for neural network (NN) based policies with bounded weights (Mondal et al., 2022a; Pasztor et al., 2021; Cui and Koeppl, 2021).

We are now ready to state the main result. The proof of the theorem stated below is relegated to Appendix A.

**Theorem 1.** *Let $\boldsymbol{x}_0 \triangleq \{x_0^i\}_{i \in \{1, \cdots, N\}}$ and $g_0$ be the initial states and $\boldsymbol{\mu}_0$ denote the empirical distribution of $\boldsymbol{x}_0$. If Assumption 1 holds and the set of admissible policies, $\Pi$ satisfies Assumption 2, then the following relation is true whenever $\gamma S_P < 1$.*

$$|\sup_{\boldsymbol{\pi}} \ V_N(\boldsymbol{x}_0, g_0, \boldsymbol{\pi}) - \sup_{\boldsymbol{\pi}} \ V_\infty(\boldsymbol{\mu}_0, g_0, \boldsymbol{\pi})|$$

$$\leq \sup_{\boldsymbol{\pi}} \ |V_N(\boldsymbol{x}_0, g_0, \boldsymbol{\pi}) - V_\infty(\boldsymbol{\mu}_0, g_0, \boldsymbol{\pi})| \leq \left(\frac{M_R + L_R\sqrt{|\mathcal{U}|}}{1-\gamma}\right)\frac{1}{\sqrt{N}} + \sqrt{\frac{|\mathcal{U}|}{N}}\frac{M_R L_G \gamma}{(1-\gamma)^2}$$

$$+ \left(\frac{C_P}{S_P - 1}\right)\left[\left(\frac{M_R S_G}{S_P - 1} + S_R\right)\left\{\frac{1}{1-\gamma S_P} - \frac{1}{1-\gamma}\right\} - \frac{\gamma M_R S_G}{(1-\gamma)^2}\right] \times \frac{1}{\sqrt{N}}\left[\sqrt{|\mathcal{X}|} + \sqrt{|\mathcal{U}|}\right]$$

*where $S_P \triangleq 1 + 2L_P + L_Q(1 + L_P)$, $S_R \triangleq M_R + 2L_R + L_Q(M_R + L_R)$, $S_G \triangleq L_G(2 + L_Q)$ and $C_P \triangleq 2 + L_P$. Suprema are performed over the class of all admissible policy sequences, $\Pi^\infty$.*

Theorem 1 states that if the discount factor $\gamma$ is sufficiently small, then under assumptions 1 and 2, the optimal $N$-agent value function is at most $\mathcal{O}\left(\frac{1}{\sqrt{N}}\left[\sqrt{|\mathcal{X}|} + \sqrt{|\mathcal{U}|}\right]\right)$ error away from the optimal mean-field value function. In other words, if the number of agents, $N$ is large and the sizes of local states and actions of individual agents is sufficiently small, then an optimal solution of MFC well approximates the optimal solution of a MARL problem as described in section 2. Interestingly, notice that the approximation error does not depend on the size of global state space $|\mathcal{G}|$. Thus, the approximation error can be kept small even when $|\mathcal{G}|$ is dramatically large or even potentially infinite.

Let, $\boldsymbol{\pi}_\epsilon^{\mathrm{MF}} \in \Pi^\infty$ be an admissible policy sequence that solves the MFC problem with $\epsilon$ accuracy. Note that,

$$|\sup_{\boldsymbol{\pi}} V_N(\boldsymbol{x}_0, g_0, \boldsymbol{\pi}) - V_N(\boldsymbol{x}_0, g_0, \boldsymbol{\pi}_\epsilon^{\mathrm{MF}})| \leq \underbrace{|\sup_{\boldsymbol{\pi}} V_N(\boldsymbol{x}_0, g_0, \boldsymbol{\pi}) - \sup_{\boldsymbol{\pi}} V_\infty(\boldsymbol{\mu}_0, g_0, \boldsymbol{\pi})|}_{\triangleq J_1}$$

$$+ \underbrace{|\sup_{\boldsymbol{\pi}} V_\infty(\boldsymbol{\mu}_0, g_0, \boldsymbol{\pi}) - V_\infty(\boldsymbol{\mu}_0, g_0, \boldsymbol{\pi}_\epsilon^{\mathrm{MF}})|}_{\triangleq J_2} + \underbrace{|V_\infty(\boldsymbol{\mu}_0, g_0, \boldsymbol{\pi}_\epsilon^{\mathrm{MF}}) - V_N(\boldsymbol{x}_0, g_0, \boldsymbol{\pi}_\epsilon^{\mathrm{MF}})|}_{\triangleq J_3}$$

The terms $J_1$, $J_3$ can be bounded by Theorem 1 while $J_2$ can be bounded by $\epsilon$. This result suggests that if we can come up with a way to approximately solve the MFC problem, then that solution must be a good proxy

for the optimal MARL solution. Before discussing the algorithmic aspects of solving the MFC problem, we would like to first discuss a special case where the approximation error can be lower in comparison to that stated in Theorem 1.

## 5 Improvement in a Special Case

In this section, we shall demonstrate that if certain structural restrictions are imposed on the reward and state transition functions, then the approximation error stated in Theorem 1 can be improved further. In particular, the assumption imposed on the stated functions can be mathematically described as follows.

**Assumption 3.** *The following relations hold* $\forall \boldsymbol{\mu} \in \Delta(\mathcal{X})$, $\forall \boldsymbol{\nu} \in \Delta(\mathcal{U})$, $\forall x \in \mathcal{X}$, $\forall g \in \mathcal{G}$, $\forall u \in \mathcal{U}$,

$$(a)\ r(x, u, \boldsymbol{\mu}, g, \boldsymbol{\nu}) = r(x, u, \boldsymbol{\mu}, g),$$
$$(b)\ P(x, u, \boldsymbol{\mu}, g, \boldsymbol{\nu}) = P(x, u, \boldsymbol{\mu}, g),$$
$$(c)\ P_G(\boldsymbol{\mu}, g, \boldsymbol{\nu}) = P_G(\boldsymbol{\mu}, g)$$

*Note that the above relations are stated with a slight abuse of notations.*

Assumption 3 dictates that the reward function, $r$, the local state transition function, $P$, and the global state transition function, $P_G$ are independent of the action distribution of the population. We would like to point out that the functions $r$ and $P$ may still depend on the action of the associated agent even if they are independent of the actions of others. In Theorem 2, we discuss the implication of Assumption 3. The proof of Theorem 2 is relegated to Appendix N.

**Theorem 2.** *Let* $\boldsymbol{x}_0 \triangleq \{x_0^i\}_{i \in \{1, \cdots, N\}}$ *and* $g_0$ *be the initial states and* $\boldsymbol{\mu}_0$ *indicate the empirical distribution of* $\boldsymbol{x}_0$. *If assumptions 1 and 3 hold and the set of admissible policies,* $\Pi$ *obeys assumption 2, then the following relation is true whenever* $\gamma Q_P < 1$,

$$|\sup_{\boldsymbol{\pi}} V_N(\boldsymbol{x}_0, g_0, \boldsymbol{\pi}) - \sup_{\boldsymbol{\pi}} V_\infty(\boldsymbol{\mu}_0, g_0, \boldsymbol{\pi})| \leq \left(\frac{M_R}{1-\gamma}\right)\frac{1}{\sqrt{N}} + \frac{\sqrt{|\mathcal{X}|}}{\sqrt{N}}$$
$$\times \left(\frac{2}{Q_P - 1}\right)\left[\left(\frac{M_R L_G}{Q_P - 1} + Q_R\right)\left\{\frac{1}{1-\gamma Q_P} - \frac{1}{1-\gamma}\right\} - \frac{\gamma M_R L_G}{(1-\gamma)^2}\right]$$

*where* $Q_P \triangleq 1 + L_P + L_Q$ *and* $Q_R \triangleq M_R(1 + L_Q) + L_R$. *Suprema are computed over the set of all admissible policy sequences,* $\Pi^\infty$.

Theorem 2 dictates that if the reward, $r$, and state transition functions, $P$ and $P_G$ are independent of the action distribution of the entire agent population, then the approximation error can be improved to $\mathcal{O}\left(\frac{\sqrt{|\mathcal{X}|}}{\sqrt{N}}\right)$. Therefore, in addition to large global state space, such system can also afford to have large action space without compromising the MFC based approximation accuracy.

## 6 Proof Outline

Here we present a brief outline of the proof of Theorem 1. The proof of Theorem 2 is similar. In order to describe the proof steps, the infinite agent value function given in (11) needs to be written in an equivalent but slightly different representation.

### 6.1 An Equivalent Representation

Let us focus on the expected infinite-agent reward at time $t$, denoted by the expression $\mathbb{E}[r^{\text{MF}}(\boldsymbol{\mu}_t, g_t, \pi_t)]$. Note that, starting from $\boldsymbol{\mu}_0, g_0$, the sequence of local state distribution and global states $\{(\boldsymbol{\mu}_l, g_l)\}$, $l \in \{1, \cdots, t\}$ can be recursively obtained as follows.

$$\boldsymbol{\mu}_{l+1} = P^{\text{MF}}(\boldsymbol{\mu}_l, g_l, \pi_l) \tag{12}$$
$$g_{l+1} \sim P_G^{\text{MF}}(\boldsymbol{\mu}_l, g_l, \pi_l),\ l \in \{0, \cdots, t-1\} \tag{13}$$

where $P^{\mathrm{MF}}, P_G^{\mathrm{MF}}$ are defined in (8), (9) respectively. Note that (12) is a deterministic relation. Therefore, if we fix a particular realization of the global states $\{g_l\}$, $l \in \{1, \cdots, t-1\}$, then, by recursively applying (12), $\boldsymbol{\mu}_{l+1}$ can be written as follows, $\forall l \in \{0, \cdots, t-1\}$.

$$
\begin{aligned}
\boldsymbol{\mu}_{l+1} &= \tilde{P}^{\mathrm{MF}}(\boldsymbol{\mu}_0, g_{0:l}, \pi_{0:l}) \\
&\triangleq P^{\mathrm{MF}}(\cdot, g_l, \pi_l) \circ P^{\mathrm{MF}}(\cdot, g_{l-1}, \pi_{l-1}) \circ \cdots \circ P^{\mathrm{MF}}(\cdot, g_1, \pi_1) \circ P^{\mathrm{MF}}(\cdot, g_0, \pi_0)(\boldsymbol{\mu}_0)
\end{aligned}
\tag{14}
$$

where $\circ$ indicates function composition, $g_{0:l} \triangleq \{g_0, \cdots, g_l\}$ and $\pi_{0:l} \triangleq \{\pi_0, \cdots, \pi_l\}$. On the other hand, recursively using (13), the probability of occurrence of a particular realisation of the sequence $g_{1:t} \triangleq \{g_1, \cdots, g_t\}$ can be written as follows.

$$
\mathbb{P}(g_{1:t}|\boldsymbol{\mu}_0, g_0, \pi_{0:t-1}) \triangleq P_G^{\mathrm{MF}}(\boldsymbol{\mu}_0, g_0, \pi_0)(g_1) \times \cdots \times P_G^{\mathrm{MF}}(\boldsymbol{\mu}_{t-1}, g_{t-1}, \pi_{t-1})(g_t)
\tag{15}
$$

For notational convenience, we denote this conditional joint probability as $\tilde{P}_G^{\mathrm{MF}}(\boldsymbol{\mu}_0, g_{0:t-1}, \pi_{0:t-1})(g_t)$. Invoking (14), this can be alternatively written as follows.

$$
\begin{aligned}
\tilde{P}_G^{\mathrm{MF}}(\boldsymbol{\mu}_0, g_{0:t-1}, \pi_{0:t-1})(g_t) &= P_G^{\mathrm{MF}}(\boldsymbol{\mu}_0, g_0, \pi_0)(g_1) \times P_G^{\mathrm{MF}}(\tilde{P}^{\mathrm{MF}}(\boldsymbol{\mu}_0, g_{0:0}, \pi_{0:0}), g_1, \pi_1)(g_2) \\
&\quad \times \cdots \times P_G^{\mathrm{MF}}(\tilde{P}^{\mathrm{MF}}(\boldsymbol{\mu}_0, g_{0:t-2}, \pi_{0:t-2}), g_{t-1}, \pi_{t-1})(g_t)
\end{aligned}
\tag{16}
$$

Using these notations, we can now define $\tilde{r}^{\mathrm{MF}}(\boldsymbol{\mu}_0, g_0, \pi_{0:t}) \triangleq \mathbb{E}[r^{\mathrm{MF}}(\boldsymbol{\mu}_t, g_t, \pi_t)]$ as follows.

$$
\tilde{r}^{\mathrm{MF}}(\boldsymbol{\mu}_0, g_0, \pi_{0:t}) \triangleq \sum_{1:t} \tilde{P}_G^{\mathrm{MF}}(\boldsymbol{\mu}_0, g_{0:t-1}, \pi_{0:t-1})(g_t) \times r^{\mathrm{MF}}(\tilde{P}^{\mathrm{MF}}(\boldsymbol{\mu}_0, g_{0:t-1}, \pi_{0:t-1}), g_t, \pi_t)
\tag{17}
$$

where $\sum_{1:t}$ is the summation operation over $g_{1:t} \in \mathcal{G}^t$ for $t \geq 1$. For $t = 0$, we define $\tilde{r}^{\mathrm{MF}}(\boldsymbol{\mu}_0, g_0, \pi_{0:0}) \triangleq r^{\mathrm{MF}}(\boldsymbol{\mu}_0, g_0, \pi_0)$.

## 6.2 Proof Outline

With the new representation defined above, we are now ready to sketch a brief outline of the proof.

**Step 0:** Recall from the definitions (6) and (11) that both $N$-agent and infinite agent value functions are $\gamma$-discounted sum of expected rewards. To obtain their difference corresponding to a given policy sequence, $\boldsymbol{\pi} = \{\pi_t\}_{t \in \{0,1,\cdots\}}$, we therefore must calculate the difference between the expected $N$-agent reward at time $t$ and the expected infinite agent reward at time $t$. Mathematically, this difference can be expressed as follows.

$$
\Delta R_t \triangleq \left| \frac{1}{N} \sum_{i=1}^{N} \mathbb{E}\left[r(x_t^i, u_t^i, \boldsymbol{\mu}_t^N, g_t^N, \boldsymbol{\nu}_t^N)\right] - \mathbb{E}\left[r^{\mathrm{MF}}(\boldsymbol{\mu}_t, g_t, \pi_t)\right] \right|
$$

where all the notations are the same as used in sections 2 and 3.

**Step 1:** We upper bound $\Delta R_t$ as $\Delta R_t \leq \Delta R_t^1 + \Delta R_t^2$. The first term is given as follows.

$$
\Delta R_t^1 \triangleq \mathbb{E}\left| \frac{1}{N} \sum_{i=1}^{N} r(x_t^i, u_t^i, \boldsymbol{\mu}_t^N, g_t^N, \boldsymbol{\nu}_t^N) - r^{\mathrm{MF}}(\boldsymbol{\mu}_t^N, g_t^N, \pi_t) \right|
$$

We show that $\Delta R_t^1 = \mathcal{O}\left(\sqrt{|\mathcal{U}|/N}\right)$ in Lemma 13 (Appendix A.3). The second term $\Delta R_t^2$ is the following.

$$
\Delta R_t^2 \triangleq \left| \mathbb{E}[r^{\mathrm{MF}}(\boldsymbol{\mu}_t^N, g_t^N, \pi_t)] - \mathbb{E}[r^{\mathrm{MF}}(\boldsymbol{\mu}_t, g_t, \pi_t)] \right|
\tag{18}
$$

**Step 2:** Using the definition of $\tilde{r}^{\mathrm{MF}}$ in section 6.1, we can write, $\mathbb{E}[r^{\mathrm{MF}}(\boldsymbol{\mu}_t^N, g_t^N, \pi_t)] = \mathbb{E}[\tilde{r}^{\mathrm{MF}}(\boldsymbol{\mu}_t^N, g_t^N, \pi_{t:t})]$ and $\mathbb{E}[r^{\mathrm{MF}}(\boldsymbol{\mu}_t, g_t, \pi_t)] = \tilde{r}^{\mathrm{MF}}(\boldsymbol{\mu}_0, g_0, \pi_{0:t}) = \mathbb{E}[\tilde{r}^{\mathrm{MF}}(\boldsymbol{\mu}_0, g_0, \pi_{0:t})]$. We now further bound $\Delta R_t^2$ as follows.

$$
\begin{aligned}
\Delta R_t^2 &= \left| \mathbb{E}[\tilde{r}^{\mathrm{MF}}(\boldsymbol{\mu}_t^N, g_t^N, \pi_{t:t})] - \mathbb{E}[\tilde{r}^{\mathrm{MF}}(\boldsymbol{\mu}_0, g_0, \pi_{0:t})] \right| \\
&\overset{(a)}{=} \left| \sum_{k=0}^{t-1} \mathbb{E}[\tilde{r}^{\mathrm{MF}}(\boldsymbol{\mu}_{k+1}^N, g_{k+1}^N, \pi_{k+1:t})] - \mathbb{E}[\tilde{r}^{\mathrm{MF}}(\boldsymbol{\mu}_k^N, g_k^N, \pi_{k:t})] \right| \\
&\leq \sum_{k=0}^{t-1} \underbrace{\left| \mathbb{E}[\tilde{r}^{\mathrm{MF}}(\boldsymbol{\mu}_{k+1}^N, g_{k+1}^N, \pi_{k+1:t})] - \mathbb{E}[\tilde{r}^{\mathrm{MF}}(\boldsymbol{\mu}_k^N, g_k^N, \pi_{k:t})] \right|}_{\triangleq \Delta R_{k,t}^2}
\end{aligned}
\tag{19}
$$

In the above inequality, we implicitly use the convention that $\boldsymbol{\mu}_0^N = \boldsymbol{\mu}_0$, $g_0^N = g_0$. Equality $(a)$ is essentially a telescoping series.

**Step 3:** We shall now focus on the first term of $\Delta R_{k,t}^2$. Note that,

$$
\begin{aligned}
&\mathbb{E}[\tilde{r}^{\mathrm{MF}}(\boldsymbol{\mu}_{k+1}^N, g_{k+1}^N, \pi_{k+1:t})] \\
&= \mathbb{E}\left[ \mathbb{E}\left[ \tilde{r}^{\mathrm{MF}}(\boldsymbol{\mu}_{k+1}^N, g_{k+1}^N, \pi_{k+1:t}) \big| \boldsymbol{\mu}_k^N, g_k^N, \boldsymbol{\nu}_k^N \right] \right] \\
&\overset{(a)}{=} \mathbb{E}\left[ \mathbb{E}\left[ \sum_{g \in \mathcal{G}} \tilde{r}^{\mathrm{MF}}(\boldsymbol{\mu}_{k+1}^N, g, \pi_{k+1:t}) P_G(\boldsymbol{\mu}_k^N, g_k^N, \boldsymbol{\nu}_k^N)(g) \big| \boldsymbol{\mu}_k^N, g_k^N, \boldsymbol{\nu}_k^N \right] \right] \\
&= \mathbb{E}\left[ \sum_{g \in \mathcal{G}} \tilde{r}^{\mathrm{MF}}(\boldsymbol{\mu}_{k+1}^N, g, \pi_{k+1:t}) P_G(\boldsymbol{\mu}_k^N, g_k^N, \boldsymbol{\nu}_k^N)(g) \right]
\end{aligned}
$$

where $(a)$ follows from the fact that $\boldsymbol{\mu}_{k+1}^N, g_{k+1}^N$ are conditionally independent given $\boldsymbol{\mu}_k^N, g_k^N, \boldsymbol{\nu}_k^N$ and $g_{k+1}^N \sim P_G(\boldsymbol{\mu}_k^N, g_k^N, \boldsymbol{\nu}_k^N)$.

**Step 4:** Using Lemma 9 (stated in Appendix A.2), we can expand the second term of $\Delta R_{k,t}^2$ as follows.

$$
\mathbb{E}[\tilde{r}^{\mathrm{MF}}(\boldsymbol{\mu}_k^N, g_k^N, \pi_{k:t})] = \mathbb{E}\left[ \sum_{g \in \mathcal{G}} P_G(\boldsymbol{\mu}_k^N, g_k^N, \nu^{\mathrm{MF}}(\boldsymbol{\mu}_k^N, g_k^N, \pi_k))(g) \times \tilde{r}^{\mathrm{MF}}(P^{\mathrm{MF}}(\boldsymbol{\mu}_k^N, g_k^N, \pi_k), g, \pi_{k+1:t}) \right]
$$

**Step 5:** Lemma 8 (Appendix A.2) shows that $\tilde{r}^{\mathrm{MF}}(\cdot, g, \pi_{k+1:t})$ is Lipschitz continuous with a $(k,t)$-dependent Lipschitz parameter for any $g, \pi_{k+1:t}$. Therefore, one can write the following.

$$
\left| \tilde{r}^{\mathrm{MF}}(\boldsymbol{\mu}_{k+1}^N, g, \pi_{k+1:t}) - \tilde{r}^{\mathrm{MF}}(P^{\mathrm{MF}}(\boldsymbol{\mu}_k^N, g_k^N, \pi_k), g, \pi_{k+1:t}) \right| = \mathcal{O}\left( \left| \boldsymbol{\mu}_{k+1}^N - P^{\mathrm{MF}}(\boldsymbol{\mu}_k^N, g_k^N, \pi_k) \right|_1 \right)
$$

The leading constants in the above bound are $(k,t)$ dependent. In Lemma 12 (Appendix A.3), we further show that,

$$
\mathbb{E}\left| \boldsymbol{\mu}_{k+1}^N - P^{\mathrm{MF}}(\boldsymbol{\mu}_k^N, g_k^N, \pi_k) \right|_1 = \mathcal{O}\left( \frac{1}{\sqrt{N}} \left[ \sqrt{|\mathcal{X}|} + \sqrt{|\mathcal{U}|} \right] \right)
$$

Using Assumption 1(d), one can write the following for any choice of $\boldsymbol{\mu}_k^N, g_k^N, \boldsymbol{\nu}_k^N$ and $\pi_k$.

$$
|P_G(\boldsymbol{\mu}_k^N, g_k^N, \boldsymbol{\nu}_k^N) - P_G(\boldsymbol{\mu}_k^N, g_k^N, \nu^{\mathrm{MF}}(\boldsymbol{\mu}_k^N, g_k^N, \pi_k))|_1 \leq L_G \left| \boldsymbol{\nu}_k^N - \nu^{\mathrm{MF}}(\boldsymbol{\mu}_k^N, g_k^N, \pi_k) \right|_1
$$

Moreover, in Lemma 11 (Appendix A.3), we show that,

$$
\mathbb{E}\left| \boldsymbol{\nu}_k^N - \nu^{\mathrm{MF}}(\boldsymbol{\mu}_k^N, g_k^N, \pi_k) \right|_1 = \mathcal{O}\left( \frac{\sqrt{|\mathcal{U}|}}{\sqrt{N}} \right)
$$

Combining all these results with the expansions obtained in steps 3 and 4, we finally conclude that,

$$\Delta R_{k,t}^2 = \mathcal{O}\left(\frac{1}{\sqrt{N}}\left[\sqrt{|\mathcal{X}|} + \sqrt{|\mathcal{U}|}\right]\right)$$

where the leading constants are $(k, t)$-dependent.

**Step 6:** Substituting the bound of $\Delta R_{k,t}^2$ in (19), we can obtain a bound for $\Delta R_t^2$ which, combined with the previously obtained bound for $\Delta R_t^1$, yields a bound for $\Delta R_t$. We finally obtain the difference between $N$-agent and infinite agent value corresponding to $\boldsymbol{\pi}$ by computing the upper bound of the sum $\sum_{t=0}^\infty \gamma^t \Delta R_t$. We would like to point out here that the leading coefficient of $\Delta R_t$ turns out to be an exponential function of $t$. In order for the sum to converge, one must have $\gamma$ to be sufficiently small.

**Step 7:** We establish the desired result by observing that $|\sup_{\boldsymbol{\pi}} V_N(\boldsymbol{x}_0, g_0, \boldsymbol{\pi}) - \sup_{\boldsymbol{\pi}} V_\infty(\boldsymbol{\mu}_0, g_0, \boldsymbol{\pi})| \leq \sup_{\boldsymbol{\pi}} |V_N(\boldsymbol{x}_0, g_0, \boldsymbol{\pi}) - V_\infty(\boldsymbol{\mu}_0, g_0, \boldsymbol{\pi})|$ where the suprema are calculated over the class of all admissible policy sequences $\Pi^\infty$.

We would like to conclude this section by pointing out how our proof technique fundamentally differs from the techniques used in the existing papers such as Mondal et al. (2022a); Gu et al. (2021) where the common global state is not considered. Note that in absence[2] of the shared state-space $\mathcal{G}$, the infinite agent state distributions $\{\boldsymbol{\mu}_t\}_{t \in \{0,1,\cdots\}}$ are all deterministic and the error $\Delta R_t^2$ defined in (18) can be bounded as,

$$\begin{aligned}
\Delta R_t^2 &\triangleq \left|\mathbb{E}[r^{\mathrm{MF}}(\boldsymbol{\mu}_t^N, \pi_t)] - \mathbb{E}[r^{\mathrm{MF}}(\boldsymbol{\mu}_t, \pi_t)]\right| \\
&\leq \mathbb{E}\left|r^{\mathrm{MF}}(\boldsymbol{\mu}_t^N, \pi_t) - r^{\mathrm{MF}}(\boldsymbol{\mu}_t, \pi_t)\right| \overset{(a)}{=} \mathcal{O}\left(\mathbb{E}|\boldsymbol{\mu}_t^N - \boldsymbol{\mu}_t|_1\right)
\end{aligned} \tag{20}$$

Relation $(a)$ is proven by establishing that $r^{\mathrm{MF}}(\cdot, \pi_t)$ is Lipschitz continuous. The error term $\mathbb{E}|\boldsymbol{\mu}_t^N - \boldsymbol{\mu}_t|_1$ shown in (20) is bounded by establishing a recursion on $t$. Unfortunately, such simplified recursion does not hold once the global state space is introduced. In our analysis, we rather need to keep track of the whole trajectory induced by the policy sequence, $\pi_{k:t}$ for $k < t$. The need to trace out convoluted trajectories and the lack of any simplified recursion makes our analysis much more difficult and incompatible with the proof techniques available in the literature.

# 7 Algorithm to solve MFC

In this section, we present a natural policy gradient (NPG) based algorithm to obtain an optimal policy for the mean-field control (MFC) problem. As clarified in section 3, the size of the agent population is presumed to be infinite in an MFC framework and we can arbitrarily select any agent to be a representative. Assume that at time $t$ the representative agent chooses an action $u_t \in \mathcal{U}$ after observing its local state $x_t \in \mathcal{X}$, the global state $g_t \in \mathcal{G}$, and the distribution of local states of all agents denoted by $\boldsymbol{\mu}_t \in \Delta(\mathcal{X})$. The goal of MFC, therefore, reduces to maximizing the discounted expected reward of this representative agent. Clearly, it is a single agent Markov Decision Process (MDP) with a state space of $\mathcal{X} \times \Delta(\mathcal{X}) \times \mathcal{G}$ and action space $\mathcal{U}$.

Without loss of generality, we can assume that the optimal policy sequence is stationary (Puterman, 2014, Theorem 6.2.12). Note that the collection of admissible policies is denoted by the set $\Pi$. We assume that the elements of $\Pi$ are characterized by a $d$-dimensional parameter $\Phi \in \mathbb{R}^d$. In the forthcoming discussion, a policy with the parameter $\Phi$ will be denoted as $\pi_\Phi$ whereas, with a slight abuse of notations, the stationary sequence generated by it will also be denoted as $\pi_\Phi$. Let the Q-value corresponding to the policy $\pi_\Phi$ be defined as follows $\forall x, \boldsymbol{\mu}, g, u$.

$$Q_\Phi(x, \boldsymbol{\mu}, g, u) \triangleq \mathbb{E}\left[\sum_{t=0}^\infty \gamma^t r(x_t, u_t, \boldsymbol{\mu}_t, g_t, \boldsymbol{\nu}_t)\Big| x_0 = x, \boldsymbol{\mu}_0 = \boldsymbol{\mu}, g_0 = g, u_0 = u\right] \tag{21}$$

where the quantities $\boldsymbol{\mu}_t$ and $\boldsymbol{\nu}_t$ are recursively calculated by applying (8) and (7) respectively. Moreover, the expectation is taken over $x_{t+1} \sim P(x_t, u_t, \boldsymbol{\mu}_t, g_t, \boldsymbol{\nu}_t)$, $u_t \sim \pi_\Phi(x_t, \boldsymbol{\mu}_t, g_t)$, and $g_{t+1} \sim P_G(\boldsymbol{\mu}_t, g_t, \boldsymbol{\nu}_t)$, $\forall t \geq 0$.

---

[2]This can be considered a special case of our model by imposing $|\mathcal{G}| = 1$.

The advantage function corresponding to $\pi_\Phi$ is defined as follows.

$$A_\Phi(x, \boldsymbol{\mu}, g, u) \triangleq Q_\Phi(x, \boldsymbol{\mu}, g, u) - \mathbb{E}[Q_\Phi(x, \boldsymbol{\mu}, g, u')] \tag{22}$$

where the expectation is over $u' \sim \pi_\Phi(x, \boldsymbol{\mu}, g)$.

Let $\boldsymbol{\mu}_0 \in \Delta(\mathcal{X})$ be the distribution of initial local states and $g_0$ be the initial global state. Define $V_\infty^*(\boldsymbol{\mu}_0, g_0) \triangleq \sup_{\Phi \in \mathbb{R}^d} V_\infty(\boldsymbol{\mu}_0, g_0, \pi_\Phi)$ to be the optimal mean-field value obtained over the collection of admissible policies, $\Pi$ corresponding to the initial state $(\boldsymbol{\mu}_0, g_0)$. Assume that $\{\Phi_j\}_{j=1}^J$ is a sequence of $d-$dimensional parameters generated by the NPG algorithm (Liu et al., 2020; Agarwal et al., 2021) as follows.

$$\Phi_{j+1} = \Phi_j + \eta \mathbf{w}_j, \mathbf{w}_j \triangleq \arg\min_{\mathbf{w} \in \mathbb{R}^d} L_{\zeta^{\Phi_j}_{(\boldsymbol{\mu}_0, g_0)}}(\mathbf{w}, \Phi_j) \tag{23}$$

where $\eta > 0$ is the learning parameter. The function $L_{\zeta^{\Phi_j}_{(\boldsymbol{\mu}_0, g_0)}}$ and the occupancy measure $\zeta^{\Phi_j}_{(\boldsymbol{\mu}_0, g_0)}$ are defined below.

$$L_{\zeta^{\Phi'}_{(\boldsymbol{\mu}_0, g_0)}}(\mathbf{w}, \Phi) \triangleq \mathbb{E}_{(x, \boldsymbol{\mu}, g, u) \sim \zeta^{\Phi'}_{(\boldsymbol{\mu}_0, g_0)}} \left[ \left( A_\Phi(x, \boldsymbol{\mu}, g, u) - (1 - \gamma) \mathbf{w}^\mathrm{T} \nabla_\Phi \log \pi_\Phi(x, \boldsymbol{\mu}, g)(u) \right)^2 \right], \tag{24}$$

$$\zeta^{\Phi'}_{(\boldsymbol{\mu}_0, g_0)}(x, \boldsymbol{\mu}, g, u) \triangleq \sum_{\tau=0}^\infty \gamma^\tau \mathbb{P}(x_\tau = x, \boldsymbol{\mu}_\tau = \boldsymbol{\mu}, g_\tau = g, u_\tau = u | x_0 = x, \boldsymbol{\mu}_0 = \boldsymbol{\mu}, g_0 = g, u_0 = u, \boldsymbol{\pi}_{\Phi'})(1 - \gamma)$$

$$\tag{25}$$

Note that in order to calculate the update direction, $\mathbf{w}_j$ at the $j$-th NPG update (23), we must solve another optimization problem over $\mathbb{R}^d$. The second minimization problem can be handled via a stochastic gradient descent (SGD) approach. Particularly, for a given $\Phi_j \in \mathbb{R}^d$, the minimizer of $L_{\zeta^{\Phi_j}_{(\boldsymbol{\mu}_0, g_0)}}(\cdot, \Phi)$ can be obtained via the following SGD updates: $\mathbf{w}_{j,l+1} = \mathbf{w}_{j,l} - \alpha \mathbf{h}_{j,l}$ (Liu et al., 2020) where $\alpha > 0$ is the learning parameter for the sub-problem and $\mathbf{h}_{j,l}$, the gradient at the $l$-th iteration is given as follows.

$$\mathbf{h}_{j,l} \triangleq \left( \mathbf{w}_{j,l}^\mathrm{T} \nabla_{\Phi_j} \log \pi_{\Phi_j}(x, \boldsymbol{\mu}, g)(u) - \frac{1}{1 - \gamma} \hat{A}_{\Phi_j}(x, \boldsymbol{\mu}, g, u) \right) \nabla_{\Phi_j} \log \pi_{\Phi_j}(x, \boldsymbol{\mu}, g)(u) \tag{26}$$

where the tuple $(x, \boldsymbol{\mu}, g, u)$ is sampled from the occupancy measure $\zeta^{\Phi_j}_{\boldsymbol{\lambda}_0}$, and $\hat{A}_{\Phi_j}(x, \boldsymbol{\mu}, g, u)$ denotes a unbiased estimator of $A_{\Phi_j}(x, \boldsymbol{\mu}, g, u)$. The sampling procedure and the method to obtain the estimator has been described in Algorithm 2 in Appendix V. It is to be clarified that Algorithm 2 is based on Algorithm 3 of (Agarwal et al., 2021). The whole NPG procedure is summarized in Algorithm 1.

Based on the result (Theorem 4.9) of (Liu et al., 2020), in Lemma 1, we establish the convergence of Algorithm 1 and characterize its sample complexity. However, the following assumptions are needed to prove the Lemma. The assumptions stated below are respectively similar to Assumptions 2.1, 4.2, 4.4 of (Liu et al., 2020).

**Assumption 4.** $\forall \Phi \in \mathbb{R}^d$, $\forall (\boldsymbol{\mu}_0, g_0) \in \Delta(\mathcal{X}) \times \mathcal{G}$, for some $\chi > 0$, $F_{(\boldsymbol{\mu}_0, g_0)}(\Phi) - \chi I_d$ is positive semi-definite where $F_{(\boldsymbol{\mu}_0, g_0)}(\Phi)$ is defined as stated below.

$$F_{(\boldsymbol{\mu}_0, g_0)}(\Phi) \triangleq \mathbb{E}_{(x, \boldsymbol{\mu}, g, u) \sim \zeta^{\Phi}_{(\boldsymbol{\mu}_0, g_0)}} \left[ \{ \nabla_\Phi \pi_\Phi(x, \boldsymbol{\mu}, g)(u) \} \times \{ \nabla_\Phi \log \pi_\Phi(x, \boldsymbol{\mu}, g)(u) \}^\mathrm{T} \right]$$

**Assumption 5.** $\forall \Phi \in \mathbb{R}^d$, $\forall \boldsymbol{\mu} \in \Delta(\mathcal{X})$, $\forall x \in \mathcal{X}$, $\forall g \in \mathcal{G}$, $\forall u \in \mathcal{U}$, the following holds

$$|\nabla_\Phi \log \pi_\Phi(x, \boldsymbol{\mu}, g)(u)|_1 \leq G$$

for some positive constant $G$.

**Assumption 6.** $\forall \Phi_1, \Phi_2 \in \mathbb{R}^d$, $\forall \boldsymbol{\mu} \in \Delta(\mathcal{X})$, $\forall x \in \mathcal{X}$, $\forall g \in \mathcal{G}$, $\forall u \in \mathcal{U}$, the following holds,

$$|\nabla_{\Phi_1} \log \pi_{\Phi_1}(x, \boldsymbol{\mu}, g)(u) - \nabla_{\Phi_2} \log \pi_{\Phi_2}(x, \boldsymbol{\mu}, g)(u)|_1 \leq M |\Phi_1 - \Phi_2|_1$$

for some positive constant $M$.

---

**Algorithm 1** Natural Policy Gradient

---

1: **Input:** $\eta, \alpha$: Learning rates, $J, L$: Number of execution steps
       $\mathbf{w}_0, \Phi_0$: Initial parameters,
       $\boldsymbol{\mu}_0$: Initial local state distribution
       $g_0$: Initial global state
2: **Initialization:** $\Phi \leftarrow \Phi_0$
3: **for** $j \in \{0, 1, \cdots, J-1\}$ **do**
4:    $\mathbf{w}_{j,0} \leftarrow \mathbf{w}_0$
5:    **for** $l \in \{0, 1, \cdots, L-1\}$ **do**
6:       Sample $(x, \boldsymbol{\mu}, g, u) \sim \zeta^{\Phi_j}_{(\boldsymbol{\mu}_0, g_0)}$ and $\hat{A}_{\Phi_j}(x, \boldsymbol{\mu}, g, u)$ using Algorithm 2
7:       Compute $\mathbf{h}_{j,l}$ using (26)
8:       $\mathbf{w}_{j,l+1} \leftarrow \mathbf{w}_{j,l} - \alpha \mathbf{h}_{j,l}$
9:    **end for**
10:   $\mathbf{w}_j \leftarrow \frac{1}{L} \sum_{l=1}^{L} \mathbf{w}_{j,l}$
11:   $\Phi_{j+1} \leftarrow \Phi_j + \eta \mathbf{w}_j$
12: **end for**
13: **Output:** $\{\Phi_1, \cdots, \Phi_J\}$: Policy parameters

---

**Assumption 7.** $\forall \Phi \in \mathbb{R}^d$, $\forall(\boldsymbol{\mu}_0, g_0) \in \Delta(\mathcal{X}) \times \mathcal{G}$,

$$L_{\zeta^{\Phi^*}_{(\boldsymbol{\mu}_0, g_0)}}(\mathbf{w}^*_\Phi, \Phi) \leq \epsilon_{\text{bias}}, \quad \mathbf{w}^*_\Phi \triangleq \arg\min_{\mathbf{w} \in \mathbb{R}^d} L_{\zeta^\Phi_{(\boldsymbol{\mu}_0, g_0)}}(\mathbf{w}, \Phi)$$

*where $\Phi^*$ is the parameter of the optimal policy.*

**Remark 1.** *Note that Assumption 7 is trivially satisfied with $\epsilon_{\text{bias}} = 2M_R/(1-\gamma)$. Assume that, $\mathcal{U} = \{1, 2\}$, and a restricted class of parameterized policies is defined as follows.*

$$\pi_\Phi(x, \boldsymbol{\mu}, g) = \left[\frac{\exp(\Phi)}{1 + \exp(\Phi)} \quad \frac{1}{1 + \exp(\Phi)}\right]^T$$

*where $(x, \boldsymbol{\mu}, g) \in \mathcal{X} \times \Delta(\mathcal{X}) \times \mathcal{G}$, and $\Phi \in [-\xi, \xi]$ for some constant, $\xi > 0$. One can check that, for the set of policies stated above, assumptions $4-6$ are satisfied with $\chi = \exp(2\xi)/(1 + \exp(\xi))^4$, $G = 1$, and $M = 1/4$.*

We are now ready to state the convergence result which is a direct application of Theorem 4.9 of (Liu et al., 2020).

**Lemma 1.** *Let $\{\Phi_j\}_{j=1}^J$ be the sequence of policy parameters obtained from Algorithm 1. If Assumptions $4-7$ hold, then the following inequality holds for $\eta = \frac{(1-\gamma)^2\chi^2}{4G^2 M_R M}$, $\alpha = \frac{1}{4G^2}$, $J = \mathcal{O}\left(\frac{1}{(1-\gamma)^2\epsilon}\right)$, $L = \mathcal{O}\left(\frac{1}{(1-\gamma)^4\epsilon^2}\right)$,*

$$V^*_\infty(\boldsymbol{\mu}_0, g_0) - \frac{1}{J}\sum_{j=1}^J \mathbb{E}\left[V_\infty(\boldsymbol{\mu}_0, g_0, \pi_{\Phi_j})\right] \leq \frac{\sqrt{\epsilon_{\text{bias}}}}{1 - \gamma} + \epsilon,$$

*for arbitrary initial parameter $\Phi_0$, initial local state distribution $\boldsymbol{\mu}_0 \in \Delta(\mathcal{X})$ and initial global state $g_0$. The parameters $\{M_R, \chi, G, M\}$ are defined in Assumptions 1, 4, 5, 6 respectively. The term $\epsilon_{\text{bias}}$ is a positive constant. The sample complexity of Algorithm 1 is $\mathcal{O}(\epsilon^{-3})$.*

The term $\epsilon_{\text{bias}}$ introduced in Lemma 1 is termed as the expressivity error of the policy class $\Pi$ parameterized by the $d-$dimensional parameter. For dense neural network-based policies, $\epsilon_{\text{bias}}$ appears to be small (Liu et al., 2020).

Lemma 1 states that Algorithm 1 can approximate the optimal mean-field value, $V^*_\infty(\boldsymbol{\mu}_0, g_0)$ for any $(\boldsymbol{\mu}_0, g_0) \in \Delta(\mathcal{X}) \times \mathcal{G}$ with an error bound of $\epsilon$, and a sample complexity of $\mathcal{O}(\epsilon^{-3})$. The constant term hiding in $\mathcal{O}(\cdot)$ is dependent on the parameters $\{M_R, \chi, G, M\}$ defined in Assumptions 1, 4, 5, 6. Combining Lemma 1 with Theorem 1, we now arrive at the following result.

**Theorem 3.** *Let $\boldsymbol{x}_0 \triangleq \{x_0^i\}_{i \in \{1, \cdots, N\}}$ and $g_0$ be the initial local and global states respectively and $\boldsymbol{\mu}_0$ be the empirical distribution of $\boldsymbol{x}_0$. Assume that $\{\Phi_j\}_{j=1}^J$ are the policy parameters generated from Algorithm 1 corresponding to the initial condition $(\boldsymbol{\mu}_0, g_0, \Phi_0)$, and and the set of policies, $\Pi$ follows Assumption 2. If assumptions 1, 2, 4 - 7 are true, then, $\forall \epsilon > 0$, the following inequality holds for the choice of the parameters $\{\eta, \alpha, J, L\}$ stated in Lemma 1.*

$$\left| V_N^*(\boldsymbol{x}_0, g_0) - \frac{1}{J} \sum_{j=1}^J \mathbb{E}\left[ V_\infty(\boldsymbol{\mu}_0, g_0, \pi_{\Phi_j}) \right] \right| \leq \frac{\sqrt{\epsilon_{\text{bias}}}}{1 - \gamma} + C \max\{e, \epsilon\}$$

$$\text{where } V_N^*(\boldsymbol{x}_0, g_0) = \sup_{\Phi \in \mathbb{R}^d} V_N(\boldsymbol{x}_0, g_0, \pi_\Phi), \; e \triangleq \frac{1}{\sqrt{N}} \left[ \sqrt{|\mathcal{X}|} + \sqrt{|\mathcal{U}|} \right]$$

*whenever $\gamma S_P < 1$ where $S_P$ is defined in Theorem 1. The parameter, $C$ is a constant and the parameter $\epsilon_{\text{bias}}$ is defined in Lemma 1. The sample complexity of the process is $\mathcal{O}(\epsilon^{-3})$.*

*Proof.* Note that,

$$\left| V_N^*(\boldsymbol{x}_0, g_0) - \frac{1}{J} \sum_{j=1}^J V_\infty(\boldsymbol{\mu}_0, g_0, \pi_{\Phi_j}) \right| \leq \left| V_N^*(\boldsymbol{x}_0, g_0) - V_\infty^*(\boldsymbol{\mu}_0, g_0) \right| + \left| V_\infty^*(\boldsymbol{\mu}_0, g_0) - \frac{1}{J} \sum_{j=1}^J V_\infty(\boldsymbol{\mu}_0, g_0, \pi_{\Phi_j}) \right|$$

Applying Theorem 1, the first term can be bounded by $C'e$ for some constant $C'$. The second term can be bounded by $\sqrt{\epsilon_{\text{bias}}}/(1 - \gamma) + \epsilon$ with a sample complexity of $\mathcal{O}(\epsilon^{-3})$ (Lemma 1). Using $C = 2 \max\{C', 1\}$, we conclude. $\qquad\square$

Theorem 3 states that Algorithm 1 generates a policy such that its associated $N-$agent value is at most $\mathcal{O}(\max\{e, \epsilon\})$ error away from the optimal $N-$agent value. Additionally, it also guarantees that such a policy can be obtained with a sample complexity of $\mathcal{O}(\epsilon^{-3})$ where $\epsilon$ is an arbitrarily chosen positive number. We would like to point out that in Theorem 3, $e = \left[ \sqrt{|\mathcal{X}|} + \sqrt{|\mathcal{U}|} \right] / \sqrt{N}$. However, if we assume that Assumption 3 holds in addition to all the assumptions mentioned in Theorem 3, then using Theorem 2, one can similarly show that the average difference between the optimal MARL value and the sequence of values generated by Algorithm 1 is $\mathcal{O}(\max\{e, \epsilon\})$ where $e = \sqrt{|\mathcal{X}|}/\sqrt{N}$ and such an error can be achieved with a sample complexity of $\mathcal{O}(\epsilon^{-3})$ where $\epsilon > 0$ is a tunable parameter.

## 8 Experiments

For numerical experiment, we shall consider a variant of the model described in Subramanian and Mahajan (2019). The model consists of $N$ collaborative firms operated by a single company. All of them produces the same product but with different quality. At time $t$, the $i$-th firm may decide to improve the current quality of its product (denoted by $x_t^i$) by investing $\lambda_R$ amount of money. The action corresponding to investment is denoted as $u_t^i = 1$ whereas no investment is indicated as $u_t^i = 0$. Clearly, the action space is $\mathcal{U} = \{0, 1\}$. On the other hand, the state space is assumed to be $\mathcal{X} = \{0, 1, \cdots, Q - 1\}$. The state transition model of the $i$-th firm is as follows.

$$x_{t+1}^i = \begin{cases} x_t^i & \text{if } u_t^i = 0 \\ x_t^i + \left\lfloor \chi \left( Q - 1 - x_t^i \right) \left( 1 - \frac{\bar{\boldsymbol{\mu}}_t^N}{Q} \right) \right\rfloor & \text{elsewhere} \end{cases} \tag{27}$$

where $\chi$ is a uniform random variable in $[0, 1]$, $\boldsymbol{\mu}_t^N$ is the empirical distribution of $\boldsymbol{x}_t^N \triangleq \{x_t^i\}_{i \in \{1, \cdots, N\}}$ and $\bar{\boldsymbol{\mu}}_t^N$ is the mean of $\boldsymbol{\mu}_t^N$. The intuition for (27) is that the product quality does not change when no investment is made and it improves probabilistically otherwise. However, the improvement also depends on the average product quality, $\bar{\boldsymbol{\mu}}_t^N$ of the economy. Specifically, it is harder to improve the quality when $\bar{\boldsymbol{\mu}}_t^N$ is high. The

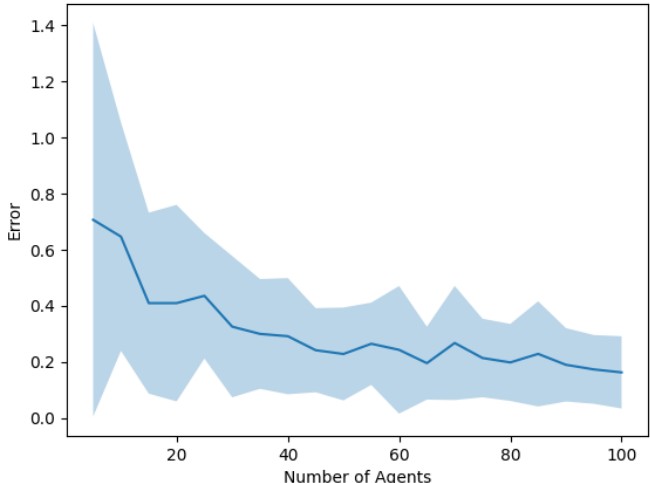

Figure 1: The error as a function of $N$. The bold line and half-width of the shaded region respectively denote the mean and standard deviation of the error obtained over 25 experiments conducted with different random seeds. The chosen model parameters are as follows: $\lambda_0 = 1$, $\lambda_1 = 0.5$, $\beta_R = 0.5$, $\lambda_R = 0.5$, $Q = 10$.

term $(1 - \bar{\boldsymbol{\mu}}_t^N/Q)$ signifies the resistance to improvement. The reward experienced by the $i$-th firm at time $t$ is expressed as follows.

$$r(x_t^i, u_t^i, \boldsymbol{\mu}_t^N, \alpha_t^N) = \alpha_t^N x_t^i - \beta_R \bar{\boldsymbol{\mu}}_t^N - \lambda_R u_t^i \tag{28}$$

The first term $\alpha_t^N x_t^i$ is the revenue earned by the $i$-th firm. One can interpret $\alpha_t^N$ as the price per unit quality at time $t$. Clearly, $\alpha_t^N$ plays the role of shared global state. In (Subramanian and Mahajan, 2019), $\alpha_t^N$ was taken to be a constant. Here we assume it to be linearly dependent on the average product quality at $t-1$ i.e., $\alpha_t^N = \lambda_0(1 - \lambda_1(\bar{\boldsymbol{\mu}}_{t-1}^N/Q))$ where $\lambda_0, \lambda_1$ are arbitrary positive constants. The intuition is that the average quality at $t-1$ influences the price at $t$. Specifically, as $\bar{\boldsymbol{\mu}}_{t-1}$ rises, the firms can charge less price per quality, $\alpha_t^N$. The second term, $\beta_R \bar{\boldsymbol{\mu}}_t^N$ denotes the cost incurred due to the average product quality, $\bar{\boldsymbol{\mu}}_t^N$. Finally, the third term, $\lambda_R u_t^i$ indicates the investment cost. Let $\pi^*$ be the policy given by the Algorithm 1. We define the error as,

$$\text{error} \triangleq |V_N(\boldsymbol{x}_0, \alpha_0, \pi^*) - V_\infty(\boldsymbol{\mu}_0, \alpha_0, \pi^*)|$$

where $\boldsymbol{x}_0 = \{x_0^i\}_{i \in \{1, \cdots, N\}}$ is the initial local states, $\boldsymbol{\mu}_0$ is its empirical distribution and $\alpha_0$ is the initial global price. Moreover, $V_N$ and $V_\infty$ are defined via (6) and (11) respectively. Fig. 1 plots the error as a function of $N$. We can observe[3] that the error decreases with $N$.

## 9 Conclusions

In this paper, we introduce a MARL framework where the agents, in addition to their local states that transitions conditionally independently, also possess a shared global state. As a result, the combined state transition processes of each agent becomes correlated with each other. Our contribution is to show that mean-field based approximations are valid even in presence of such correlation. We obtain the expression for the approximation error as a function of different parameters of the model and surprisingly observe that it is not dependent on the size of the global state-space. Furthermore, we designed an algorithm that approximately obtains an optimal solution to the MARL problem. Although our work shows approximation

---

[3]The code can be accessed at: https://github.itap.purdue.edu/Clan-labs/MeanFieldwithGlobalState

results in presence of correlated evolution, the presumed structure of correlation is not in the most generic form. Specifically, we assumed that the states of each agent can be segregated into two part−one evolving conditionally independently and the other being identical to every agent. Whether mean-field control techniques work in presence of more general form of correlated evolution is an important question that needs to be investigated in the future.

## A    Proof of Theorem 1

The following helper lemmas are needed to establish the main result.

### A.1    Continuity Lemmas

In the following lemmas, $\pi \in \Pi$ is an arbitrary policy and $\boldsymbol{\mu}, \bar{\boldsymbol{\mu}} \in \Delta(\mathcal{X})$ are arbitrary local state distributions.

**Lemma 2.** *If $\nu^{\mathrm{MF}}(\cdot, \cdot, \cdot)$ is defined by (7), then the following relation holds $\forall g \in \mathcal{G}$.*

$$|\nu^{\mathrm{MF}}(\boldsymbol{\mu}, g, \pi) - \nu^{\mathrm{MF}}(\bar{\boldsymbol{\mu}}, g, \pi)| \leq (1 + L_Q)|\boldsymbol{\mu} - \bar{\boldsymbol{\mu}}|_1$$

**Lemma 3.** *If $P^{\mathrm{MF}}(\cdot, \cdot, \cdot)$ is defined by (8), then the following relation holds $\forall g \in \mathcal{G}$.*

$$|P^{\mathrm{MF}}(\boldsymbol{\mu}, g, \pi) - P^{\mathrm{MF}}(\bar{\boldsymbol{\mu}}, g, \pi)|_1 \leq S_P|\boldsymbol{\mu} - \bar{\boldsymbol{\mu}}|_1$$

*where $S_P \triangleq 1 + 2L_P + L_Q(1 + L_P)$.*

**Lemma 4.** *If $P_G^{\mathrm{MF}}(\cdot, \cdot, \cdot)$ is defined by (9), then the following relation holds $\forall g \in \mathcal{G}$.*

$$|P_G^{\mathrm{MF}}(\boldsymbol{\mu}, g, \pi) - P_G^{\mathrm{MF}}(\bar{\boldsymbol{\mu}}, g, \pi)|_1 \leq S_G|\boldsymbol{\mu} - \bar{\boldsymbol{\mu}}|_1$$

*where $S_G \triangleq L_G(2 + L_Q)$.*

**Lemma 5.** *If $r^{\mathrm{MF}}(\cdot, \cdot, \cdot)$ is defined by (10), then the following relation holds $\forall g \in \mathcal{G}$.*

$$|r^{\mathrm{MF}}(\boldsymbol{\mu}, g, \pi) - r^{\mathrm{MF}}(\bar{\boldsymbol{\mu}}, g, \pi)| \leq S_R|\boldsymbol{\mu} - \bar{\boldsymbol{\mu}}|_1$$

*where $S_R \triangleq M_R + 2L_R + L_Q(M + L_R)$.*

The lemmas stated above exhibit that the functions $\nu^{\mathrm{MF}}(\cdot, \cdot, \cdot)$, $P^{\mathrm{MF}}(\cdot, \cdot, \cdot)$, $P_G^{\mathrm{MF}}(\cdot, \cdot, \cdot)$ and $r^{\mathrm{MF}}(\cdot, \cdot, \cdot)$ defined in section 3 are Lipschitz continuous. Proofs of Lemma $2 - 5$ are relegated to Appendix $B-E$.

### A.2    Continuity of Some Relevant Functions

Let $\{\pi_l\}_{l \in \{0,1,\cdots\}} \in \Pi^\infty$ be an arbitrary policy sequence and $\{g_l\}_{l \in \{0,1,\cdots\}} \in \mathcal{G}^\infty$ be a given sequence of global states. Recall the definition of $P^{\mathrm{MF}}$ given in (8). For every $\boldsymbol{\mu}_l \in \Delta(\mathcal{X})$, we define the following.

$$\tilde{P}^{\mathrm{MF}}(\boldsymbol{\mu}_l, g_{l:l+r}, \pi_{l:l+r}) \triangleq P^{\mathrm{MF}}(\cdot, g_{l+r}, \pi_{l+r}) \circ P^{\mathrm{MF}}(\cdot, g_{l+r-1}, \pi_{l+r-1}) \circ \cdots \circ P^{\mathrm{MF}}(\cdot, g_l, \pi_l)(\boldsymbol{\mu}_l) \qquad (29)$$

where $\circ$ denotes function composition and $l, r \in \{0, 1, \cdots\}$. Note that, using $r = 0$ in (29), we get $\tilde{P}^{\mathrm{MF}}(\boldsymbol{\mu}_l, g_{l:l}, \pi_{l:l}) = P^{\mathrm{MF}}(\boldsymbol{\mu}_l, g_l, \pi_l)$. Similarly, using the definition of $P_G^{\mathrm{MF}}$ given in (9), we define the following $\forall \boldsymbol{\mu}_l \in \Delta(\mathcal{X})$.

$$\tilde{P}_G^{\mathrm{MF}}(\boldsymbol{\mu}_l, g_{l:l+r}, \pi_{l:l+r})(g_{l+r+1}) \triangleq P_G^{\mathrm{MF}}(\boldsymbol{\mu}_l, g_l, \pi_l)(g_{l+1}) \times P_G^{\mathrm{MF}}(\tilde{P}^{\mathrm{MF}}(\boldsymbol{\mu}_l, g_{l:l}, \pi_{l:l}), g_{l+1}, \pi_{l+1})(g_{l+2}) \times$$
$$\cdots \times P_G^{\mathrm{MF}}(\tilde{P}^{\mathrm{MF}}(\boldsymbol{\mu}_l, g_{l:l+r-1}, \pi_{l:l+r-1}), g_{l+r}, \pi_{l+r})(g_{l+r+1}) \qquad (30)$$

Finally, using the definition of $r^{\mathrm{MF}}$ given in (10), we define the following $\forall \boldsymbol{\mu}_l \in \Delta(\mathcal{X})$.

$$\tilde{r}^{\mathrm{MF}}(\boldsymbol{\mu}_l, g_l, \pi_{l:l+r})$$
$$= \begin{cases} \sum\limits_{l+1:l+r} r^{\mathrm{MF}}(\tilde{P}^{\mathrm{MF}}(\boldsymbol{\mu}_l, g_{l:l+r-1}, \pi_{l:l+r-1}), g_{l+r}, \pi_{l+r}) \tilde{P}_G^{\mathrm{MF}}(\boldsymbol{\mu}_l, g_{l:l+r-1}, \pi_{l:l+r-1})(g_{l+r}), & \text{if } r \geq 1 \\ \\ r^{\mathrm{MF}}(\boldsymbol{\mu}_l, g_l, \pi_l) & \text{if } r = 0 \end{cases}$$
$$\qquad (31)$$

where $\sum_{l+1:l+r}$ indicates a summation operation over $\{g_{l+1}, \cdots, g_{l+r}\} \in \mathcal{G}^r$. We prove the continuity property of these newly defined functions in the following lemmas. Proofs of Lemma $6-8$ are relegated to Appendix F$-$H.

**Lemma 6.** *The following relations hold* $\forall l, r \in \{0, 1, \cdots\}$, $\forall \boldsymbol{\mu}_l, \bar{\boldsymbol{\mu}}_l \in \Delta(\mathcal{X})$, $\forall g_{l:l+r} \in \mathcal{G}^{r+1}$ *and* $\forall \pi_{l:l+r} \in \Pi^{r+1}$.

$$|\tilde{P}^{\mathrm{MF}}(\boldsymbol{\mu}_l, g_{l:l+r}, \pi_{l:l+r}) - \tilde{P}^{\mathrm{MF}}(\bar{\boldsymbol{\mu}}_l, g_{l:l+r}, \pi_{l:l+r})|_1 \le S_P^{r+1}|\boldsymbol{\mu}_l - \bar{\boldsymbol{\mu}}_l|_1$$

*where $S_P$ is defined in Lemma 3.*

Lemma 6 establishes the Lipschitz continuity of $\tilde{P}^{\mathrm{MF}}$ with respect to its first argument.

**Lemma 7.** *The following relations hold* $\forall l, r \in \{0, 1, \cdots\}$, $\forall \boldsymbol{\mu}_l, \bar{\boldsymbol{\mu}}_l \in \Delta(\mathcal{X})$, $\forall g_{l:l+r} \in \mathcal{G}^{r+1}$ *and* $\forall \pi_{l:l+r} \in \Pi^{r+1}$.

$$\sum_{l+1:l+r} \left| \tilde{P}_G^{\mathrm{MF}}(\boldsymbol{\mu}_l, g_{l:l+r}, \pi_{l:l+r}) - \tilde{P}_G^{\mathrm{MF}}(\bar{\boldsymbol{\mu}}_l, g_{l:l+r}, \pi_{l:l+r}) \right|_1 \le S_G(1 + S_P + \cdots + S_P^r)|\boldsymbol{\mu}_l - \bar{\boldsymbol{\mu}}_l|_1$$

*where $\sum_{l+1:l+r}$ indicates a summation operation over $\{g_{l+1}, \cdots, g_{l+r}\} \in \mathcal{G}^r$ for $r \ge 1$ and an identity operation for $r = 0$. The terms $S_P, S_G$ are defined in Lemma 3 and 4 respectively.*

Lemma 7 establishes the Lipschitz continuity of $\tilde{P}_G^{\mathrm{MF}}$ with respect to its first argument. Finally we establish the Lipschitz continuity of $\tilde{r}^{\mathrm{MF}}$ with respect to its first argument in the following lemma.

**Lemma 8.** *The following relations hold* $\forall l, r \in \{0, 1, \cdots\}$, $\forall \boldsymbol{\mu}_l, \bar{\boldsymbol{\mu}}_l \in \Delta(\mathcal{X})$, $\forall g_l \in \mathcal{G}$ *and* $\forall \pi_{l:l+r} \in \Pi^r$.

$$|\tilde{r}^{\mathrm{MF}}(\boldsymbol{\mu}_l, g_l, \pi_{l:l+r}) - \tilde{r}^{\mathrm{MF}}(\bar{\boldsymbol{\mu}}_l, g_l, \pi_{l:l+r})| \le \left[ \left( \frac{M_R S_G}{S_P - 1} \right) (S_P^r - 1) + S_R S_P^r \right] |\boldsymbol{\mu}_l - \bar{\boldsymbol{\mu}}_l|_1$$

*where $S_P, S_G$ and $S_R$ are defined in Lemma 3, 4 and 5 respectively.*

Finally, we prove an important property of the function $\tilde{r}^{\mathrm{MF}}$ that directly follows from its definition.

**Lemma 9.** *The following relations hold* $\forall l \in \{0, 1, \cdots\}$, $\forall r \in \{1, 2, \cdots\}$, $\forall \boldsymbol{\mu}_l \in \Delta(\mathcal{X})$, $\forall g_l \in \mathcal{G}$ *and* $\forall \pi_{l:l+r} \in \Pi^r$.

$$\tilde{r}^{\mathrm{MF}}(\boldsymbol{\mu}_l, g_l, \pi_{l:l+r}) = \sum_{g_{l+1} \in \mathcal{G}} \tilde{r}^{\mathrm{MF}}(P^{\mathrm{MF}}(\boldsymbol{\mu}_l, g_l, \pi_l), g_{l+1}, \pi_{l+1:l+r}) P_G^{\mathrm{MF}}(\boldsymbol{\mu}_l, g_l, \pi_l)(g_{l+1})$$

*where $P^{\mathrm{MF}}$ and $P_G^{\mathrm{MF}}$ are defined in (8) and (9) respectively.*

Proof of Lemma 9 is relegated to Appendix I.

### A.3 Approximation Lemmas

First we would like to state an important result that will be useful in proving the following lemmas.

**Lemma 10.** *If $\forall m \in \{1, \cdots, M\}$, $\{X_{mn}\}_{n \in \{1, \cdots, N\}}$ are independent random variables that lie in $[0, 1]$, and satisfy $\sum_{m \in \{1, \cdots, M\}} \mathbb{E}[X_{mn}] \le 1$, $\forall n \in \{1, \cdots, N\}$, then the following holds,*

$$\sum_{m=1}^{M} \mathbb{E} \left| \sum_{n=1}^{N} (X_{mn} - \mathbb{E}[X_{mn}]) \right| \le \sqrt{MN} \tag{32}$$

In the following, $\boldsymbol{\pi} \triangleq \{\pi_t\}_{t \in \{0, 1, \cdot\}}$ is an arbitrary sequence of policies, $\{\boldsymbol{\mu}_t^N, \boldsymbol{\nu}_t^N, g_t^N\}$ denote the empirical state distribution, action distribution and global state of $N$ agent system at time $t$ and $\{x_t^i, u_t^i\}$ are the state and action of $i$th agent at time $t$ induced by $\boldsymbol{\pi}$ from initial states $\boldsymbol{x}_0, g_0$.

**Lemma 11.** *The following inequality holds* $\forall t \in \{0, 1, \cdots\}$.

$$\mathbb{E}\left| \boldsymbol{\nu}_t^N - \nu^{\mathrm{MF}}(\boldsymbol{\mu}_t^N, g_t^N, \pi_t) \right|_1 \le \frac{1}{\sqrt{N}} \sqrt{|\mathcal{U}|} \tag{33}$$

**Lemma 12.** *The following inequality holds* $\forall t \in \{0, 1, \cdots\}$.

$$\mathbb{E}\left|\boldsymbol{\mu}_{t+1}^N - P^{\mathrm{MF}}(\boldsymbol{\mu}_t^N, g_t^N, \pi_t)\right|_1 \leq \frac{C_P}{\sqrt{N}}\left[\sqrt{|\mathcal{X}|} + \sqrt{|\mathcal{U}|}\right] \tag{34}$$

*where* $C_P \triangleq 2 + L_P$.

**Lemma 13.** *The following inequality holds* $\forall t \in \{0, 1, \cdots\}$.

$$\mathbb{E}\left|\frac{1}{N}\sum_{i=1}^N r(x_t^i, u_t^i, \boldsymbol{\mu}_t^N, g_t^N, \boldsymbol{\nu}_t^N) - r^{\mathrm{MF}}(\boldsymbol{\mu}_t^N, g_t^N, \pi_t)\right| \leq \frac{M_R}{\sqrt{N}} + \frac{L_R}{\sqrt{N}}\sqrt{|\mathcal{U}|}$$

The proofs of Lemma $10 - 13$ are relegated to Appendix $\mathrm{J} - \mathrm{M}$.

### A.4 Proof of the Theorem

Let, $\boldsymbol{\pi} = \{\pi_t\}_{t \in \{0,1,\cdots\}}$ be an arbitrary policy sequence. We shall use the notations introduced in section A.3. Additionally, we shall consider $\{\boldsymbol{\mu}_t, g_t\}$ as the local state distribution and the global state of the infinite agent system at time $t$. Consider the following.

$$|V_N(\boldsymbol{x}_0, g_0, \boldsymbol{\pi}) - V_\infty(\boldsymbol{\mu}_0, g_0, \boldsymbol{\pi})|$$

$$\leq \sum_{t=0}^\infty \gamma^t \mathbb{E}\left|\frac{1}{N}\sum_{i=1}^N r(x_t^i, u_t^i, \boldsymbol{\mu}_t^N, g_t^N, \boldsymbol{\nu}_t^N) - r^{\mathrm{MF}}(\boldsymbol{\mu}_t^N, g_t^N, \pi_t)\right| + \sum_{t=0}^\infty \gamma^t \underbrace{|\mathbb{E}[r^{\mathrm{MF}}(\boldsymbol{\mu}_t^N, g_t^N, \pi_t)] - \mathbb{E}[r^{\mathrm{MF}}(\boldsymbol{\mu}_t, g_t, \pi_t)]|}_{\triangleq J_t}$$

$$\overset{(a)}{\leq} \left(\frac{M_R + L_R\sqrt{|\mathcal{U}|}}{1 - \gamma}\right)\frac{1}{\sqrt{N}} + \sum_{t=0}^\infty \gamma^t J_t$$

$$\tag{35}$$

Inequality $(a)$ follows from Lemma 13. Note that, using the definition (31), we can write the following.

$$J_t = \left|\mathbb{E}[\tilde{r}^{\mathrm{MF}}(\boldsymbol{\mu}_t^N, g_t^N, \pi_{t:t})] - \mathbb{E}[\tilde{r}^{\mathrm{MF}}(\boldsymbol{\mu}_0, g_0, \pi_{0:t})]\right|$$

$$\leq \sum_{k=0}^{t-1} \left|\mathbb{E}[\tilde{r}^{\mathrm{MF}}(\boldsymbol{\mu}_{k+1}^N, g_{k+1}^N, \pi_{k+1:t})] - \mathbb{E}[\tilde{r}^{\mathrm{MF}}(\boldsymbol{\mu}_k^N, g_k^N, \pi_{k:t})]\right|$$

$$\overset{(a)}{\leq} \sum_{k=0}^{t-1} \left|\mathbb{E}[\tilde{r}^{\mathrm{MF}}(\boldsymbol{\mu}_{k+1}^N, g_{k+1}^N, \pi_{k+1:t})] - \mathbb{E}\left[\sum_{g \in \mathcal{G}} \tilde{r}^{\mathrm{MF}}(P^{\mathrm{MF}}(\boldsymbol{\mu}_k^N, g_k^N, \pi_k), g, \pi_{k+1:t})P_G^{\mathrm{MF}}(\boldsymbol{\mu}_k^N, g_k^N, \pi_k)(g)\right]\right|$$

$$\leq \sum_{k=0}^{t-1} \left|\mathbb{E}\left[\mathbb{E}\left[\tilde{r}^{\mathrm{MF}}(\boldsymbol{\mu}_{k+1}^N, g_{k+1}^N, \pi_{k+1:t})\big|\boldsymbol{x}_k^N, g_k^N, \boldsymbol{u}_k^N\right]\right] - \mathbb{E}\left[\sum_{g \in \mathcal{G}} \tilde{r}^{\mathrm{MF}}(P^{\mathrm{MF}}(\boldsymbol{\mu}_k^N, g_k^N, \pi_k), g, \pi_{k+1:t})P_G^{\mathrm{MF}}(\boldsymbol{\mu}_k^N, g_k^N, \pi_k)(g)\right]\right|$$

$$\overset{(b)}{=} \sum_{k=0}^{t-1} \left|\mathbb{E}\left[\mathbb{E}\left[\sum_{g \in \mathcal{G}} \tilde{r}^{\mathrm{MF}}(\boldsymbol{\mu}_{k+1}^N, g, \pi_{k+1:t})P_G(\boldsymbol{\mu}_k^N, g_k^N, \boldsymbol{\nu}_k^N)(g)\big|\boldsymbol{x}_k^N, g_k^N, \boldsymbol{u}_k^N\right]\right]\right.$$

$$\left. - \mathbb{E}\left[\sum_{g \in \mathcal{G}} \tilde{r}^{\mathrm{MF}}(P^{\mathrm{MF}}(\boldsymbol{\mu}_k^N, g_k^N, \pi_k), g, \pi_{k+1:t})P_G^{\mathrm{MF}}(\boldsymbol{\mu}_k^N, g_k^N, \pi_k)(g)\right]\right|$$

$$\leq \sum_{k=0}^{t-1}\sum_{g \in \mathcal{G}} \mathbb{E}\left|\tilde{r}^{\mathrm{MF}}(\boldsymbol{\mu}_{k+1}^N, g, \pi_{k+1:t})P_G(\boldsymbol{\mu}_k^N, g_k^N, \boldsymbol{\nu}_k^N)(g) - \tilde{r}^{\mathrm{MF}}(P^{\mathrm{MF}}(\boldsymbol{\mu}_k^N, g_k^N, \pi_k), g, \pi_{k+1:t})P_G^{\mathrm{MF}}(\boldsymbol{\mu}_k^N, g_k^N, \pi_k)(g)\right|$$

$$\leq \sum_{k=0}^{t-1} J_{k,t}^1 + J_{k,2}^2$$

where we use the notation that $\boldsymbol{\mu}_0^N = \boldsymbol{\mu}_0$ and $g_0^N = g_0$. Inequality $(a)$ follows from Lemma 9 whereas $(b)$ is a result of the fact that $\boldsymbol{\mu}_{k+1}^N$ and $g_{k+1}^N$ are conditionally independent given $\boldsymbol{x}_k^N, g_k^N, \boldsymbol{u}_k^N$. Moreover, $g_{k+1}^N \sim P_G(\boldsymbol{\mu}_k^N, g_k^N, \boldsymbol{\nu}_k^N)$. The first term $J_{k,t}^1$ can be bounded as follows.

$$
\begin{aligned}
J_{k,t}^1 &\triangleq \sum_{g \in \mathcal{G}} \mathbb{E}\left[\left|\tilde{r}^{\mathrm{MF}}(\boldsymbol{\mu}_{k+1}^N, g, \pi_{k+1:t}) - \tilde{r}^{\mathrm{MF}}(P^{\mathrm{MF}}(\boldsymbol{\mu}_k^N, g_k^N, \pi_k), g, \pi_{k+1:t})\right| \times P_G(\boldsymbol{\mu}_k^N, g_k^N, \boldsymbol{\nu}_k^N)(g)\right] \\
&\overset{(a)}{\leq} \left[\left(\frac{M_R S_G}{S_P - 1}\right)(S_P^{t-k-1} - 1) + S_R S_P^{t-k-1}\right] \times \mathbb{E}\left|\boldsymbol{\mu}_{k+1}^N - P^{\mathrm{MF}}(\boldsymbol{\mu}_k^N, g_k^N, \pi_k)\right|_1 \times \underbrace{\left|P_G(\boldsymbol{\mu}_k^N, g_k^N, \boldsymbol{\nu}_k^N)\right|_1}_{=1} \\
&\overset{(b)}{\leq} C_P \left[\left(\frac{M_R S_G}{S_P - 1}\right)(S_P^{t-k-1} - 1) + S_R S_P^{t-k-1}\right] \times \frac{1}{\sqrt{N}}\left[\sqrt{|\mathcal{X}|} + \sqrt{|\mathcal{U}|}\right]
\end{aligned}
$$

Inequality $(a)$ follows from Lemma 8 whereas $(b)$ results from Lemma 12. Using (9), the second term $J_{k,t}^2$ can be bounded as follows.

$$
\begin{aligned}
J_{k,t}^2 &\triangleq \sum_{g \in \mathcal{G}} \mathbb{E}\left[\left|P_G(\boldsymbol{\mu}_k^N, g_k^N, \boldsymbol{\nu}_k^N)(g) - P_G(\boldsymbol{\mu}_k^N, g_k^N, \nu^{\mathrm{MF}}(\boldsymbol{\mu}_k^N, g_k^N, \pi_k))(g)\right| \times \left|\tilde{r}^{\mathrm{MF}}(P^{\mathrm{MF}}(\boldsymbol{\mu}_k^N, g_k^N, \pi_k), g, \pi_{k+1:t})\right|\right] \\
&\overset{(a)}{\leq} M_R \times \mathbb{E}\left|P_G(\boldsymbol{\mu}_k^N, g_k^N, \boldsymbol{\nu}_k^N) - P_G(\boldsymbol{\mu}_k^N, g_k^N, \nu^{\mathrm{MF}}(\boldsymbol{\mu}_k^N, g_k^N, \pi_k))\right|_1 \\
&\overset{(b)}{\leq} M_R \times L_G \times \mathbb{E}\left|\boldsymbol{\nu}_k^N - \nu^{\mathrm{MF}}(\boldsymbol{\mu}_k^N, g_k^N, \pi_k)\right|_1 \overset{(c)}{\leq} M_R L_G \sqrt{\frac{|\mathcal{U}|}{N}}
\end{aligned}
$$

Inequality $(a)$ is a consequence of Assumption 1$(a)$ and the definition of $\tilde{r}^{\mathrm{MF}}$ given in (31). Inequality $(b)$ follows from Assumption 1$(d)$. Finally, $(c)$ is a consequence of Lemma 11. Combining, we obtain,

$$
J_t \leq C_P \left[\left(\frac{M_R S_G}{S_P - 1} + S_R\right)\left(\frac{S_P^t - 1}{S_P - 1}\right) - \left(\frac{M_R S_G}{S_P - 1}\right)t\right] \times \frac{1}{\sqrt{N}}\left[\sqrt{|\mathcal{X}|} + \sqrt{|\mathcal{U}|}\right] + M_R L_G \sqrt{\frac{|\mathcal{U}|}{N}}t
$$

Substituting in (35), we obtain the following result.

$$
\begin{aligned}
|V_N(\boldsymbol{x}_0, g_0, \boldsymbol{\pi}) - V_\infty(\boldsymbol{\mu}_0, g_0, \boldsymbol{\pi})| &\leq \left(\frac{M_R + L_R\sqrt{|\mathcal{U}|}}{1 - \gamma}\right)\frac{1}{\sqrt{N}} + M_R L_G \sqrt{\frac{|\mathcal{U}|}{N}}\frac{\gamma}{(1 - \gamma)^2} \\
&+ \left(\frac{C_P}{S_P - 1}\right)\left[\left(\frac{M_R S_G}{S_P - 1} + S_R\right)\left\{\frac{1}{1 - \gamma S_P} - \frac{1}{1 - \gamma}\right\} - \frac{\gamma M_R S_G}{(1 - \gamma)^2}\right] \times \frac{1}{\sqrt{N}}\left[\sqrt{|\mathcal{X}|} + \sqrt{|\mathcal{U}|}\right]
\end{aligned}
$$

We conclude by noting that $|\sup_{\boldsymbol{\pi}} V_N(\boldsymbol{x}_0, g_0, \boldsymbol{\pi}) - \sup_{\boldsymbol{\pi}} V_\infty(\boldsymbol{\mu}_0, g_0, \boldsymbol{\pi})| \leq \sup_{\boldsymbol{\pi}} |V_N(\boldsymbol{x}_0, g_0, \boldsymbol{\pi}) - V_\infty(\boldsymbol{\mu}_0, g_0, \boldsymbol{\pi})|$ where the suprema are taken over the set of all admissible policy sequences $\Pi^\infty$.

## B    Proof of Lemma 2

Note the inequalities stated below.

$$
\begin{aligned}
|\nu^{\mathrm{MF}}(\boldsymbol{\mu}, g, \pi) - \nu^{\mathrm{MF}}(\bar{\boldsymbol{\mu}}, g, \pi)|_1 &\stackrel{(a)}{=} \left| \sum_{x \in \mathcal{X}} \pi(x, \boldsymbol{\mu}, g) \boldsymbol{\mu}(x) - \sum_{x \in \mathcal{X}} \pi(x, \bar{\boldsymbol{\mu}}, g) \bar{\boldsymbol{\mu}}(x) \right|_1 \\
&= \sum_{u \in \mathcal{U}} \left| \sum_{x \in \mathcal{X}} \pi(x, \boldsymbol{\mu}, g)(u) \boldsymbol{\mu}(x) - \sum_{x \in \mathcal{X}} \pi(x, \bar{\boldsymbol{\mu}}, g)(u) \bar{\boldsymbol{\mu}}(x) \right| \\
&\leq \sum_{x \in \mathcal{X}} \sum_{u \in \mathcal{U}} |\pi(x, \boldsymbol{\mu}, g)(u) \boldsymbol{\mu}(x) - \pi(x, \bar{\boldsymbol{\mu}}, g)(u) \bar{\boldsymbol{\mu}}(x)| \\
&\leq \sum_{x \in \mathcal{X}} |\boldsymbol{\mu}(x) - \bar{\boldsymbol{\mu}}(x)| \underbrace{\sum_{u \in \mathcal{U}} \pi(x, \boldsymbol{\mu}, g)(u)}_{=1} + \sum_{x \in \mathcal{X}} \bar{\boldsymbol{\mu}}(x) \sum_{u \in \mathcal{U}} |\pi(x, \boldsymbol{\mu}, g)(u) - \pi(x, \bar{\boldsymbol{\mu}}, g)(u)| \\
&\stackrel{(b)}{\leq} |\boldsymbol{\mu} - \bar{\boldsymbol{\mu}}|_1 + \underbrace{\sum_{x \in \mathcal{X}} \bar{\boldsymbol{\mu}}(x)}_{=1} L_Q |\boldsymbol{\mu} - \bar{\boldsymbol{\mu}}|_1 \\
&\stackrel{(c)}{=} (1 + L_Q) |\boldsymbol{\mu} - \bar{\boldsymbol{\mu}}|_1
\end{aligned}
$$

Inequality (a) follows from the definition of $\nu^{\mathrm{MF}}(\cdot, \cdot)$ as given in (7). On the other hand, (b) is a consequence of Assumption 2 and the fact that $\pi(x, \boldsymbol{\mu}, g)$ is a valid probability distribution. Finally, $(c)$ uses the fact that $\bar{\boldsymbol{\mu}}$ is a distribution. This concludes the lemma.

## C    Proof of Lemma 3

Observe that,

$$
\begin{aligned}
&|P^{\mathrm{MF}}(\boldsymbol{\mu}, g, \pi) - P^{\mathrm{MF}}(\bar{\boldsymbol{\mu}}, g, \pi)|_1 \\
&\stackrel{(a)}{=} \left| \sum_{x \in \mathcal{X}} \sum_{u \in \mathcal{U}} P(x, u, \boldsymbol{\mu}, g, \nu^{\mathrm{MF}}(\boldsymbol{\mu}, g, \pi)) \pi(x, \boldsymbol{\mu}, g)(u) \boldsymbol{\mu}(x) - P(x, u, \bar{\boldsymbol{\mu}}, g, \nu^{\mathrm{MF}}(\bar{\boldsymbol{\mu}}, g, \pi)) \pi(x, \bar{\boldsymbol{\mu}}, g)(u) \bar{\boldsymbol{\mu}}(x) \right|_1 \\
&\leq J_1 + J_2
\end{aligned}
$$

Equality (a) follows from the definition of $P^{\mathrm{MF}}(\cdot, \cdot, \cdot)$ as depicted in (8). The term $J_1$ satisfies the following bound.

$$
\begin{aligned}
J_1 &\triangleq \sum_{x \in \mathcal{X}} \sum_{u \in \mathcal{U}} \left| P(x, u, \boldsymbol{\mu}, g, \nu^{\mathrm{MF}}(\boldsymbol{\mu}, g, \pi)) - P(x, u, \bar{\boldsymbol{\mu}}, g, \nu^{\mathrm{MF}}(\bar{\boldsymbol{\mu}}, g, \pi)) \right|_1 \times \pi(x, \boldsymbol{\mu}, g)(u) \boldsymbol{\mu}(x) \\
&\stackrel{(a)}{\leq} L_P \left[ |\boldsymbol{\mu} - \bar{\boldsymbol{\mu}}|_1 + |\nu^{\mathrm{MF}}(\boldsymbol{\mu}, g, \pi) - \nu^{\mathrm{MF}}(\bar{\boldsymbol{\mu}}, g, \pi)|_1 \right] \times \underbrace{\sum_{x \in \mathcal{X}} \boldsymbol{\mu}(x) \sum_{u \in \mathcal{U}} \pi(x, \boldsymbol{\mu}, g)(u)}_{=1} \\
&\stackrel{(b)}{\leq} L_P (2 + L_Q) |\boldsymbol{\mu} - \bar{\boldsymbol{\mu}}|_1
\end{aligned}
$$

Inequality $(a)$ is a consequence of Assumption 1(c) whereas $(b)$ follows from Lemma 2, and the fact that $\pi(x, \boldsymbol{\mu}, g)$, $\boldsymbol{\mu}$ are probability distributions. The second term, $J_2$ obeys the following bound.

$$
\begin{aligned}
J_2 &\triangleq \sum_{x \in \mathcal{X}} \sum_{u \in \mathcal{U}} \underbrace{|P(x, u, \bar{\boldsymbol{\mu}}, g, \nu^{\mathrm{MF}}(\bar{\boldsymbol{\mu}}, g, \pi))|_1}_{=1} \times |\pi(x, \boldsymbol{\mu}, g)(u)\boldsymbol{\mu}(x) - \pi(x, \bar{\boldsymbol{\mu}}, g)(u)\bar{\boldsymbol{\mu}}(x)| \\
&\leq \sum_{x \in \mathcal{X}} |\boldsymbol{\mu}(x) - \bar{\boldsymbol{\mu}}(x)| \underbrace{\sum_{u \in \mathcal{U}} \pi(x, \boldsymbol{\mu}, g)(u)}_{=1} + \sum_{x \in \mathcal{X}} \bar{\boldsymbol{\mu}}(x) \sum_{u \in \mathcal{U}} |\pi(x, \boldsymbol{\mu}, g)(u) - \pi(x, \bar{\boldsymbol{\mu}}, g)(u)| \\
&\overset{(a)}{\leq} |\boldsymbol{\mu} - \bar{\boldsymbol{\mu}}|_1 + \underbrace{\sum_{x \in \mathcal{X}} \bar{\boldsymbol{\mu}}(x)}_{=1} L_Q |\boldsymbol{\mu} - \bar{\boldsymbol{\mu}}|_1 \overset{(b)}{=} (1 + L_Q)|\boldsymbol{\mu} - \bar{\boldsymbol{\mu}}|_1
\end{aligned}
$$

Inequality $(a)$ results from Assumption 2 and the fact that $\pi(x, \boldsymbol{\mu}, g)$ is a probability distribution whereas $(b)$ utilizes the fact that $\bar{\boldsymbol{\mu}}$ is a distribution. This concludes the result.

## D  Proof of Lemma 4

Note the following relations.

$$
\begin{aligned}
|P_G^{\mathrm{MF}}(\boldsymbol{\mu}, g, \pi) - P_G^{\mathrm{MF}}(\bar{\boldsymbol{\mu}}, g, \pi)|_1 &\overset{(a)}{=} |P_G(\boldsymbol{\mu}, g, \nu^{\mathrm{MF}}(\boldsymbol{\mu}, g, \pi)) - P_G^{\mathrm{MF}}(\bar{\boldsymbol{\mu}}, g, \nu^{\mathrm{MF}}(\bar{\boldsymbol{\mu}}, g, \pi))|_1 \\
&\overset{(b)}{\leq} L_G \left\{ |\boldsymbol{\mu} - \bar{\boldsymbol{\mu}}|_1 + |\nu^{\mathrm{MF}}(\boldsymbol{\mu}, g, \pi) - \nu^{\mathrm{MF}}(\bar{\boldsymbol{\mu}}, g, \pi)|_1 \right\} \overset{(c)}{\leq} L_G(2 + L_Q)|\boldsymbol{\mu}_l - \bar{\boldsymbol{\mu}}_l|_1
\end{aligned}
$$

Equality $(a)$ follows from the definition (9) while $(b)$ follows from Assumption 1($d$). Finally, $(c)$ is a result of Lemma 2.

## E  Proof of Lemma 5

Observe that,

$$
\begin{aligned}
&|r^{\mathrm{MF}}(\boldsymbol{\mu}, g, \pi) - r^{\mathrm{MF}}(\bar{\boldsymbol{\mu}}, g, \pi)|_1 \\
&\overset{(a)}{=} \left| \sum_{x \in \mathcal{X}} \sum_{u \in \mathcal{U}} r(x, u, \boldsymbol{\mu}, g, \nu^{\mathrm{MF}}(\boldsymbol{\mu}, g, \pi))\pi(x, \boldsymbol{\mu}, g)(u)\boldsymbol{\mu}(x) - r(x, u, \bar{\boldsymbol{\mu}}, g, \nu^{\mathrm{MF}}(\bar{\boldsymbol{\mu}}, g, \pi))\pi(x, \bar{\boldsymbol{\mu}}, g)(u)\bar{\boldsymbol{\mu}}(x) \right|_1 \\
&\leq J_1 + J_2
\end{aligned}
$$

Equality (a) follows from the definition of $r^{\mathrm{MF}}(\cdot, \cdot, \cdot)$ as depicted in (10). The term $J_1$ satisfies the following bound.

$$
\begin{aligned}
J_1 &\triangleq \sum_{x \in \mathcal{X}} \sum_{u \in \mathcal{U}} \left| r(x, u, \boldsymbol{\mu}, g, \nu^{\mathrm{MF}}(\boldsymbol{\mu}, g, \pi)) - r(x, u, \bar{\boldsymbol{\mu}}, g, \nu^{\mathrm{MF}}(\bar{\boldsymbol{\mu}}, g, \pi)) \right| \times \pi(x, \boldsymbol{\mu}, g)(u)\boldsymbol{\mu}(x) \\
&\overset{(a)}{\leq} L_R \left[ |\boldsymbol{\mu} - \bar{\boldsymbol{\mu}}|_1 + |\nu^{\mathrm{MF}}(\boldsymbol{\mu}, g, \pi) - \nu^{\mathrm{MF}}(\bar{\boldsymbol{\mu}}, g, \pi)|_1 \right] \times \underbrace{\sum_{x \in \mathcal{X}} \boldsymbol{\mu}(x) \sum_{u \in \mathcal{U}} \pi(x, \boldsymbol{\mu}, g)(u)}_{=1} \overset{(b)}{\leq} L_R(2 + L_Q)|\boldsymbol{\mu} - \bar{\boldsymbol{\mu}}|_1
\end{aligned}
$$

Inequality $(a)$ is a consequence of Assumption 1$(b)$ whereas $(b)$ follows from Lemma 2, and the fact that $\pi(x, \boldsymbol{\mu}, g)$, $\boldsymbol{\mu}$ are probability distributions. The second term, $J_2$ obeys the following bound.

$$
\begin{aligned}
J_2 &\triangleq \sum_{x \in \mathcal{X}} \sum_{u \in \mathcal{U}} |r(x, u, \bar{\boldsymbol{\mu}}, g, \nu^{\mathrm{MF}}(\bar{\boldsymbol{\mu}}, g, \pi))| \times |\pi(x, \boldsymbol{\mu}, g)(u)\boldsymbol{\mu}(x) - \pi(x, \bar{\boldsymbol{\mu}}, g)(u)\bar{\boldsymbol{\mu}}(x)| \\
&\stackrel{(a)}{\leq} M_R \sum_{x \in \mathcal{X}} |\boldsymbol{\mu}(x) - \bar{\boldsymbol{\mu}}(x)| \underbrace{\sum_{u \in \mathcal{U}} \pi(x, \boldsymbol{\mu}, g)(u)}_{=1} + M_R \sum_{x \in \mathcal{X}} \bar{\boldsymbol{\mu}}(x) \sum_{u \in \mathcal{U}} |\pi(x, \boldsymbol{\mu}, g)(u) - \pi(x, \bar{\boldsymbol{\mu}}, g)(u)| \\
&\stackrel{(b)}{\leq} M_R |\boldsymbol{\mu} - \bar{\boldsymbol{\mu}}|_1 + M_R \underbrace{\sum_{x \in \mathcal{X}} \bar{\boldsymbol{\mu}}(x)}_{=1} L_Q |\boldsymbol{\mu} - \bar{\boldsymbol{\mu}}|_1 \stackrel{(c)}{=} M_R(1 + L_Q)|\boldsymbol{\mu} - \bar{\boldsymbol{\mu}}|_1
\end{aligned}
$$

Inequality $(a)$ results from Assumption 1(a) where $(b)$ follows from Assumption 2 and the fact that $\pi(x, \boldsymbol{\mu}, g)$ is a probability distribution. Finally, $(c)$ utilizes the fact that $\bar{\boldsymbol{\mu}}$ is a valid distribution. This concludes the result.

## F  Proof of Lemma 6

Fix $l \in \{0, 1, \cdots\}$. We shall prove the lemma via induction on $r$. Note that, for $r = 0$, we have,

$$
|\tilde{P}^{\mathrm{MF}}(\boldsymbol{\mu}_l, g_{l:l}, \pi_{l:l}) - \tilde{P}^{\mathrm{MF}}(\bar{\boldsymbol{\mu}}_l, g_{l:l}, \pi_{l:l})|_1 = |P^{\mathrm{MF}}(\boldsymbol{\mu}_l, g_l, \pi_l) - P^{\mathrm{MF}}(\bar{\boldsymbol{\mu}}_l, g_l, \pi_l)|_1 \stackrel{(a)}{\leq} S_P |\boldsymbol{\mu}_l - \bar{\boldsymbol{\mu}}_l|_1
$$

Inequality $(a)$ follows from Lemma 3. Assume that the lemma holds for some $r \in \{0, 1, \cdots\}$. We shall demonstrate below that the relation holds for $r + 1$ as well. Note that,

$$
\begin{aligned}
&|\tilde{P}^{\mathrm{MF}}(\boldsymbol{\mu}_l, g_{l:l+r+1}, \pi_{l:l+r+1}) - \tilde{P}^{\mathrm{MF}}(\bar{\boldsymbol{\mu}}_l, g_{l:l+r+1}, \pi_{l:l+r+1})|_1 \\
&= |P^{\mathrm{MF}}(\tilde{P}^{\mathrm{MF}}(\boldsymbol{\mu}_l, g_{l:l+r}, \pi_{l:l+r}), g_{l+r+1}, \pi_{l+r+1}) - P^{\mathrm{MF}}(\tilde{P}^{\mathrm{MF}}(\bar{\boldsymbol{\mu}}_l, g_{l:l+r}, \pi_{l:l+r}), g_{l+r+1}, \pi_{l+r+1})|_1 \\
&\stackrel{(a)}{\leq} S_P |\tilde{P}^{\mathrm{MF}}(\boldsymbol{\mu}_l, g_{l:l+r}, \pi_{l:l+r}) - \tilde{P}^{\mathrm{MF}}(\bar{\boldsymbol{\mu}}_l, g_{l:l+r}, \pi_{l:l+r})|_1 \stackrel{(b)}{\leq} S_P^{r+2} |\boldsymbol{\mu}_l - \bar{\boldsymbol{\mu}}_l|_1
\end{aligned}
$$

Inequality $(a)$ follows from Lemma 3 and $(b)$ is a consequence of the induction hypothesis.

## G  Proof of Lemma 7

Fix $l \in \{0, 1, \cdots\}$. We shall prove the lemma via induction on $r$. Note that, for $r = 0$, we have,

$$
|\tilde{P}_G^{\mathrm{MF}}(\boldsymbol{\mu}_l, g_{l:l}, \pi_{l:l}) - \tilde{P}_G^{\mathrm{MF}}(\bar{\boldsymbol{\mu}}_l, g_{l:l}, \pi_{l:l})|_1 = |P_G^{\mathrm{MF}}(\boldsymbol{\mu}_l, g_l, \pi_l) - P_G^{\mathrm{MF}}(\bar{\boldsymbol{\mu}}_l, g_l, \pi_l)|_1 \stackrel{(a)}{\leq} S_G |\boldsymbol{\mu}_l - \bar{\boldsymbol{\mu}}_l|_1 \tag{36}
$$

Equality $(a)$ follows from Lemma 4. Assume that the lemma holds for some $r \in \{0, 1, \cdots\}$. We shall now show that the lemma holds for $r + 1$ as well. Note that,

$$
\begin{aligned}
&\sum_{l+1:l+r+1} \left|\tilde{P}_G^{\mathrm{MF}}(\boldsymbol{\mu}_l, g_{l:l+r+1}, \pi_{l:l+r+1}) - \tilde{P}_G^{\mathrm{MF}}(\bar{\boldsymbol{\mu}}_l, g_{l:l+r+1}, \pi_{l:l+r+1})\right|_1 \\
&\stackrel{(a)}{=} \sum_{l+1:l+r+1} \left|\tilde{P}_G^{\mathrm{MF}}(\boldsymbol{\mu}_l, g_{l:l+r}, \pi_{l:l+r})(g_{l+r+1}) P_G^{\mathrm{MF}}(\tilde{P}^{\mathrm{MF}}(\boldsymbol{\mu}_l, g_{l:l+r}, \pi_{l:l+r}), g_{l+r+1}, \pi_{l+r+1})\right. \\
&\qquad \left. - \tilde{P}_G^{\mathrm{MF}}(\bar{\boldsymbol{\mu}}_l, g_{l:l+r}, \pi_{l:l+r})(g_{l+r+1}) P_G^{\mathrm{MF}}(\tilde{P}^{\mathrm{MF}}(\bar{\boldsymbol{\mu}}_l, g_{l:l+r}, \pi_{l:l+r}), g_{l+r+1}, \pi_{l+r+1})\right|_1 \leq J_1 + J_2
\end{aligned}
$$

Equality $(a)$ follows from (30). The first term $J_1$ can be upper bounded as follows.

$$
\begin{aligned}
J_1 &\triangleq \sum_{l+1:l+r+1} \tilde{P}_G^{\mathrm{MF}}(\boldsymbol{\mu}_l, g_{l:l+r}, \pi_{l:l+r})(g_{l+r+1}) \\
&\quad \times \left| P_G^{\mathrm{MF}}(\tilde{P}^{\mathrm{MF}}(\boldsymbol{\mu}_l, g_{l:l+r}, \pi_{l:l+r}), g_{l+r+1}, \pi_{l+r+1}) - P_G^{\mathrm{MF}}(\tilde{P}^{\mathrm{MF}}(\bar{\boldsymbol{\mu}}_l, g_{l:l+r}, \pi_{l:l+r}), g_{l+r+1}, \pi_{l+r+1}) \right|_1 \\
&\overset{(a)}{\leq} S_G \sum_{l+1:l+r+1} \left| \tilde{P}^{\mathrm{MF}}(\boldsymbol{\mu}_l, g_{l:l+r}, \pi_{l:l+r}) - \tilde{P}^{\mathrm{MF}}(\boldsymbol{\mu}_l, g_{l:l+r}, \pi_{l:l+r}) \right|_1 \times \tilde{P}_G^{\mathrm{MF}}(\boldsymbol{\mu}_l, g_{l:l+r}, \pi_{l:l+r})(g_{l+r+1}) \\
&\overset{(b)}{\leq} S_G S_P^{r+1} |\boldsymbol{\mu}_l - \bar{\boldsymbol{\mu}}_l|_1 \sum_{l+1:l+r+1} \tilde{P}_G^{\mathrm{MF}}(\boldsymbol{\mu}_l, g_{l:l+r}, \pi_{l:l+r})(g_{l+r+1}) \overset{(c)}{=} S_G S_P^{r+1} |\boldsymbol{\mu}_l - \bar{\boldsymbol{\mu}}_l|_1
\end{aligned}
$$

Inequality $(a)$ follows from Lemma 4 while $(b)$ results from Lemma 6. Finally, $(c)$ can be shown following the definition (30). The second term $J_2$ can be bounded as follows.

$$
\begin{aligned}
J_2 &\triangleq \sum_{l+1:l+r+1} \underbrace{|P_G^{\mathrm{MF}}(\tilde{P}^{\mathrm{MF}}(\bar{\boldsymbol{\mu}}_l, g_{l:l+r}, \pi_{l:l+r}), g_{l+r+1}, \pi_{l+r+1})|_1}_{=1} \\
&\quad \times |\tilde{P}_G^{\mathrm{MF}}(\boldsymbol{\mu}_l, g_{l:l+r}, \pi_{l:l+r})(g_{l+r+1}) - \tilde{P}_G^{\mathrm{MF}}(\bar{\boldsymbol{\mu}}_l, g_{l:l+r}, \pi_{l:l+r})(g_{l+r+1})| \\
&= \sum_{l+1:l+r+1} |\tilde{P}_G^{\mathrm{MF}}(\boldsymbol{\mu}_l, g_{l:l+r}, \pi_{l:l+r}) - \tilde{P}_G^{\mathrm{MF}}(\bar{\boldsymbol{\mu}}_l, g_{l:l+r}, \pi_{l:l+r})|_1 \overset{(a)}{\leq} S_G(1 + S_P + \cdots + S_P^r)|\boldsymbol{\mu}_l - \bar{\boldsymbol{\mu}}_l|_1
\end{aligned}
$$

Inequality $(a)$ follows from induction hypothesis. This concludes the lemma.

## H    Proof of Lemma 8

Note that the result readily follows for $r = 0$ from Lemma 5. Therefore, we assume $r \geq 1$.

$$
\begin{aligned}
&|\tilde{r}^{\mathrm{MF}}(\boldsymbol{\mu}_l, g_l, \pi_{l:l+r}) - \tilde{r}^{\mathrm{MF}}(\bar{\boldsymbol{\mu}}_l, g_l, \pi_{l:l+r})| \\
&\overset{(a)}{\leq} \sum_{l+1:l+r} |r^{\mathrm{MF}}(\tilde{P}^{\mathrm{MF}}(\boldsymbol{\mu}_l, g_{l:l+r-1}, \pi_{l:l+r-1}), g_{l+r}, \pi_{l+r}) \tilde{P}_G^{\mathrm{MF}}(\boldsymbol{\mu}_l, g_{l:l+r-1}, \pi_{l:l+r-1})(g_{l+r}) \\
&\qquad - r^{\mathrm{MF}}(\tilde{P}^{\mathrm{MF}}(\bar{\boldsymbol{\mu}}_l, g_{l:l+r-1}, \pi_{l:l+r-1}), g_{l+r}, \pi_{l+r}) \tilde{P}_G^{\mathrm{MF}}(\bar{\boldsymbol{\mu}}_l, g_{l:l+r-1}, \pi_{l:l+r-1})(g_{l+r})| \leq J_1 + J_2
\end{aligned}
$$

Inequality $(a)$ follows from the definition (31). The first term can be bounded as follows.

$$
\begin{aligned}
J_1 &\triangleq \sum_{l+1:l+r} |r^{\mathrm{MF}}(\tilde{P}^{\mathrm{MF}}(\boldsymbol{\mu}_l, g_{l:l+r-1}, \pi_{l:l+r-1}), g_{l+r}, \pi_{l+r})| \\
&\quad \times |\tilde{P}_G^{\mathrm{MF}}(\boldsymbol{\mu}_l, g_{l:l+r-1}, \pi_{l:l+r-1})(g_{l+r}) - \tilde{P}_G^{\mathrm{MF}}(\bar{\boldsymbol{\mu}}_l, g_{l:l+r-1}, \pi_{l:l+r-1})(g_{l+r})| \\
&\overset{(a)}{\leq} M_R \sum_{l+1:l+r-1} |\tilde{P}_G^{\mathrm{MF}}(\boldsymbol{\mu}_l, g_{l:l+r-1}, \pi_{l:l+r-1}) - \tilde{P}_G^{\mathrm{MF}}(\bar{\boldsymbol{\mu}}_l, g_{l:l+r-1}, \pi_{l:l+r-1})|_1 \\
&\overset{(b)}{\leq} M_R S_G(1 + S_P + \cdots + S_P^{r-1})|\boldsymbol{\mu}_l - \bar{\boldsymbol{\mu}}_l|_1 = \left(\frac{M_R S_G}{S_P - 1}\right)(S_P^r - 1)|\boldsymbol{\mu}_l - \bar{\boldsymbol{\mu}}_l|_1
\end{aligned}
$$

The bound $(a)$ can be proven using Assumption 1$(a)$ and the definition of $r^{\text{MF}}$ given in (10). The bound $(b)$ follows from Lemma 7. The term $J_2$ can be bounded as follows.

$$
\begin{aligned}
J_2 &\triangleq \sum_{l+1:l+r} \tilde{P}_G^{\text{MF}}(\bar{\boldsymbol{\mu}}_l, g_{l:l+r-1}, \pi_{l:l+r-1})(g_{l+r}) \\
&\quad \times |r^{\text{MF}}(\tilde{P}^{\text{MF}}(\boldsymbol{\mu}_l, g_{l:l+r-1}, \pi_{l:l+r-1}), g_{l+r}, \pi_{l+r}) - r^{\text{MF}}(\tilde{P}^{\text{MF}}(\bar{\boldsymbol{\mu}}_l, g_{l:l+r-1}, \pi_{l:l+r-1}), g_{l+r}, \pi_{l+r})| \\
&\stackrel{(a)}{\leq} \sum_{l+1:l+r} \tilde{P}_G^{\text{MF}}(\bar{\boldsymbol{\mu}}_l, g_{l:l+r-1}, \pi_{l:l+r-1})(g_{l+r}) \times S_R|\tilde{P}^{\text{MF}}(\boldsymbol{\mu}_l, g_{l:l+r-1}, \pi_{l:l+r-1}) - \tilde{P}^{\text{MF}}(\bar{\boldsymbol{\mu}}_l, g_{l:l+r-1}, \pi_{l:l+r-1})| \\
&\stackrel{(b)}{\leq} S_R S_P^r |\boldsymbol{\mu}_l - \bar{\boldsymbol{\mu}}_l|_1 \times \sum_{l+1:l+r} \tilde{P}_G^{\text{MF}}(\bar{\boldsymbol{\mu}}_l, g_{l:l+r-1}, \pi_{l:l+r-1})(g_{l+r}) \stackrel{(c)}{=} S_R S_P^r |\boldsymbol{\mu}_l - \bar{\boldsymbol{\mu}}_l|_1
\end{aligned}
$$

Inequality $(a)$ follows from Lemma 5 while $(b)$ results from Lemma 6. Finally, $(c)$ can be proven from the definition (30). This concludes the lemma.

## I  Proof of Lemma 9

$$
\begin{aligned}
&\tilde{r}^{\text{MF}}(\boldsymbol{\mu}_l, g_l, \pi_{l:l+r}) \\
&= \sum_{l+1:l+r} r^{\text{MF}}(\tilde{P}^{\text{MF}}(\boldsymbol{\mu}_l, g_{l:l+r-1}, \pi_{l:l+r-1}), g_{l+r}, \pi_{l+r})\tilde{P}_G^{\text{MF}}(\boldsymbol{\mu}_l, g_{l:l+r-1}, \pi_{l:l+r-1})(g_{l+r}) \\
&\stackrel{(a)}{=} \sum_{l+1:l+r} r^{\text{MF}}(\tilde{P}^{\text{MF}}(P^{\text{MF}}(\boldsymbol{\mu}_l, g_l, \pi_l), g_{l+1:l+r-1}, \pi_{l+1:l+r-1}), g_{l+r}, \pi_{l+r}) \\
&\qquad \times \tilde{P}_G^{\text{MF}}(P^{\text{MF}}(\boldsymbol{\mu}_l, g_l, \pi_l), g_{l+1:l+r-1}, \pi_{l+1:l+r-1})(g_{l+r})P_G^{\text{MF}}(\boldsymbol{\mu}_l, g_l, \pi_l)(g_{l+1}) \\
&= \sum_{g_{l+1}\in\mathcal{G}} \tilde{r}^{\text{MF}}(P^{\text{MF}}(\boldsymbol{\mu}_l, g_l, \pi_l), g_{l+1}, \pi_{l+1:l+r})P_G^{\text{MF}}(\boldsymbol{\mu}_l, g_l, \pi_l)(g_{l+1})
\end{aligned}
$$

Equality $(a)$ follows from the definitions (29) and (30).

## J  Proof of Lemma 10

Let $Y_{mn} \triangleq X_{mn} - \mathbb{E}[X_{mn}]$, $\forall m \in \{1, \cdots, M\}$, $\forall n \in \{1, \cdots, N\}$. Note that, as $X_{mn} \in [0, 1]$, we have, $E[Y_{mn}^2] = E[X_{mn}^2] - [E[X_{mn}]]^2 \leq E[X_{mn}]$. Using independence of $\{Y_{mn}\}_{n\in\{1,\cdots,N\}}$, for any given $m \in \{1, \cdots, M\}$, we get,

$$
\mathbb{E}\left[\sum_{n=1}^{N} Y_{m,n}\right]^2 = \mathbb{E}\left[\sum_{n_1=1}^{N}\sum_{n_2=1}^{N} Y_{m,n_1}Y_{m,n_2}\right] = \sum_{n=1}^{N}\mathbb{E}\left[Y_{m,n}^2\right] + 2\sum_{n_1=1}^{N}\sum_{n_2>n_1}^{N}\mathbb{E}[Y_{m,n_1}]\mathbb{E}[Y_{m,n_2}] = \sum_{n=1}^{N}\mathbb{E}\left[Y_{m,n}^2\right]
$$

Using the above relation, we finally obtain the following.

$$
\begin{aligned}
\sum_{m=1}^{M}\mathbb{E}\left|\sum_{n=1}^{N} Y_{m,n}\right| &\stackrel{(a)}{\leq} \sqrt{M}\left\{\sum_{m=1}^{M}\mathbb{E}\left[\sum_{n=1}^{N} Y_{m,n}\right]^2\right\}^{\frac{1}{2}} = \sqrt{M}\left\{\sum_{m=1}^{M}\sum_{n=1}^{N}\mathbb{E}\left[Y_{m,n}^2\right]\right\}^{\frac{1}{2}} \\
&= \sqrt{M}\left\{\sum_{n=1}^{N}\sum_{m=1}^{M}\mathbb{E}\left[X_{m,n}\right]\right\}^{\frac{1}{2}} \leq \sqrt{MN}
\end{aligned}
$$

## K    Proof of Lemma 11

Notice the following relations.

$$\mathbb{E}\left|\boldsymbol{\nu}_t^N - \nu^{\mathrm{MF}}(\boldsymbol{\mu}_t^N, g_t^N, \pi_t)\right|_1$$

$$\overset{(a)}{=} \mathbb{E}\left|\boldsymbol{\nu}_t^N - \sum_{x\in\mathcal{X}} \pi_t(x, \boldsymbol{\mu}_t^N, g_t^N)\boldsymbol{\mu}_t^N(x)\right|_1$$

$$= \mathbb{E}\left[\mathbb{E}\left[\sum_{u\in\mathcal{U}}\left|\boldsymbol{\nu}_t^N(u) - \sum_{x\in\mathcal{X}}\pi_t(x,\boldsymbol{\mu}_t^N,g_t^N)(u)\boldsymbol{\mu}_t^N(x)\right|\,\middle|\,\boldsymbol{x}_t^N, g_t^N\right]\right]$$

$$\overset{(b)}{=} \mathbb{E}\left[\sum_{u\in\mathcal{U}}\mathbb{E}\left[\frac{1}{N}\left|\sum_{i=1}^N\delta(u_t^i = u) - \frac{1}{N}\sum_{x\in\mathcal{X}}\pi_t(x,\boldsymbol{\mu}_t^N,g_t^N)(u)\sum_{i=1}^N\delta(x_t^i = x)\right|\,\middle|\,\boldsymbol{x}_t^N, g_t^N\right]\right]$$

$$= \mathbb{E}\left[\sum_{u\in\mathcal{U}}\mathbb{E}\left[\left|\frac{1}{N}\sum_{i=1}^N\delta(u_t^i = u) - \frac{1}{N}\sum_{i=1}^N\pi_t(x_t^i,\boldsymbol{\mu}_t^N,g_t^N)(u)\right|\,\middle|\,\boldsymbol{x}_t^N, g_t^N\right]\right] \overset{(c)}{\leq} \frac{1}{\sqrt{N}}\sqrt{|\mathcal{U}|}$$

Equality $(a)$ follows from the definition of $\nu^{\mathrm{MF}}(\cdot,\cdot,\cdot)$ given in (7) while $(b)$ is a consequence of the definitions of $\boldsymbol{\mu}_t^N, \boldsymbol{\nu}_t^N$. Finally, $(c)$ uses Lemma 10. Specifically, it utilises the facts that, $\{u_t^i\}_{i\in\{1,\cdots,N\}}$ are conditionally independent given $\boldsymbol{x}_t^N$, $g_t^N$ and the following holds

$$\mathbb{E}\left[\delta(u_t^i = u)\,\middle|\,\boldsymbol{x}_t^N, g_t^N\right] = \pi_t(x_t^i, \boldsymbol{\mu}_t^N, g_t^N)(u),$$

$$\sum_{u\in\mathcal{U}}\mathbb{E}\left[\delta(u_t^i = u)\,\middle|\,\boldsymbol{x}_t^N, g_t^N\right] = 1$$

$\forall i \in \{1,\cdots,N\}, \forall u \in \mathcal{U}$. This concludes the lemma.

## L    Proof of Lemma 12

Notice the following decomposition.

$$\mathbb{E}\left|\boldsymbol{\mu}_{t+1}^N - P^{\mathrm{MF}}(\boldsymbol{\mu}_t^N, g_t^N, \pi_t)\right|_1$$

$$\overset{(a)}{=} \mathbb{E}\left|\boldsymbol{\mu}_{t+1}^N - \sum_{x'\in\mathcal{X}}\sum_{u\in\mathcal{U}}P(x',u,\boldsymbol{\mu}_t^N,g_t^N,\nu^{\mathrm{MF}}(\boldsymbol{\mu}_t^N,g_t^N,\pi_t))\pi_t(x',\boldsymbol{\mu}_t^N,g_t^N)(u)\boldsymbol{\mu}_t^N(x)\right|_1$$

$$\overset{(b)}{=} \sum_{x\in\mathcal{X}}\mathbb{E}\left|\frac{1}{N}\sum_{i=1}^N\delta(x_{t+1}^i = x) - \sum_{x'\in\mathcal{X}}\sum_{u\in\mathcal{U}}P(x',u,\boldsymbol{\mu}_t^N,g_t^N,\nu^{\mathrm{MF}}(\boldsymbol{\mu}_t^N,g_t^N,\pi_t))(x)\pi_t(x',\boldsymbol{\mu}_t^N,g')(u)\frac{1}{N}\sum_{i=1}^N\delta(x_t^i = x')\right|$$

$$= \sum_{x\in\mathcal{X}}\mathbb{E}\left|\frac{1}{N}\sum_{i=1}^N\delta(x_{t+1}^i = x) - \frac{1}{N}\sum_{i=1}^N\sum_{u\in\mathcal{U}}P(x_t^i,u,\boldsymbol{\mu}_t^N,g_t^N,\nu^{\mathrm{MF}}(\boldsymbol{\mu}_t^N,g_t^N,\pi_t))(x)\pi_t(x_t^i,\boldsymbol{\mu}_t^N,g_t^N)(u)\right|$$

$$\leq J_1 + J_2 + J_3$$

Equality (a) uses the definition of $P^{\mathrm{MF}}(\cdot,\cdot,\cdot)$ as shown in (8) and equality $(b)$ uses the definition of $\boldsymbol{\mu}_t^N$. The term $J_1$ obeys the following.

$$J_1 \triangleq \frac{1}{N}\sum_{x\in\mathcal{X}}\mathbb{E}\left|\sum_{i=1}^N\delta(x_{t+1}^i = x) - \sum_{i=1}^N P(x_t^i,u_t^i,\boldsymbol{\mu}_t^N,g_t^N,\boldsymbol{\nu}_t^N)(x)\right|$$

$$= \frac{1}{N}\sum_{x\in\mathcal{X}}\mathbb{E}\left[\mathbb{E}\left[\left|\sum_{i=1}^N\delta(x_{t+1}^i = x) - \sum_{i=1}^N P(x_t^i,u_t^i,\boldsymbol{\mu}_t^N,g_t^N,\boldsymbol{\nu}_t^N)(x)\right|\,\middle|\,\boldsymbol{x}_t^N, g_t^N, \boldsymbol{u}_t^N\right]\right] \overset{(a)}{\leq} \frac{1}{\sqrt{N}}\sqrt{|\mathcal{X}|}$$

Inequality ($a$) is obtained from Lemma 10. In particular, it uses the facts that $\{x_{t+1}^i\}_{i\in\{1,\cdots,N\}}$ are conditionally independent given $\{\boldsymbol{x}_t^N, g_t^N, \boldsymbol{u}_t^N\}$, and the following relations hold

$$\mathbb{E}\left[\delta(x_{t+1}^i = x)\Big|\boldsymbol{x}_t^N, g_t^N, \boldsymbol{u}_t^N\right] = P(x_t^i, u_t^i, \boldsymbol{\mu}_t^N, g_t^N, \boldsymbol{\nu}_t^N)(x),$$

$$\sum_{x\in\mathcal{X}} \mathbb{E}\left[\delta(x_{t+1}^i = x)\Big|\boldsymbol{x}_t^N, g_t^N, \boldsymbol{u}_t^N\right] = 1$$

$\forall i \in \{1, \cdots, N\}$, and $\forall x \in \mathcal{X}$. The second term satisfies the following bound.

$$J_2 \triangleq \frac{1}{N} \sum_{x\in\mathcal{X}} \mathbb{E}\left|\sum_{i=1}^N P(x_t^i, u_t^i, \boldsymbol{\mu}_t^N, g_t^N, \boldsymbol{\nu}_t^N)(x) - \sum_{i=1}^N P(x_t^i, u_t^i, \boldsymbol{\mu}_t^N, g_t^N, \nu^{\mathrm{MF}}(\boldsymbol{\mu}_t^N, g_t^N, \pi_t))(x)\right|$$

$$\leq \frac{1}{N} \sum_{i=1}^N \mathbb{E}\left|P(x_t^i, u_t^i, \boldsymbol{\mu}_t^N, g_t^N, \boldsymbol{\nu}_t^N) - P(x_t^i, u_t^i, \boldsymbol{\mu}_t^N, g_t^N, \nu^{\mathrm{MF}}(\boldsymbol{\mu}_t^N, g_t^N, \pi_t))\right|_1$$

$$\overset{(a)}{\leq} L_P \mathbb{E}\left|\boldsymbol{\nu}_t^N - \nu^{\mathrm{MF}}(\boldsymbol{\mu}_t^N, g_t^N, \pi_t)\right|_1 \overset{(b)}{\leq} \frac{L_P}{\sqrt{N}}\sqrt{|\mathcal{U}|}$$

Inequality (a) is a consequence of Assumption 1(c) while (b) follows from Lemma 11. Finally, the term, $J_3$ can be upper bounded as follows.

$$J_3 \triangleq \frac{1}{N} \sum_{x\in\mathcal{X}} \mathbb{E}\left|\sum_{i=1}^N P(x_t^i, u_t^i, \boldsymbol{\mu}_t^N, g_t^N, \nu^{\mathrm{MF}}(\boldsymbol{\mu}_t^N, g_t^N, \pi_t))(x)\right.$$

$$\left. - \sum_{i=1}^N \sum_{u\in\mathcal{U}} P(x_t^i, u, \boldsymbol{\mu}_t^N, g_t^N, \nu^{\mathrm{MF}}(\boldsymbol{\mu}_t^N, g_t^N, \pi_t))(x)\pi_t(x_t^i, \boldsymbol{\mu}_t^N, g_t^N)(u)\right|$$

$$\overset{(a)}{\leq} \frac{1}{\sqrt{N}}\sqrt{|\mathcal{X}|}$$

Inequality (a) is a result of Lemma 10. In particular, it uses the facts that, $\{u_t^i\}_{i\in\{1,\cdots,N\}}$ are conditionally independent given $\boldsymbol{x}_t^N, g_t^N$, and the following relations hold

$$\mathbb{E}\left[P(x_t^i, u_t^i, \boldsymbol{\mu}_t^N, g_t^N, \nu^{\mathrm{MF}}(\boldsymbol{\mu}_t^N, g_t^N, \pi_t))(x)\Big|\boldsymbol{x}_t^N, g_t^N\right] = \sum_{u\in\mathcal{U}} P(x_t^i, u, \boldsymbol{\mu}_t^N, \nu^{\mathrm{MF}}(\boldsymbol{\mu}_t^N, g_t^N, \pi_t))(x)\pi_t(x_t^i, \boldsymbol{\mu}_t^N, g_t^N)(u),$$

$$\sum_{x\in\mathcal{X}} \mathbb{E}\left[P(x_t^i, u_t^i, \boldsymbol{\mu}_t^N, g_t^N, \nu^{\mathrm{MF}}(\boldsymbol{\mu}_t^N, g_t^N, \pi_t))(x)\Big|\boldsymbol{x}_t^N, g_t^N\right] = 1$$

$\forall i \in \{1, \cdots, N\}$, and $\forall x \in \mathcal{X}$. This concludes the Lemma.

## M   Proof of Lemma 13

Observe the following decomposition.

$$\mathbb{E}\left|\frac{1}{N}\sum_{i=1}^N r(x_t^i, u_t^i, \boldsymbol{\mu}_t^N, g_t^N, \boldsymbol{\nu}_t^N) - r^{\mathrm{MF}}(\boldsymbol{\mu}_t^N, g_t^N, \pi_t)\right|$$

$$\overset{(a)}{=} \mathbb{E}\left|\frac{1}{N}\sum_{i=1}^N r(x_t^i, u_t^i, \boldsymbol{\mu}_t^N, g_t^N, \boldsymbol{\nu}_t^N) - \sum_{x\in\mathcal{X}}\sum_{u\in\mathcal{U}} r(x, u, \boldsymbol{\mu}_t^N, g_t^N, \nu^{\mathrm{MF}}(\boldsymbol{\mu}_t^N, g_t^N, \pi_t))\pi_t(x, \boldsymbol{\mu}_t^N, g_t^N)(u)\boldsymbol{\mu}_t^N(x)\right|$$

$$\overset{(b)}{=} \mathbb{E}\left|\frac{1}{N}\sum_{i=1}^N r(x_t^i, u_t^i, \boldsymbol{\mu}_t^N, g_t^N, \boldsymbol{\nu}_t^N) - \sum_{x\in\mathcal{X}}\sum_{u\in\mathcal{U}} r(x, u, \boldsymbol{\mu}_t^N, g_t^N, \nu^{\mathrm{MF}}(\boldsymbol{\mu}_t^N, g_t^N, \pi_t))\pi_t(x, \boldsymbol{\mu}_t^N, g_t^N)(u)\frac{1}{N}\sum_{i=1}^N\delta(x_t^i = x)\right|$$

$$= \mathbb{E}\left|\frac{1}{N}\sum_{i=1}^N r(x_t^i, u_t^i, \boldsymbol{\mu}_t^N, g_t^N, \boldsymbol{\nu}_t^N) - \frac{1}{N}\sum_{i=1}^N\sum_{u\in\mathcal{U}} r(x_t^i, u, \boldsymbol{\mu}_t^N, g_t^N, \nu^{\mathrm{MF}}(\boldsymbol{\mu}_t^N, g_t^N, \pi_t))\pi_t(x_t^i, \boldsymbol{\mu}_t^N, g_t^N)(u)\right| \leq J_1 + J_2$$

Equation (a) uses the definition of $r^{\mathrm{MF}}(\cdot,\cdot,\cdot)$ as shown in (10). Inequality (b) uses the definition of $\boldsymbol{\mu}_t^N$. The term, $J_1$, obeys the following.

$$
\begin{aligned}
J_1 &\triangleq \frac{1}{N}\mathbb{E}\left|\sum_{i=1}^N r(x_t^i,u_t^i,\boldsymbol{\mu}_t^N,g_t^N,\boldsymbol{\nu}_t^N) - \sum_{i=1}^N r(x_t^i,u_t^i,\boldsymbol{\mu}_t^N,g_t^N,\nu^{\mathrm{MF}}(\boldsymbol{\mu}_t^N,g_t^N,\pi_t))\right| \\
&\leq \frac{1}{N}\mathbb{E}\sum_{i=1}^N \left|r(x_t^i,u_t^i,\boldsymbol{\mu}_t^N,g_t^N,\boldsymbol{\nu}_t^N) - r(x_t^i,u_t^i,\boldsymbol{\mu}_t^N,g_t^N,\nu^{\mathrm{MF}}(\boldsymbol{\mu}_t^N,g_t^N,\pi_t))\right| \\
&\overset{(a)}{\leq} L_R\mathbb{E}\left|\boldsymbol{\nu}_t^N - \nu^{\mathrm{MF}}(\boldsymbol{\mu}_t^N,g_t^N,\pi_t)\right|_1 \overset{(b)}{\leq} \frac{L_R}{\sqrt{N}}\sqrt{|\mathcal{U}|}
\end{aligned}
$$

Inequality (a) results from Assumption 1(b), whereas (b) is a consequence of Lemma 11. The term, $J_2$, obeys the following.

$$
\begin{aligned}
J_2 &\triangleq \frac{1}{N}\mathbb{E}\left|\sum_{i=1}^N r(x_t^i,u_t^i,\boldsymbol{\mu}_t^N,g_t^N,\nu^{\mathrm{MF}}(\boldsymbol{\mu}_t^N,g_t^N,\pi_t)) - \sum_{i=1}^N \sum_{u\in\mathcal{U}} r(x_t^i,u,\boldsymbol{\mu}_t^N,g_t^N,\nu^{\mathrm{MF}}(\boldsymbol{\mu}_t^N,g_t^N,\pi_t))\pi_t(x_t^i,\boldsymbol{\mu}_t^N,g_t^N)(u)\right| \\
&= \frac{1}{N}\mathbb{E}\left[\mathbb{E}\left[\left|\sum_{i=1}^N r(x_t^i,u_t^i,\boldsymbol{\mu}_t^N,g_t^N,\nu^{\mathrm{MF}}(\boldsymbol{\mu}_t^N,g_t^N,\pi_t))\right.\right.\right. \\
&\qquad\qquad \left.\left.\left. - \sum_{i=1}^N \sum_{u\in\mathcal{U}} r(x_t^i,u,\boldsymbol{\mu}_t^N,g_t^N,\nu^{\mathrm{MF}}(\boldsymbol{\mu}_t^N,g_t^N,\pi_t))\pi_t(x_t^i,\boldsymbol{\mu}_t^N,g_t^N)(u)\right|\,\right|\boldsymbol{x}_t^N,g_t^N\right]\right] \\
&= \frac{M}{N}\mathbb{E}\left[\mathbb{E}\left[\left|\sum_{i=1}^N r_0(x_t^i,u_t^i,\boldsymbol{\mu}_t^N,g_t^N,\nu^{\mathrm{MF}}(\boldsymbol{\mu}_t^N,g_t^N,\pi_t))\right.\right.\right. \\
&\qquad\qquad \left.\left.\left. - \sum_{i=1}^N \sum_{u\in\mathcal{U}} r_0(x_t^i,u,\boldsymbol{\mu}_t^N,g_t^N,\nu^{\mathrm{MF}}(\boldsymbol{\mu}_t^N,g_t^N,\pi_t))\pi_t(x_t^i,\boldsymbol{\mu}_t^N,g_t^N)(u)\right|\,\right|\boldsymbol{x}_t^N,g_t^N\right]\right] \overset{(a)}{\leq} \frac{M_R}{\sqrt{N}}
\end{aligned}
$$

where $r_0(\cdot,\cdot,\cdot,\cdot) \triangleq r(\cdot,\cdot,\cdot,\cdot)/M_R$. Inequality (a) follows from Lemma 10. In particular, it uses the fact that $\{u_t^i\}_{i\in\{1,\cdots,N\}}$ are conditionally independent given $\boldsymbol{x}_t^N,g_t^N$, and the following relations hold.

$$
|r_0(x_t^i,u_t^i,\boldsymbol{\mu}_t^N,g_t^N,\nu^{\mathrm{MF}}(\boldsymbol{\mu}_t^N,g_t^N,\pi_t))| \leq 1,
$$

$$
\mathbb{E}\left[r_0(x_t^i,u_t^i,\boldsymbol{\mu}_t^N,g_t^N,\nu^{\mathrm{MF}}(\boldsymbol{\mu}_t^N,g_t^N,\pi_t))\Big|\boldsymbol{x}_t^N,g_t^N\right] = \sum_{u\in\mathcal{U}} r_0(x_t^i,u,\boldsymbol{\mu}_t^N,g_t^N,\nu^{\mathrm{MF}}(\boldsymbol{\mu}_t^N,g_t^N,\pi_t))\pi_t(x_t^i,\boldsymbol{\mu}_t^N,g_t^N)(u)
$$

$\forall i \in \{1,\cdots,N\}, \forall u \in \mathcal{U}$.

## N Proof of Theorem 2

The following results are needed to prove the theorem.

### N.1 Continuity Lemmas

In the following lemmas, $\pi \in \Pi$ is an arbitrary policy and $\boldsymbol{\mu}, \bar{\boldsymbol{\mu}} \in \Delta(\mathcal{X})$ are arbitrary local state distributions.

**Lemma 14.** *If $P^{\mathrm{MF}}(\cdot,\cdot,\cdot)$ is defined by (8), then the following relation holds $\forall g \in \mathcal{G}$.*

$$
|P^{\mathrm{MF}}(\boldsymbol{\mu},g,\pi) - P^{\mathrm{MF}}(\bar{\boldsymbol{\mu}},g,\pi)|_1 \leq Q_P|\boldsymbol{\mu} - \bar{\boldsymbol{\mu}}|_1
$$

*where $Q_P \triangleq 1 + L_P + L_Q$.*

**Lemma 15.** *If $r^{\mathrm{MF}}(\cdot,\cdot,\cdot)$ is defined by (10), then the following relation holds $\forall g \in \mathcal{G}$.*

$$
|r^{\mathrm{MF}}(\boldsymbol{\mu},g,\pi) - r^{\mathrm{MF}}(\bar{\boldsymbol{\mu}},g,\pi)| \leq Q_R|\boldsymbol{\mu} - \bar{\boldsymbol{\mu}}|_1
$$

*where $Q_R \triangleq M_R(1 + L_Q) + L_R$.*

The proofs of Lemma $14 - 15$ are relegated to Appendix $O-P$. In the following three lemmas, we show the continuity of the functions defined in Appendix A.2. Proofs of the following lemmas are relegated to Appendix $Q-S$.

**Lemma 16.** *The following relations hold* $\forall l, r \in \{0, 1, \cdots\}$, $\forall \boldsymbol{\mu}_l, \bar{\boldsymbol{\mu}}_l \in \Delta(\mathcal{X})$, $\forall g_{l:l+r} \in \mathcal{G}^{r+1}$ *and* $\forall \pi_{l:l+r} \in \Pi^{r+1}$.

$$|\tilde{P}^{\mathrm{MF}}(\boldsymbol{\mu}_l, g_{l:l+r}, \pi_{l:l+r}) - \tilde{P}^{\mathrm{MF}}(\bar{\boldsymbol{\mu}}_l, g_{l:l+r}, \pi_{l:l+r})|_1 \le Q_P^{r+1}|\boldsymbol{\mu}_l - \bar{\boldsymbol{\mu}}_l|_1$$

*where* $Q_P$ *is defined in Lemma 14.*

**Lemma 17.** *The following relations hold* $\forall l, r \in \{0, 1, \cdots\}$, $\forall \boldsymbol{\mu}_l, \bar{\boldsymbol{\mu}}_l \in \Delta(\mathcal{X})$, $\forall g_{l:l+r} \in \mathcal{G}^{r+1}$ *and* $\forall \pi_{l:l+r} \in \Pi^{r+1}$.

$$\sum_{l+1:l+r} \left|\tilde{P}_G^{\mathrm{MF}}(\boldsymbol{\mu}_l, g_{l:l+r}, \pi_{l:l+r}) - \tilde{P}_G^{\mathrm{MF}}(\bar{\boldsymbol{\mu}}_l, g_{l:l+r}, \pi_{l:l+r})\right|_1 \le L_G(1 + Q_P + \cdots + Q_P^r)|\boldsymbol{\mu}_l - \bar{\boldsymbol{\mu}}_l|_1$$

*where* $\sum_{l+1:l+r}$ *indicates a summation operation over* $\{g_{l+1}, \cdots, g_{l+r}\} \in \mathcal{G}^r$ *for* $r \ge 1$ *and an identity operation for* $r = 0$. *The term* $Q_P$ *is defined in Lemma 14.*

**Lemma 18.** *The following relations hold* $\forall l, r \in \{0, 1, \cdots\}$, $\forall \boldsymbol{\mu}_l, \bar{\boldsymbol{\mu}}_l \in \Delta(\mathcal{X})$, $\forall g_l \in \mathcal{G}$ *and* $\forall \pi_{l:l+r} \in \Pi^r$.

$$|\tilde{r}^{\mathrm{MF}}(\boldsymbol{\mu}_l, g_l, \pi_{l:l+r}) - \tilde{r}^{\mathrm{MF}}(\bar{\boldsymbol{\mu}}_l, g_l, \pi_{l:l+r})| \le \left[\left(\frac{M_R L_G}{Q_P - 1}\right)(Q_P^r - 1) + Q_R Q_P^r\right]|\boldsymbol{\mu}_l - \bar{\boldsymbol{\mu}}_l|_1$$

*where* $Q_P$ *and* $Q_R$ *are defined in Lemma 14 and 15 respectively.*

## N.2   Approximation Lemmas

We use the same notation introduced in A.3.

**Lemma 19.** *The following inequality holds* $\forall t \in \{0, 1, \cdots\}$.

$$\mathbb{E}\left|\boldsymbol{\mu}_{t+1}^N - P^{\mathrm{MF}}(\boldsymbol{\mu}_t^N, g_t^N, \pi_t)\right|_1 \le \frac{2}{\sqrt{N}}\sqrt{|\mathcal{X}|} \tag{37}$$

**Lemma 20.** *The following inequality holds* $\forall t \in \{0, 1, \cdots\}$.

$$\mathbb{E}\left|\frac{1}{N}\sum_{i=1}^N r(x_t^i, u_t^i, \boldsymbol{\mu}_t^N, g_t^N) - r^{\mathrm{MF}}(\boldsymbol{\mu}_t^N, g_t^N, \pi_t)\right| \le \frac{M_R}{\sqrt{N}}$$

The proofs of Lemma $19 - 20$ are relegated to Appendix $T-U$.

## N.3   Proof of the Theorem

Let, $\boldsymbol{\pi} = \{\pi_t\}_{t \in \{0, 1, \cdots\}}$ be an arbitrary policy sequence. We shall use the notations introduced in section A.3. Additionally, we shall consider $\{\boldsymbol{\mu}_t, g_t\}$ as the local state distribution and the global state of the infinite agent system at time $t$. Consider the following.

$$|V_N(\boldsymbol{x}_0, g_0, \boldsymbol{\pi}) - V_\infty(\boldsymbol{\mu}_0, g_0, \boldsymbol{\pi})|$$

$$\le \sum_{t=0}^\infty \gamma^t \mathbb{E}\left|\frac{1}{N}\sum_{i=1}^N r(x_t^i, u_t^i, \boldsymbol{\mu}_t^N, g_t^N) - r^{\mathrm{MF}}(\boldsymbol{\mu}_t^N, g_t^N, \pi_t)\right| + \sum_{t=0}^\infty \gamma^t \underbrace{[\mathbb{E}[r^{\mathrm{MF}}(\boldsymbol{\mu}_t^N, g_t^N, \pi_t)] - \mathbb{E}[r^{\mathrm{MF}}(\boldsymbol{\mu}_t, g_t, \pi_t)]]}_{\triangleq J_t}$$

$$\stackrel{(a)}{\le} \left(\frac{M_R}{1-\gamma}\right)\frac{1}{\sqrt{N}} + \sum_{t=0}^\infty \gamma^t J_t$$

$$\tag{38}$$

Inequality ($a$) follows from Lemma 20. Note that, using the definition (31), we can write the following.

$$
\begin{aligned}
J_t &= \left| \mathbb{E}[\tilde{r}^{\mathrm{MF}}(\boldsymbol{\mu}_t^N, g_t^N, \pi_{t:t})] - \mathbb{E}[\tilde{r}^{\mathrm{MF}}(\boldsymbol{\mu}_0, g_0, \pi_{0:t})] \right| \\
&\leq \sum_{k=0}^{t-1} \left| \mathbb{E}[\tilde{r}^{\mathrm{MF}}(\boldsymbol{\mu}_{k+1}^N, g_{k+1}^N, \pi_{k+1:t})] - \mathbb{E}[\tilde{r}^{\mathrm{MF}}(\boldsymbol{\mu}_k^N, g_k^N, \pi_{k:t})] \right| \\
&\overset{(a)}{\leq} \sum_{k=0}^{t-1} \left| \mathbb{E}[\tilde{r}^{\mathrm{MF}}(\boldsymbol{\mu}_{k+1}^N, g_{k+1}^N, \pi_{k+1:t})] - \mathbb{E}\left[ \sum_{g \in \mathcal{G}} \tilde{r}^{\mathrm{MF}}(P^{\mathrm{MF}}(\boldsymbol{\mu}_k^N, g_k^N, \pi_k), g, \pi_{k+1:t}) P_G^{\mathrm{MF}}(\boldsymbol{\mu}_k^N, g_k^N, \pi_k)(g) \right] \right| \\
&\leq \sum_{k=0}^{t-1} \left| \mathbb{E}\left[ \mathbb{E}\left[ \tilde{r}^{\mathrm{MF}}(\boldsymbol{\mu}_{k+1}^N, g_{k+1}^N, \pi_{k+1:t}) \middle| \boldsymbol{x}_k^N, g_k^N, \boldsymbol{u}_k^N \right] \right] \right. \\
&\qquad\qquad \left. - \mathbb{E}\left[ \sum_{g \in \mathcal{G}} \tilde{r}^{\mathrm{MF}}(P^{\mathrm{MF}}(\boldsymbol{\mu}_k^N, g_k^N, \pi_k), g, \pi_{k+1:t}) P_G^{\mathrm{MF}}(\boldsymbol{\mu}_k^N, g_k^N, \pi_k)(g) \right] \right| \\
&\overset{(b)}{=} \sum_{k=0}^{t-1} \left| \mathbb{E}\left[ \mathbb{E}\left[ \sum_{g \in \mathcal{G}} \tilde{r}^{\mathrm{MF}}(\boldsymbol{\mu}_{k+1}^N, g, \pi_{k+1:t}) P_G(\boldsymbol{\mu}_k^N, g_k^N)(g) \middle| \boldsymbol{x}_k^N, g_k^N, \boldsymbol{u}_k^N \right] \right] \right. \\
&\qquad\qquad \left. - \mathbb{E}\left[ \sum_{g \in \mathcal{G}} \tilde{r}^{\mathrm{MF}}(P^{\mathrm{MF}}(\boldsymbol{\mu}_k^N, g_k^N, \pi_k), g, \pi_{k+1:t}) P_G^{\mathrm{MF}}(\boldsymbol{\mu}_k^N, g_k^N, \pi_k)(g) \right] \right| \\
&\leq \sum_{k=0}^{t-1} \sum_{g \in \mathcal{G}} \mathbb{E}\left| \tilde{r}^{\mathrm{MF}}(\boldsymbol{\mu}_{k+1}^N, g, \pi_{k+1:t}) P_G(\boldsymbol{\mu}_k^N, g_k^N)(g) - \tilde{r}^{\mathrm{MF}}(P^{\mathrm{MF}}(\boldsymbol{\mu}_k^N, g_k^N, \pi_k), g, \pi_{k+1:t}) P_G(\boldsymbol{\mu}_k^N, g_k^N)(g) \right| \\
&\leq \sum_{k=0}^{t-1} \sum_{g \in \mathcal{G}} \left| \tilde{r}^{\mathrm{MF}}(\boldsymbol{\mu}_{k+1}^N, g, \pi_{k+1:t}) - \tilde{r}^{\mathrm{MF}}(P^{\mathrm{MF}}(\boldsymbol{\mu}_k^N, g_k^N, \pi_k), g, \pi_{k+1:t}) \right| \times P_G(\boldsymbol{\mu}_k^N, g_k^N)(g) \\
&\overset{(c)}{\leq} \sum_{k=0}^{t-1} \left[ \left( \frac{M_R L_G}{Q_P - 1} \right) (Q_P^{t-k-1} - 1) + Q_R Q_P^{t-k-1} \right] \times \mathbb{E}\left| \boldsymbol{\mu}_{k+1}^N - P^{\mathrm{MF}}(\boldsymbol{\mu}_k^N, g_k^N, \pi_k) \right|_1 \times \underbrace{\left| P_G(\boldsymbol{\mu}_k^N, g_k^N) \right|_1}_{=1} \\
&\overset{(d)}{\leq} \sum_{k=0}^{t-1} \left[ \left( \frac{M_R L_G}{Q_P - 1} \right) (Q_P^{t-k-1} - 1) + Q_R Q_P^{t-k-1} \right] \times \frac{2}{\sqrt{N}} \sqrt{|\mathcal{X}|} \\
&= \left( \frac{2}{Q_P - 1} \right) \left[ \left( \frac{M_R L_G}{Q_P - 1} + Q_R \right) (Q_P^t - 1) - M_R L_G t \right] \times \frac{1}{\sqrt{N}} \sqrt{|\mathcal{X}|}
\end{aligned}
$$

where we use the notation that $\boldsymbol{\mu}_0^N = \boldsymbol{\mu}_0$ and $g_0^N = g_0$. Inequality ($a$) follows from Lemma 9 whereas ($b$) is a consequence of the fact that $\boldsymbol{\mu}_{k+1}^N$ and $g_{k+1}^N$ are conditionally independent given $\boldsymbol{x}_k^N, g_k^N, \boldsymbol{u}_k^N$. Moreover, $g_{k+1}^N \sim P_G(\boldsymbol{\mu}_k^N, g_k^N)$. Inequality ($c$) follows from Lemma 18 while ($d$) is a consequence of Lemma 19. Substituting in (35), we obtain the following result.

$$
\begin{aligned}
|V_N(\boldsymbol{x}_0, g_0, \boldsymbol{\pi}) - V_\infty(\boldsymbol{\mu}_0, g_0, \boldsymbol{\pi})| &\leq \left( \frac{M_R}{1 - \gamma} \right) \frac{1}{\sqrt{N}} \\
&+ \left( \frac{2}{Q_P - 1} \right) \left[ \left( \frac{M_R S_G}{Q_P - 1} + Q_R \right) \left\{ \frac{1}{1 - \gamma Q_P} - \frac{1}{1 - \gamma} \right\} - \left( \frac{M_R L_G}{Q_P - 1} \right) \frac{\gamma}{(1 - \gamma)^2} \right] \times \frac{1}{\sqrt{N}} \sqrt{|\mathcal{X}|}
\end{aligned}
$$

We conclude by noting that $|\sup_{\boldsymbol{\pi}} V_N(\boldsymbol{x}_0, g_0, \boldsymbol{\pi}) - \sup_{\boldsymbol{\pi}} V_\infty(\boldsymbol{\mu}_0, g_0, \boldsymbol{\pi})| \leq \sup_{\boldsymbol{\pi}} |V_N(\boldsymbol{x}_0, g_0, \boldsymbol{\pi}) - V_\infty(\boldsymbol{\mu}_0, g_0, \boldsymbol{\pi})|$ where the suprema are taken over the set of all admissible policy sequences $\Pi^\infty$.

## O   Proof of Lemma 14

Observe that,

$$|P^{\mathrm{MF}}(\boldsymbol{\mu}, g, \pi) - P^{\mathrm{MF}}(\bar{\boldsymbol{\mu}}, g, \pi)|_1 \overset{(a)}{=} \left| \sum_{x \in \mathcal{X}} \sum_{u \in \mathcal{U}} P(x, u, \boldsymbol{\mu}, g) \pi(x, \boldsymbol{\mu}, g)(u) \boldsymbol{\mu}(x) - P(x, u, \bar{\boldsymbol{\mu}}, g) \pi(x, \bar{\boldsymbol{\mu}}, g)(u) \bar{\boldsymbol{\mu}}(x) \right|_1$$

$$\leq J_1 + J_2$$

Equality (a) follows from the definition of $P^{\mathrm{MF}}(\cdot, \cdot, \cdot)$ as depicted in (8). The term $J_1$ satisfies the following bound.

$$J_1 \triangleq \sum_{x \in \mathcal{X}} \sum_{u \in \mathcal{U}} \left| P(x, u, \boldsymbol{\mu}, g) - P(x, u, \bar{\boldsymbol{\mu}}, g) \right|_1 \times \pi(x, \boldsymbol{\mu}, g)(u) \boldsymbol{\mu}(x)$$

$$\overset{(a)}{\leq} L_P |\boldsymbol{\mu} - \bar{\boldsymbol{\mu}}|_1 \times \underbrace{\sum_{x \in \mathcal{X}} \boldsymbol{\mu}(x) \sum_{u \in \mathcal{U}} \pi(x, \boldsymbol{\mu}, g)(u)}_{=1} \overset{(b)}{=} L_P |\boldsymbol{\mu} - \bar{\boldsymbol{\mu}}|_1$$

Inequality $(a)$ is a consequence of Assumption 1(c) whereas $(b)$ follows from the fact that $\pi(x, \boldsymbol{\mu}, g)$, $\boldsymbol{\mu}$ are probability distributions. The second term, $J_2$ obeys the following bound.

$$J_2 \triangleq \sum_{x \in \mathcal{X}} \sum_{u \in \mathcal{U}} \underbrace{|P(x, u, \bar{\boldsymbol{\mu}}, g)|_1}_{=1} \times |\pi(x, \boldsymbol{\mu}, g)(u) \boldsymbol{\mu}(x) - \pi(x, \bar{\boldsymbol{\mu}}, g)(u) \bar{\boldsymbol{\mu}}(x)|$$

$$\leq \sum_{x \in \mathcal{X}} |\boldsymbol{\mu}(x) - \bar{\boldsymbol{\mu}}(x)| \underbrace{\sum_{u \in \mathcal{U}} \pi(x, \boldsymbol{\mu}, g)(u)}_{=1} + \sum_{x \in \mathcal{X}} \bar{\boldsymbol{\mu}}(x) \sum_{u \in \mathcal{U}} |\pi(x, \boldsymbol{\mu}, g)(u) - \pi(x, \bar{\boldsymbol{\mu}}, g)(u)|$$

$$\overset{(a)}{\leq} |\boldsymbol{\mu} - \bar{\boldsymbol{\mu}}|_1 + \underbrace{\sum_{x \in \mathcal{X}} \bar{\boldsymbol{\mu}}(x)}_{=1} L_Q |\boldsymbol{\mu} - \bar{\boldsymbol{\mu}}|_1 \overset{(b)}{=} (1 + L_Q)|\boldsymbol{\mu} - \bar{\boldsymbol{\mu}}|_1$$

Inequality $(a)$ results from Assumption 2 and the fact that $\pi(x, \boldsymbol{\mu}, g)$ is a probability distribution whereas $(b)$ utilizes the fact that $\bar{\boldsymbol{\mu}}$ is a distribution. This concludes the result.

## P   Proof of Lemma 15

Observe that,

$$|r^{\mathrm{MF}}(\boldsymbol{\mu}, g, \pi) - r^{\mathrm{MF}}(\bar{\boldsymbol{\mu}}, g, \pi)|_1 \overset{(a)}{=} \left| \sum_{x \in \mathcal{X}} \sum_{u \in \mathcal{U}} r(x, u, \boldsymbol{\mu}, g) \pi(x, \boldsymbol{\mu}, g)(u) \boldsymbol{\mu}(x) - r(x, u, \bar{\boldsymbol{\mu}}, g) \pi(x, \bar{\boldsymbol{\mu}}, g)(u) \bar{\boldsymbol{\mu}}(x) \right|_1$$

$$\leq J_1 + J_2$$

Equality (a) follows from the definition of $r^{\mathrm{MF}}(\cdot, \cdot, \cdot)$ as depicted in (10). The term $J_1$ satisfies the following bound.

$$J_1 \triangleq \sum_{x \in \mathcal{X}} \sum_{u \in \mathcal{U}} \left| r(x, u, \boldsymbol{\mu}, g) - r(x, u, \bar{\boldsymbol{\mu}}, g) \right| \times \pi(x, \boldsymbol{\mu}, g)(u) \boldsymbol{\mu}(x) \overset{(a)}{\leq} L_R |\boldsymbol{\mu} - \bar{\boldsymbol{\mu}}|_1 \times \underbrace{\sum_{x \in \mathcal{X}} \boldsymbol{\mu}(x) \sum_{u \in \mathcal{U}} \pi(x, \boldsymbol{\mu}, g)(u)}_{=1}$$

$$\overset{(b)}{=} L_R |\boldsymbol{\mu} - \bar{\boldsymbol{\mu}}|_1$$

Inequality $(a)$ is a consequence of Assumption 1$(b)$ whereas $(b)$ follows from the fact that $\pi(x, \boldsymbol{\mu}, g)$, $\boldsymbol{\mu}$ are probability distributions. The second term, $J_2$ obeys the following bound.

$$
\begin{aligned}
J_2 &\triangleq \sum_{x \in \mathcal{X}} \sum_{u \in \mathcal{U}} |r(x, u, \bar{\boldsymbol{\mu}}, g)| \times |\pi(x, \boldsymbol{\mu}, g)(u)\boldsymbol{\mu}(x) - \pi(x, \bar{\boldsymbol{\mu}}, g)(u)\bar{\boldsymbol{\mu}}(x)| \\
&\stackrel{(a)}{\leq} M_R \sum_{x \in \mathcal{X}} |\boldsymbol{\mu}(x) - \bar{\boldsymbol{\mu}}(x)| \underbrace{\sum_{u \in \mathcal{U}} \pi(x, \boldsymbol{\mu}, g)(u)}_{=1} + M_R \sum_{x \in \mathcal{X}} \bar{\boldsymbol{\mu}}(x) \sum_{u \in \mathcal{U}} |\pi(x, \boldsymbol{\mu}, g)(u) - \pi(x, \bar{\boldsymbol{\mu}}, g)(u)| \\
&\stackrel{(b)}{\leq} M_R |\boldsymbol{\mu} - \bar{\boldsymbol{\mu}}|_1 + M_R \underbrace{\sum_{x \in \mathcal{X}} \bar{\boldsymbol{\mu}}(x)}_{=1} L_Q |\boldsymbol{\mu} - \bar{\boldsymbol{\mu}}|_1 \stackrel{(c)}{=} M_R(1 + L_Q)|\boldsymbol{\mu} - \bar{\boldsymbol{\mu}}|_1
\end{aligned}
$$

Inequality $(a)$ results from Assumption 1(a) where $(b)$ follows from Assumption 2 and the fact that $\pi(x, \boldsymbol{\mu}, g)$ is a probability distribution. Finally, $(c)$ utilizes the fact that $\bar{\boldsymbol{\mu}}$ is a valid distribution. This concludes the result.

## Q   Proof of Lemma 16

Fix $l \in \{0, 1, \cdots\}$. We shall prove the lemma via induction on $r$. Note that, for $r = 0$, we have,

$$
|\tilde{P}^{\mathrm{MF}}(\boldsymbol{\mu}_l, g_{l:l}, \pi_{l:l}) - \tilde{P}^{\mathrm{MF}}(\bar{\boldsymbol{\mu}}_l, g_{l:l}, \pi_{l:l})|_1 = |P^{\mathrm{MF}}(\boldsymbol{\mu}_l, g_l, \pi_l) - P^{\mathrm{MF}}(\bar{\boldsymbol{\mu}}_l, g_l, \pi_l)|_1 \stackrel{(a)}{\leq} Q_P |\boldsymbol{\mu}_l - \bar{\boldsymbol{\mu}}_l|_1
$$

Inequality $(a)$ follows from Lemma 14. Assume that the lemma holds for some $r \in \{0, 1, \cdots\}$. We shall demonstrate below that the relation holds for $r + 1$ as well. Note that,

$$
\begin{aligned}
&|\tilde{P}^{\mathrm{MF}}(\boldsymbol{\mu}_l, g_{l:l+r+1}, \pi_{l:l+r+1}) - \tilde{P}^{\mathrm{MF}}(\bar{\boldsymbol{\mu}}_l, g_{l:l+r+1}, \pi_{l:l+r+1})|_1 \\
&= |P^{\mathrm{MF}}(\tilde{P}^{\mathrm{MF}}(\boldsymbol{\mu}_l, g_{l:l+r}, \pi_{l:l+r}), g_{l+r+1}, \pi_{l+r+1}) - P^{\mathrm{MF}}(\tilde{P}^{\mathrm{MF}}(\bar{\boldsymbol{\mu}}_l, g_{l:l+r}, \pi_{l:l+r}), g_{l+r+1}, \pi_{l+r+1})|_1 \\
&\stackrel{(a)}{\leq} Q_P |\tilde{P}^{\mathrm{MF}}(\boldsymbol{\mu}_l, g_{l:l+r}, \pi_{l:l+r}) - \tilde{P}^{\mathrm{MF}}(\bar{\boldsymbol{\mu}}_l, g_{l:l+r}, \pi_{l:l+r})|_1 \stackrel{(b)}{\leq} Q_P^{r+2} |\boldsymbol{\mu}_l - \bar{\boldsymbol{\mu}}_l|_1
\end{aligned}
$$

Inequality $(a)$ follows from Lemma 14 and $(b)$ is a consequence of the induction hypothesis.

## R   Proof of Lemma 17

Fix $l \in \{0, 1, \cdots\}$. We shall prove the lemma via induction on $r$. Note that, for $r = 0$, we have,

$$
\begin{aligned}
|\tilde{P}_G^{\mathrm{MF}}(\boldsymbol{\mu}_l, g_{l:l}, \pi_{l:l}) - \tilde{P}_G^{\mathrm{MF}}(\bar{\boldsymbol{\mu}}_l, g_{l:l}, \pi_{l:l})|_1 &= |P_G^{\mathrm{MF}}(\boldsymbol{\mu}_l, g_l, \pi_l) - P_G^{\mathrm{MF}}(\bar{\boldsymbol{\mu}}_l, g_l, \pi_l)|_1 \\
&= |P_G(\boldsymbol{\mu}_l, g_l) - P_G(\bar{\boldsymbol{\mu}}_l, g_l)|_1 \stackrel{(a)}{\leq} L_G |\boldsymbol{\mu}_l - \bar{\boldsymbol{\mu}}_l|_1
\end{aligned}
\tag{39}
$$

Equality $(a)$ follows from Assumption 1$(d)$. Assume that the lemma holds for some $r \in \{0, 1, \cdots\}$. We shall now show that the lemma holds for $r + 1$ as well. Note that,

$$
\begin{aligned}
&\sum_{l+1:l+r+1} \left|\tilde{P}_G^{\mathrm{MF}}(\boldsymbol{\mu}_l, g_{l:l+r+1}, \pi_{l:l+r+1}) - \tilde{P}_G^{\mathrm{MF}}(\bar{\boldsymbol{\mu}}_l, g_{l:l+r+1}, \pi_{l:l+r+1})\right|_1 \\
&\stackrel{(a)}{=} \sum_{l+1:l+r+1} \left|\tilde{P}_G^{\mathrm{MF}}(\boldsymbol{\mu}_l, g_{l:l+r}, \pi_{l:l+r})(g_{l+r+1})P_G^{\mathrm{MF}}(\tilde{P}^{\mathrm{MF}}(\boldsymbol{\mu}_l, g_{l:l+r}, \pi_{l:l+r}), g_{l+r+1}, \pi_{l+r+1})\right. \\
&\qquad\qquad \left. - \tilde{P}_G^{\mathrm{MF}}(\bar{\boldsymbol{\mu}}_l, g_{l:l+r}, \pi_{l:l+r})(g_{l+r+1})P_G^{\mathrm{MF}}(\tilde{P}^{\mathrm{MF}}(\bar{\boldsymbol{\mu}}_l, g_{l:l+r}, \pi_{l:l+r}), g_{l+r+1}, \pi_{l+r+1})\right|_1 \leq J_1 + J_2
\end{aligned}
$$

Equality $(a)$ follows from (30). The first term $J_1$ can be upper bounded as follows.

$$
\begin{aligned}
J_1 &\triangleq \sum_{l+1:l+r+1} \tilde{P}_G^{\mathrm{MF}}(\boldsymbol{\mu}_l, g_{l:l+r}, \pi_{l:l+r})(g_{l+r+1}) \\
&\quad \times \left| P_G^{\mathrm{MF}}(\tilde{P}^{\mathrm{MF}}(\boldsymbol{\mu}_l, g_{l:l+r}, \pi_{l:l+r}), g_{l+r+1}, \pi_{l+r+1}) - P_G^{\mathrm{MF}}(\tilde{P}^{\mathrm{MF}}(\bar{\boldsymbol{\mu}}_l, g_{l:l+r}, \pi_{l:l+r}), g_{l+r+1}, \pi_{l+r+1}) \right|_1 \\
&= \sum_{l+1:l+r+1} \tilde{P}_G^{\mathrm{MF}}(\boldsymbol{\mu}_l, g_{l:l+r}, \pi_{l:l+r})(g_{l+r+1}) \\
&\quad \times \left| P_G(\tilde{P}^{\mathrm{MF}}(\boldsymbol{\mu}_l, g_{l:l+r}, \pi_{l:l+r}), g_{l+r+1}) - P_G(\tilde{P}^{\mathrm{MF}}(\bar{\boldsymbol{\mu}}_l, g_{l:l+r}, \pi_{l:l+r}), g_{l+r+1}) \right|_1 \\
&\overset{(a)}{\leq} L_G \sum_{l+1:l+r+1} \left| \tilde{P}^{\mathrm{MF}}(\boldsymbol{\mu}_l, g_{l:l+r}, \pi_{l:l+r}) - \tilde{P}^{\mathrm{MF}}(\boldsymbol{\mu}_l, g_{l:l+r}, \pi_{l:l+r}) \right|_1 \times \tilde{P}_G^{\mathrm{MF}}(\boldsymbol{\mu}_l, g_{l:l+r}, \pi_{l:l+r})(g_{l+r+1}) \\
&\overset{(b)}{\leq} L_G Q_P^{r+1} |\boldsymbol{\mu}_l - \bar{\boldsymbol{\mu}}_l|_1 \sum_{l+1:l+r+1} \tilde{P}_G^{\mathrm{MF}}(\boldsymbol{\mu}_l, g_{l:l+r}, \pi_{l:l+r})(g_{l+r+1}) \overset{(c)}{=} L_G Q_P^{r+1} |\boldsymbol{\mu}_l - \bar{\boldsymbol{\mu}}_l|_1
\end{aligned}
$$

Inequality $(a)$ follows from Assumption 1$(d)$ while $(b)$ results from Lemma 16. Finally, $(c)$ can be shown following the definition (30). The second term $J_2$ can be bounded as follows.

$$
\begin{aligned}
J_2 &\triangleq \sum_{l+1:l+r+1} \underbrace{|P_G^{\mathrm{MF}}(\tilde{P}^{\mathrm{MF}}(\bar{\boldsymbol{\mu}}_l, g_{l:l+r}, \pi_{l:l+r}), g_{l+r+1}, \pi_{l+r+1})|_1}_{=1} \\
&\quad \times |\tilde{P}_G^{\mathrm{MF}}(\boldsymbol{\mu}_l, g_{l:l+r}, \pi_{l:l+r})(g_{l+r+1}) - \tilde{P}_G^{\mathrm{MF}}(\bar{\boldsymbol{\mu}}_l, g_{l:l+r}, \pi_{l:l+r})(g_{l+r+1})| \\
&= \sum_{l+1:l+r+1} |\tilde{P}_G^{\mathrm{MF}}(\boldsymbol{\mu}_l, g_{l:l+r}, \pi_{l:l+r}) - \tilde{P}_G^{\mathrm{MF}}(\bar{\boldsymbol{\mu}}_l, g_{l:l+r}, \pi_{l:l+r})|_1 \overset{(a)}{\leq} L_G(1 + Q_P + \cdots + Q_P^r) |\boldsymbol{\mu}_l - \bar{\boldsymbol{\mu}}_l|_1
\end{aligned}
$$

Inequality $(a)$ follows from induction hypothesis. This concludes the lemma.

## S    Proof of Lemma 18

Note that the result readily follows for $r = 0$ from Lemma 15. Therefore, we assume $r \geq 1$.

$$
\begin{aligned}
&|\tilde{r}^{\mathrm{MF}}(\boldsymbol{\mu}_l, g_l, \pi_{l:l+r}) - \tilde{r}^{\mathrm{MF}}(\bar{\boldsymbol{\mu}}_l, g_l, \pi_{l:l+r})| \\
&\overset{(a)}{\leq} \sum_{l+1:l+r} |r^{\mathrm{MF}}(\tilde{P}^{\mathrm{MF}}(\boldsymbol{\mu}_l, g_{l:l+r-1}, \pi_{l:l+r-1}), g_{l+r}, \pi_{l+r}) \tilde{P}_G^{\mathrm{MF}}(\boldsymbol{\mu}_l, g_{l:l+r-1}, \pi_{l:l+r-1})(g_{l+r}) \\
&\quad - r^{\mathrm{MF}}(\tilde{P}^{\mathrm{MF}}(\bar{\boldsymbol{\mu}}_l, g_{l:l+r-1}, \pi_{l:l+r-1}), g_{l+r}, \pi_{l+r}) \tilde{P}_G^{\mathrm{MF}}(\bar{\boldsymbol{\mu}}_l, g_{l:l+r-1}, \pi_{l:l+r-1})(g_{l+r})| \leq J_1 + J_2
\end{aligned}
$$

Inequality $(a)$ follows from the definition (31). The first term can be bounded as follows.

$$
\begin{aligned}
J_1 &\triangleq \sum_{l+1:l+r} |r^{\mathrm{MF}}(\tilde{P}^{\mathrm{MF}}(\boldsymbol{\mu}_l, g_{l:l+r-1}, \pi_{l:l+r-1}), g_{l+r}, \pi_{l+r})| \\
&\quad \times |\tilde{P}_G^{\mathrm{MF}}(\boldsymbol{\mu}_l, g_{l:l+r-1}, \pi_{l:l+r-1})(g_{l+r}) - \tilde{P}_G^{\mathrm{MF}}(\bar{\boldsymbol{\mu}}_l, g_{l:l+r-1}, \pi_{l:l+r-1})(g_{l+r})| \\
&\overset{(a)}{\leq} M_R \sum_{l+1:l+r-1} |\tilde{P}_G^{\mathrm{MF}}(\boldsymbol{\mu}_l, g_{l:l+r-1}, \pi_{l:l+r-1}) - \tilde{P}_G^{\mathrm{MF}}(\bar{\boldsymbol{\mu}}_l, g_{l:l+r-1}, \pi_{l:l+r-1})|_1 \\
&\overset{(b)}{\leq} M_R L_G(1 + Q_P + \cdots + Q_P^{r-1}) |\boldsymbol{\mu}_l - \bar{\boldsymbol{\mu}}_l|_1 = \left( \frac{M_R L_G}{Q_P - 1} \right) (Q_P^r - 1) |\boldsymbol{\mu}_l - \bar{\boldsymbol{\mu}}_l|_1
\end{aligned}
$$

The bound $(a)$ can be proven using Assumption 1$(a)$ and the definition of $r^{\mathrm{MF}}$ given in (10). The bound $(b)$ follows from Lemma 17. The term $J_2$ can be bounded as follows.

$$
\begin{aligned}
J_2 &\triangleq \sum_{l+1:l+r} \tilde{P}_G^{\mathrm{MF}}(\bar{\boldsymbol{\mu}}_l, g_{l:l+r-1}, \pi_{l:l+r-1})(g_{l+r}) \\
&\qquad \times |r^{\mathrm{MF}}(\tilde{P}^{\mathrm{MF}}(\boldsymbol{\mu}_l, g_{l:l+r-1}, \pi_{l:l+r-1}), g_{l+r}, \pi_{l+r}) - r^{\mathrm{MF}}(\tilde{P}^{\mathrm{MF}}(\bar{\boldsymbol{\mu}}_l, g_{l:l+r-1}, \pi_{l:l+r-1}), g_{l+r}, \pi_{l+r})| \\
&\stackrel{(a)}{\leq} \sum_{l+1:l+r} \tilde{P}_G^{\mathrm{MF}}(\bar{\boldsymbol{\mu}}_l, g_{l:l+r-1}, \pi_{l:l+r-1})(g_{l+r}) \times Q_R |\tilde{P}^{\mathrm{MF}}(\boldsymbol{\mu}_l, g_{l:l+r-1}, \pi_{l:l+r-1}) - \tilde{P}^{\mathrm{MF}}(\bar{\boldsymbol{\mu}}_l, g_{l:l+r-1}, \pi_{l:l+r-1})| \\
&\stackrel{(b)}{\leq} Q_R Q_P^r |\boldsymbol{\mu}_l - \bar{\boldsymbol{\mu}}_l|_1 \times \sum_{l+1:l+r} \tilde{P}_G^{\mathrm{MF}}(\bar{\boldsymbol{\mu}}_l, g_{l:l+r-1}, \pi_{l:l+r-1})(g_{l+r}) \stackrel{(c)}{=} Q_R Q_P^r |\boldsymbol{\mu}_l - \bar{\boldsymbol{\mu}}_l|_1
\end{aligned}
$$

Inequality $(a)$ follows from Lemma 15 while $(b)$ results from Lemma 16. Finally, $(c)$ can be proven from the definition (30). This concludes the lemma.

## T    Proof of Lemma 19

Notice the following decomposition.

$$
\begin{aligned}
&\mathbb{E}\left|\boldsymbol{\mu}_{t+1}^N - P^{\mathrm{MF}}(\boldsymbol{\mu}_t^N, g_t^N, \pi_t)\right|_1 \\
&\stackrel{(a)}{=} \mathbb{E}\left|\boldsymbol{\mu}_{t+1}^N - \sum_{x' \in \mathcal{X}}\sum_{u \in \mathcal{U}} P(x', u, \boldsymbol{\mu}_t^N, g_t^N)\pi_t(x', \boldsymbol{\mu}_t^N, g_t^N)(u)\boldsymbol{\mu}_t^N(x)\right|_1 \\
&\stackrel{(b)}{=} \sum_{x \in \mathcal{X}} \mathbb{E}\left|\frac{1}{N}\sum_{i=1}^N \delta(x_{t+1}^i = x) - \sum_{x' \in \mathcal{X}}\sum_{u \in \mathcal{U}} P(x', u, \boldsymbol{\mu}_t^N, g_t^N)(x)\pi_t(x', \boldsymbol{\mu}_t^N, g')(u)\frac{1}{N}\sum_{i=1}^N \delta(x_t^i = x')\right| \\
&= \sum_{x \in \mathcal{X}} \mathbb{E}\left|\frac{1}{N}\sum_{i=1}^N \delta(x_{t+1}^i = x) - \frac{1}{N}\sum_{i=1}^N \sum_{u \in \mathcal{U}} P(x_t^i, u, \boldsymbol{\mu}_t^N, g_t^N)(x)\pi_t(x_t^i, \boldsymbol{\mu}_t^N, g_t^N)(u)\right| \leq J_1 + J_2
\end{aligned}
$$

Equality (a) uses the definition of $P^{\mathrm{MF}}(\cdot, \cdot, \cdot)$ as shown in (8) and equality (b) uses the definition of $\boldsymbol{\mu}_t^N$. The term $J_1$ obeys the following.

$$
\begin{aligned}
J_1 &\triangleq \frac{1}{N}\sum_{x \in \mathcal{X}} \mathbb{E}\left|\sum_{i=1}^N \delta(x_{t+1}^i = x) - \sum_{i=1}^N P(x_t^i, u_t^i, \boldsymbol{\mu}_t^N, g_t^N)(x)\right| \\
&= \frac{1}{N}\sum_{x \in \mathcal{X}} \mathbb{E}\left[\mathbb{E}\left[\left|\sum_{i=1}^N \delta(x_{t+1}^i = x) - \sum_{i=1}^N P(x_t^i, u_t^i, \boldsymbol{\mu}_t^N, g_t^N)(x)\right| \,\middle|\, \boldsymbol{x}_t^N, g_t^N, \boldsymbol{u}_t^N\right]\right] \stackrel{(a)}{\leq} \frac{1}{\sqrt{N}}\sqrt{|\mathcal{X}|}
\end{aligned}
$$

Inequality $(a)$ is obtained applying Lemma 10, and the facts that $\{x_{t+1}^i\}_{i \in \{1,\cdots,N\}}$ are conditionally independent given $\{\boldsymbol{x}_t^N, g_t^N, \boldsymbol{u}_t^N\}$, and the following relations hold

$$
\begin{aligned}
\mathbb{E}\left[\delta(x_{t+1}^i = x)\,\middle|\,\boldsymbol{x}_t^N, g_t^N, \boldsymbol{u}_t^N\right] &= P(x_t^i, u_t^i, \boldsymbol{\mu}_t^N, g_t^N)(x), \\
\sum_{x \in \mathcal{X}} \mathbb{E}\left[\delta(x_{t+1}^i = x)\,\middle|\,\boldsymbol{x}_t^N, g_t^N, \boldsymbol{u}_t^N\right] &= 1
\end{aligned}
$$

$\forall i \in \{1, \cdots, N\}$, and $\forall x \in \mathcal{X}$. The second term satisfies the following bound.

$$
J_2 \triangleq \frac{1}{N}\sum_{x \in \mathcal{X}} \mathbb{E}\left|\sum_{i=1}^N P(x_t^i, u_t^i, \boldsymbol{\mu}_t^N, g_t^N)(x) - \sum_{i=1}^N \sum_{u \in \mathcal{U}} P(x_t^i, u, \boldsymbol{\mu}_t^N, g_t^N)(x)\pi_t(x_t^i, \boldsymbol{\mu}_t^N, g_t^N)(u)\right| \stackrel{(a)}{\leq} \frac{1}{\sqrt{N}}\sqrt{|\mathcal{X}|}
$$

Inequality (a) is a result of Lemma 10. In particular, it uses the facts that, $\{u_t^i\}_{i \in \{1, \cdots, N\}}$ are conditionally independent given $\boldsymbol{x}_t^N, g_t^N$, and the following relations hold

$$\mathbb{E}\left[P(x_t^i, u_t^i, \boldsymbol{\mu}_t^N, g_t^N)(x)\Big|\boldsymbol{x}_t^N, g_t^N\right] = \sum_{u \in \mathcal{U}} P(x_t^i, u, \boldsymbol{\mu}_t^N, g_t^N)(x)\pi_t(x_t^i, \boldsymbol{\mu}_t^N, g_t^N)(u),$$

$$\sum_{x \in \mathcal{X}} \mathbb{E}\left[P(x_t^i, u_t^i, \boldsymbol{\mu}_t^N, g_t^N)(x)\Big|\boldsymbol{x}_t^N, g_t^N\right] = 1$$

$\forall i \in \{1, \cdots, N\}$, and $\forall x \in \mathcal{X}$. This concludes the Lemma.

## U   Proof of Lemma 20

Observe the following decomposition.

$$\mathbb{E}\left|\frac{1}{N}\sum_{i=1}^{N} r(x_t^i, u_t^i, \boldsymbol{\mu}_t^N, g_t^N) - r^{\mathrm{MF}}(\boldsymbol{\mu}_t^N, g_t^N, \pi_t)\right|$$

$$\overset{(a)}{=} \mathbb{E}\left|\frac{1}{N}\sum_{i=1}^{N} r(x_t^i, u_t^i, \boldsymbol{\mu}_t^N, g_t^N) - \sum_{x \in \mathcal{X}}\sum_{u \in \mathcal{U}} r(x, u, \boldsymbol{\mu}_t^N, g_t^N)\pi_t(x, \boldsymbol{\mu}_t^N, g_t^N)(u)\boldsymbol{\mu}_t^N(x)\right|$$

$$\overset{(b)}{=} \mathbb{E}\left|\frac{1}{N}\sum_{i=1}^{N} r(x_t^i, u_t^i, \boldsymbol{\mu}_t^N, g_t^N) - \sum_{x \in \mathcal{X}}\sum_{u \in \mathcal{U}} r(x, u, \boldsymbol{\mu}_t^N, g_t^N)\pi_t(x, \boldsymbol{\mu}_t^N, g_t^N)(u)\frac{1}{N}\sum_{i=1}^{N}\delta(x_t^i = x)\right|$$

$$= \mathbb{E}\left|\frac{1}{N}\sum_{i=1}^{N} r(x_t^i, u_t^i, \boldsymbol{\mu}_t^N, g_t^N) - \frac{1}{N}\sum_{i=1}^{N}\sum_{u \in \mathcal{U}} r(x_t^i, u, \boldsymbol{\mu}_t^N, g_t^N)\pi_t(x_t^i, \boldsymbol{\mu}_t^N, g_t^N)(u)\right|$$

$$= \frac{1}{N}\mathbb{E}\left[\mathbb{E}\left[\left|\sum_{i=1}^{N} r(x_t^i, u_t^i, \boldsymbol{\mu}_t^N, g_t^N) - \sum_{i=1}^{N}\sum_{u \in \mathcal{U}} r(x_t^i, u, \boldsymbol{\mu}_t^N, g_t^N)\pi_t(x_t^i, \boldsymbol{\mu}_t^N, g_t^N)(u)\right|\Big|\boldsymbol{x}_t^N, g_t^N\right]\right]$$

$$= \frac{M}{N}\mathbb{E}\left[\mathbb{E}\left[\left|\sum_{i=1}^{N} r_0(x_t^i, u_t^i, \boldsymbol{\mu}_t^N, g_t^N) - \sum_{i=1}^{N}\sum_{u \in \mathcal{U}} r_0(x_t^i, u, \boldsymbol{\mu}_t^N, g_t^N)\pi_t(x_t^i, \boldsymbol{\mu}_t^N, g_t^N)(u)\right|\Big|\boldsymbol{x}_t^N, g_t^N\right]\right] \overset{(c)}{\leq} \frac{M_R}{\sqrt{N}}$$

Equation (a) uses the definition of $r^{\mathrm{MF}}(\cdot, \cdot)$ as shown in (10) while inequality (b) uses the definition of $\boldsymbol{\mu}_t^N$. The function $r_0$ is given as, $r_0(\cdot, \cdot, \cdot, \cdot) \triangleq r(\cdot, \cdot, \cdot, \cdot)/M_R$. Finally, relation (c) follows from Lemma 10. In particular, it uses the fact that $\{u_t^i\}_{i \in \{1, \cdots, N\}}$ are conditionally independent given $\boldsymbol{x}_t^N, g_t^N$, and the following relations hold.

$$|r_0(x_t^i, u_t^i, \boldsymbol{\mu}_t^N, g_t^N)| \leq 1,$$

$$\mathbb{E}\left[r_0(x_t^i, u_t^i, \boldsymbol{\mu}_t^N, g_t^N)\Big|\boldsymbol{x}_t^N, g_t^N\right] = \sum_{u \in \mathcal{U}} r_0(x_t^i, u, \boldsymbol{\mu}_t^N, g_t^N)\pi_t(x_t^i, \boldsymbol{\mu}_t^N, g_t^N)(u)$$

$\forall i \in \{1, \cdots, N\}, \forall u \in \mathcal{U}$.

# V   Sampling Procedure

---

**Algorithm 2** Sampling Algorithm

---

1: **Input:** $\boldsymbol{\mu}_0$, $g_0$, $\boldsymbol{\pi}_{\Phi_j}$, $P$, $P_G$, $r$
2: Sample $u_0 \sim \pi_{\Phi_j}(x_0, \boldsymbol{\mu}_0, g_0)$
3: $\boldsymbol{\nu}_0 \leftarrow \nu^{\mathrm{MF}}(\boldsymbol{\mu}_0, g_0, \pi_{\Phi_j})$ where $\nu^{\mathrm{MF}}$ is defined in (7).

4: $t \leftarrow 0$
5: $\mathrm{FLAG} \leftarrow \mathrm{FALSE}$
6: **while** FLAG is FALSE **do**
7:   FLAG $\leftarrow$ TRUE with probability $1 - \gamma$.
8:   Execute Update
9: **end while**
10: $T \leftarrow t$
11: Accept $(x_T, \boldsymbol{\mu}_T, g_T, u_T)$ as a sample.

12: $\hat{V}_{\Phi_j} \leftarrow 0$, $\hat{Q}_{\Phi_j} \leftarrow 0$
13: $\mathrm{FLAG} \leftarrow \mathrm{FALSE}$
14: $\mathrm{SumRewards} \leftarrow 0$
15: **while** FLAG is FALSE **do**
16:   FLAG $\leftarrow$ TRUE with probability $1 - \gamma$.
17:   Execute Update
18:   $\mathrm{SumRewards} \leftarrow \mathrm{SumRewards} + r(x_t, u_t, \boldsymbol{\mu}_t, g_t, \boldsymbol{\nu}_t)$
19: **end while**

20: With probability $\frac{1}{2}$, $\hat{V}_{\Phi_j} \leftarrow \mathrm{SumRewards}$. Otherwise $\hat{Q}_{\Phi_j} \leftarrow \mathrm{SumRewards}$.
21: $\hat{A}_{\Phi_j}(x_T, \boldsymbol{\mu}_T, g_T, u_T) \leftarrow 2(\hat{Q}_{\Phi_j} - \hat{V}_{\Phi_j})$.
22: **Output:** $(x_T, \boldsymbol{\mu}_T, g_T, u_T)$ and $\hat{A}_{\Phi_j}(x_T, \boldsymbol{\mu}_T, g_T, u_T)$

**Procedure** Update:
23: $x_{t+1} \sim P(x_t, u_t, \boldsymbol{\mu}_t, g_t, \boldsymbol{\nu}_t)$.
24: $g_{t+1} \sim P_G(x_t, u_t, \boldsymbol{\mu}_t, g_t, \boldsymbol{\nu}_t)$.
25: $\boldsymbol{\mu}_{t+1} \leftarrow P^{\mathrm{MF}}(\boldsymbol{\mu}_t, \pi_{\Phi_j})$ where $P^{\mathrm{MF}}$ is defined in (8).
26: $u_{t+1} \sim \pi_{\Phi_j}(x_{t+1}, \boldsymbol{\mu}_{t+1}, g_{t+1})$
27: $\boldsymbol{\nu}_{t+1} \leftarrow \nu^{\mathrm{MF}}(\boldsymbol{\mu}_{t+1}, g_{t+1}, \pi_{\Phi_j})$
28: $t \leftarrow t + 1$
**EndProcedure**

---

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
