# OpenReview forum: "Mean-Field Control based Approximation of Multi-Agent Reinforcement Learning in Presence of a Non-decomposable Shared Global State"
_TMLR — Accepted by TMLR_

### Review · Reviewer_dGHi · 2023-03-27

**Summary Of Contributions:**

This paper considers multi-agent reinforcement learning (MARL, or more precisely, Markov N-player games) with both agent states and global states, and extends the mean-field control (MFC) approximation theory (which originally only works without global states) to this setting. When the dynamics and rewards are independent of the action mean-field, an approximation error bound independent of the action space is also obtained. The authors then propose a natural policy gradient (NPG) algorithm for solving MFC with global states and provides an error bound against the original N-player game problem.

**Audience:**

Yes

**Claims And Evidence:**

Yes

**Requested Changes:**

Some suggested changes (that won't affect the essential weaknesses mentioned above):
* If I understand correctly, the authors assume the conditional independence among the global states and local states. If so, this should be made clearer.
* The results of the paper are not stated in a very useful manner. In practice, people apply the MFC policy to the original N-player problem, and hence the results should all be stated in terms of such kind of plug-in errors. For example, in Theorem 1, it should state something like if we solve the MFC with local and global states to an $\epsilon$-error and get a policy $\pi$, then what is the upper bound on $\max_{\pi'} V_N(x_0,g_0,\pi') - V_N(x_0,g_0,\pi)$. Similarly for Theorems 2 and 3.
* The contractivity assumptions that $\gamma S_P<1$ and $\gamma Q_P<1$, etc. make the results only applicable to those problems with small Lipschitz constants or small discount factors, which can be easily violated in practice. This is even more stringent considering the limitation of Lipschitz continuity of the policies mentioned above. And just curious, do we need these contractivity assumptions for finite-horizon problems as well?
* The NPG algorithm is applied to MFC models, but since the original problem is the $N$-player games, it's impractical to get samples from the mean-field transitions. So this is not really reinforcement learning (RL) but some "artificial" RL with a mean-field control sampling oracle. See e.g., [E] for more explanations on the difference between using an $N$-player simulator vs. a mean-field simulator.
* Is Lemma 1 a direct application of Theorem 4.9 in Liu et al., (2020)? If so, the authors should make this clearer.
* The dependency on $J$, the number of iterations should be made explicit in Theorem 3.

[E] Yardim, B., Cayci, S., Geist, M., & He, N. (2022). Policy Mirror Ascent for Efficient and Independent Learning in Mean Field Games. arXiv preprint arXiv:2212.14449.

**Strengths And Weaknesses:**

Strengths:
* Game problems with both global and local states are ubiquitous and are important topics that are not (sufficiently) addressed in the existing mean-field control approximation theory. And to my knowledge, this is the first paper attempting at establishing the mean-field control approximation theory for Markov games with both local and (non-decomposable) global states.
* The analysis that considers policy history in a Markov game in Section 6 seems to be novel and interesting.

Weaknesses:
* The result requires lots of stringent assumptions such as Lipschitz policy class (which may not cover or not even close to the optimal N-player policy) and some sort of contractivity like $\gamma S_P<1$ in Theorem 1. Although these are typically assumed in the MFC approximation literature (which is not good for the future of this direction in general), they still largely restrict the applicability and meaningfulness of the results.
* On a related point, this paper is a variant of a series of papers [A-D] in MFC approximation beyond the classical MFC setting, but all these papers including this one follow the same routine, despite some technical differences and novelties due to the variations like the global states in this paper. No efforts are made to address the significant limitations of the assumptions in the literature as mentioned above, and the NPG convergence analysis seem to be mostly the same except for some slight differences.


[A] Mondal, W. U., Agarwal, M., Aggarwal, V., & Ukkusuri, S. V. (2022). On the approximation of cooperative heterogeneous multi-agent reinforcement learning (marl) using mean field control (mfc). Journal of Machine Learning Research, 23(129), 1-46.

[B] Mondal, W. U., Aggarwal, V., & Ukkusuri, S. V. (2022, August). Can mean field control (mfc) approximate cooperative multi agent reinforcement learning (marl) with non-uniform interaction?. In Uncertainty in Artificial Intelligence (pp. 1371-1380). PMLR.

[C] Mondal, W. U., Aggarwal, V., & Ukkusuri, S. V. (2022). On the near-optimality of local policies in large cooperative multi-agent reinforcement learning. arXiv preprint arXiv:2209.03491.

[D] Uddin Mondal, W., Aggarwal, V., & Ukkusuri, S. V. (2022). Mean-Field Approximation of Cooperative Constrained Multi-Agent Reinforcement Learning (CMARL). arXiv e-prints, arXiv-2209.

---

> ### Author Response · Authors · 2023-04-15
> **Response to Reviewer dGHi**
>
> $\textbf{Q1: Clarification regarding the conditional independence of the local and the global states.}$
>
> $\textbf{Response:}$ We have mentioned in the revised manuscript that the local and the global states are conditionally independent. Specifically, we have added the following sentence (section 2).
>
> > It is assumed that the random variables $\\{\\{x\_{t+1}^i\\}\_{i=1}^N, g\_{t+1}^N\\}$ are independent, conditioned on $\\{\boldsymbol{x}\_t^N, g\_t^N, \boldsymbol{u}\_t^N\\}$.
>
> $\textbf{Q2: Regarding the plug-in bounds.}$
>
> $\textbf{Response:}$ The plug-in bounds can be easily obtained from our main results. The process is briefly described in section 4 of the revised manuscript. We quote the following excerpt for reference.
>
> > Let, $\boldsymbol{\pi}\_{\epsilon}^{\mathrm{MF}}\in \Pi^{\infty}$ be an admissible policy sequence that solves the MFC problem with $\epsilon$ accuracy. Note that,
> > $$ |\sup\_{\boldsymbol{\pi}}V\_N(\boldsymbol{x}\_0, g\_0, \boldsymbol{\pi})-V_N(\boldsymbol{x}\_0, g\_0, \boldsymbol{\pi}\_\epsilon^{\mathrm{MF}})|\leq \underbrace{|\sup\_{\boldsymbol{\pi}}V\_N(\boldsymbol{x}\_0, g\_0, \boldsymbol{\pi})-\sup\_{\boldsymbol{\pi}}V_\infty(\boldsymbol{\mu}\_0, g\_0, \boldsymbol{\pi})|}\_{\triangleq J\_1}+\underbrace{|\sup_{\boldsymbol{\pi}}V\_\infty(\boldsymbol{\mu}_0, g\_0, \boldsymbol{\pi}) - V\_\infty(\boldsymbol{\mu}\_0, g\_0, \boldsymbol{\pi}\_\epsilon^{\mathrm{MF}})|}\_{\triangleq J_2} + \underbrace{|V\_\infty(\boldsymbol{\mu}_0, g_0, \boldsymbol{\pi}\_\epsilon^{\mathrm{MF}}) - V\_N(\boldsymbol{x}\_0, g\_0, \boldsymbol{\pi}\_\epsilon^{\mathrm{MF}})|}\_{\triangleq J\_3}$$
> > The terms $J_1, J_3$ can be bounded by  Theorem 1 while $J_2$ can be bounded by $\epsilon$. This result suggests that if we can come up with a way to approximately solve the MFC problem, then that solution must be a good proxy for the optimal MARL solution.
>
> $\textbf{Q3: On the assumption of small $\gamma$.}$
>
> $\textbf{Response:}$ We agree with the reviewer  that the constraint $\gamma S_P<1$ limits the applicability of mean-field control. We would like to state that similar restriction also appears in other mean-field related papers [1]-[3] and hence, it is not a special artifact of our techniques. In an $H$ (finite) horizon problem, the upper bound of the MARL-MFC approximation error will be proportional to $\left[1-(\gamma S_P)^H\right]/[1-\gamma S_P]$. Clearly, if $\gamma S_P<1$, then the error bound does not increase with $H$. However, if $\gamma S_P>1$, the  bound increases exponentially with $H$. More investigations are needed to assess whether the restriction $\gamma S_P<1$ is an artifact of the proof techniques available in the literature or it is a fundamental limitation of the mean-field philosophy itself.
>
> [1] Gu, H., Guo, X., Wei, X., \& Xu, R. (2021). Mean-field controls with Q-learning for cooperative MARL: convergence and complexity analysis. SIAM Journal on Mathematics of Data Science, 3(4), 1168-1196.
>
> [2] Pasztor, Barna, Ilija Bogunovic, and Andreas Krause. ``Efficient model-based multi-agent mean-field reinforcement learning." arXiv preprint arXiv:2107.04050 (2021).
>
> [3] Mondal, W. U., Agarwal, M., Aggarwal, V., and Ukkusuri, S. V. (2022). On the approximation of cooperative heterogeneous multi-agent reinforcement learning (MARL) using mean field control (MFC). Journal of Machine Learning Research, 23(129), 1-46.
>
> $\textbf{Q4: On the mean-field oracle.}$
>
> $\textbf{Response:}$ The existence of an oracle that can sequentially yield state distributions of an infinite agent system using mean-field update equations is either explicitly or implicitly assumed in most of the mean-field related papers. We are no exception. Although the emerging field of mean-field MARL (MARL) attempts to get rid of this assumption by relying only on $N$-agent samples, as pointed out in our response to Q4 of Reviewer UELm, such approaches are not without limitations. Our best bet to solve a general MARL problem is to keep exploring different competing approaches to generate novel insights.
>
> $\textbf{Q5: Is Lemma 1 a direct application of Theorem 4.9 in Liu et al., (2020)?}$
>
> $\textbf{Response:}$ We have explicitly stated in our revised manuscript that Lemma 1 is a direct application of Theorem 4.9 of the above-mentioned paper. Specifically, the following line has been added.
>
> > We are now ready to state the convergence result which is a direct application of Theorem 4.9 of (Liu et al., 2020).
>
> $\textbf{Q6: The dependency on the number of iterations should be made explicit in Theorem 3.}$
>
> $\textbf{Response:}$ In Lemma 1 and Theorem 3 of the revised manuscript, we have elaborated the connection between the number of iterations, $J$, and the approximation error, $\epsilon$.

---

### Review · Reviewer_UELm · 2023-03-27

**Summary Of Contributions:**

In this paper, the authors establish mean-field control (MFC) approximation theory to allow for both local and (non-decomposable) global states, whereas the existing MFC approximation theory only allows for local/agent states. The approximation bound is independent of the global state space sizes. An improved bound when the rewards and transitions are independent of the mean-field action distributions is obtained, which does not depend on the local action space sizes. A natural policy gradient (NPG) algorithm is also proposed and the convergence guarantee is established.

**Audience:**

Yes

**Broader Impact Concerns:**

None.

**Claims And Evidence:**

Yes

**Requested Changes:**

Here we detail some of the weaknesses above and provide some suggestions for potential changes and improvements.
* The challenge mentioned at the bottom of page 2 seems artificial, since if I understand correctly, the global and local states $g_t^N$ and $x_t^i$ are conditionally independent, and hence it's more natural to consider $(\delta(x_t^i=x))_{i=1}^N$ and $\delta(g_t^N=g)$ (OpenReview markdown does not recognize curly brackets for some reason, so I use parentheses here) separately, in which case these $N+1$ indicators are conditionally independent, and there is no mean-field approximation for the last state. I understand that there might still be other challenges, but the motivation here is weird and not convincing. The authors should provide a more convincing explanation on the challenge of the non-decomposable global state.
* The assumption that the policy is Lipschitz in the mean-field term is stringent, in that it's unclear when the globally optimal policy for the MFC is Lipschitz continuous. This is even more severe as we are also comparing against the $N$-player Lipschitz continuous policies in the main theorems, and it's more unclear when the globally optimal policy for the original $N$-player game is a restriction of a Lipschitz policy on the empirical state distribution (or how close they are). This is essentially critical as here we consider finite state and action spaces, in which case Lipschitz continuity for the $N$-player policies does not even make sense in general (without extension/restriction).



**Strengths And Weaknesses:**

Strengths:
* Global (non-decomposable) states are commonly seen in reality but existing mean-field approximation theory has ignored this, especially when both global and local states are present. This paper tries to address this issue for the first time, which is good.
* A computational algorithm is provided to solve the MFC with both local and global states.

Weaknesses (see the requested changes below for more details):
* The technical challenge mentioned at the bottom of page 2 for the additional global state is not that convincing.
* The assumptions are too restrictive, especially the contractivity related ones in all these theorems.
* The NPG algorithm and analysis seem to have nothing really new, and is merely a direct application/restatement of existing results.
* On a higher level, I'm not sure about how important it is to study the MFC approximation problem given many evidences from the recent literature that we don't need to take the mean-field limit in order to utilize the symmetry in (cooperative) MARL problems. See e.g., [1,2].


[1] Gu, H., Guo, X., Wei, X., & Xu, R. (2021). Mean-field multi-agent reinforcement learning: A decentralized network approach. arXiv preprint arXiv:2108.02731.

[2] Chen, M., Li, Y., Wang, E., Yang, Z., Wang, Z., & Zhao, T. (2021). Pessimism meets invariance: Provably efficient offline mean-field multi-agent RL. Advances in Neural Information Processing Systems, 34, 17913-17926.

---

> ### Author Response · Authors · 2023-04-15
> **Response to Reviewer UELm: Part 1**
>
> $\textbf{Q1: Clarifying the challenge of the paper.}$
>
> $\textbf{Response:}$ In traditional MFC (without non-decomposable global state), one of the crucial steps in proving the MARL-MFC approximation error is bounding the term $\mathbb{E}|\boldsymbol{\mu}\_t^N-\boldsymbol{\mu}\_t|\_1$ as $\mathcal{O}(1/\sqrt{N})$ where $\boldsymbol{\mu}\_t^N$,  $\boldsymbol{\mu}\_t$ are the state-distributions of the $N$-agent and the infinite agent systems respectively at time $t$. The key intuition in proving this bound comes from the fact that $\boldsymbol{\mu}_t^N(x)$ can be written as the average of the random variables $\\{\delta(x\_t^i=x)\\}\_{i=1}^N$ where $x\_t^i$ is the state of the $i$th agent at time $t$. Note that, the above-mentioned random variables are independent conditioned on $\boldsymbol{\mu}\_{t-1}^N$. This allows one to use the law of large numbers and obtain the desired bound.
>
> If we follow the same footsteps in our setting (with a non-decomposable state), we end up with the term $\sum_{x,g}\mathbb{E}|\boldsymbol{\mu}\_t^N(x)\delta(g_t^N=g)-\boldsymbol{\mu}\_t(x)\delta(g\_t=g)|$ where $g\_t^N$, $g\_t$ are the non-decomposable states in the $N$-agent and the infinite agent systems respectively. Note that, we can write $\boldsymbol{\mu}\_t^N(x)\delta(g\_t^N=g)$ as the average of the random variables $\\{\delta(x\_t^i=x, g\_t^N=g)\\}\_{i=1}^N$. Due to the common factor $\delta(g\_t^N=g)$, these random variables are now correlated, and hence the law of large numbers can no longer be applied. This is essentially the primary challenge addressed by our paper. We have invented new techniques, elaborated in section 6 of the paper, to prove MARL-MFC approximation guarantees even when the structure of conditional independence is altered due to the presence of a non-decomposable state. We have revised our explanation on page 2 of the paper.
>
> $\textbf{Q2: Clarification regarding the assumption of Lipschitz continuity of the policy functions.}$
>
> $\textbf{Response:}$ Although the state space, $\mathcal{X}$, and the action spaces, $\mathcal{U}$ of each agent are presumed to be finite, due to homogeneity and exchangeability of the agents, individual reward and state transition functions can be written as functions of the state and action distributions of the entire population. Specifically, the reward function, $r_i$, the local state transition function, $P_i$ of the $i$th agent, and the global state transition function, $P_G$ can be simplified as follows.
>
> > $$ r_i(\boldsymbol{x}_t^N, g_t^N, \boldsymbol{u}_t^N) = r(x_t^i, u_t^i, \boldsymbol{\mu}_t^N, g_t^N, \boldsymbol{\nu}_t^N), $$
> > $$ P_i(\boldsymbol{x}_t^N, g_t^N, \boldsymbol{u}_t^N) = P(x_t^i, u_t^i, \boldsymbol{\mu}_t^N, g_t^N, \boldsymbol{\nu}_t^N), $$
> > $$ P_G(\boldsymbol{x}_t^N, g_t^N, \boldsymbol{u}_t^N) = P_G(\boldsymbol{\mu}_t^N, g_t^N, \boldsymbol{\nu}_t^N) $$
>
> where $x_t^i, u_i^t$ respectively denote the (local) state and action of the $i$th agent at time $t$, $g_t^N$ is the non-decomposable state at the same instant,  $\boldsymbol{x}\_t^N\triangleq\\{x\_t^i\\}\_{i=1}^N$, and $\boldsymbol{u}\_t^N\triangleq\\{u\_t^i\\}\_{i=1}^N$. Moreover, $\boldsymbol{\mu}\_t^N$,  $\boldsymbol{\nu}\_t^N$ are the (local) state and action distributions of the entire population at time $t$. This simplification allows one to express the policy functions to be of the form, $\pi:\mathcal{X}\times \Delta(\mathcal{X})\rightarrow \Delta(\mathcal{U})$. Observe that, although the first argument of $\pi$ is discrete, the second argument is continuous. This explains why defining Lipschitz continuity of $\pi$ with respect to the second argument is justified despite $\mathcal{X}, \mathcal{U}$ being finite.
>
> Secondly, we would like to clarify that, although, ideally, we would like to find the optimal policy over the set of all policies, such a task is often not feasible, especially if the state space is either large or infinite. In such a case, it is a common practice to narrow down the search to a set of admissible policies, denoted as $\Pi$. In our paper, we establish the approximation guarantees (Theorem 1,  2) only within a policy set, $\Pi$, whose elements are Lipschitz continuous (Assumption 2). Although such restrictions are not desirable, the assumption of Lipschitz continuity frequently appears in the mean-field literature [1]-[3].
>
> [1] Gu, H., Guo, X., Wei, X., \& Xu, R. (2021). Mean-field controls with Q-learning for cooperative MARL: convergence and complexity analysis. SIAM Journal on Mathematics of Data Science, 3(4), 1168-1196.
>
> [2] Pasztor, Barna, Ilija Bogunovic, and Andreas Krause. ``Efficient model-based multi-agent mean-field reinforcement learning." arXiv preprint arXiv:2107.04050 (2021).
>
> [3] Mondal, W. U., Agarwal, M., Aggarwal, V., and Ukkusuri, S. V. (2022). On the approximation of cooperative heterogeneous multi-agent reinforcement learning (MARL) using mean field control (MFC). Journal of Machine Learning Research, 23(129), 1-46.

---

> > ### Comment · Reviewer_UELm · 2023-05-14
> > **About Q1 and Q2**
> >
> > Thanks for the detailed responses.
> > * For Q1, my point is that due to the conditional independence between global and local states, and since there is no mean-field approximation for the global states, I think the challenge introduced by coupling them into the same indicator and talking about the correlation among agents due to global state copying is artificial. Intuitively, the global states should just be left as is instead of worrying about anything regarding the mean-field approximation. I understand there should be other challenges and it's not a big issue anyway as it's just a motivation. But overall I think this motivation is not convincing and can be removed without any sacrifice of clarity or contribution.
> > * For Q2, I understand that even with finite state and actions, the mean-field term is a continuous vector/matrix/tensor (and in continuous state-action cases it becomes a more abstract measure stuff). And I agree that restricting policy class is reasonable in general. However, for an N-player game, even if we represent the problem using empirical distributions, they are not really continuous as they cannot take arbitrary values in the probability simplex. To define Lipschitz continuity w.r.t., these empirical distributions, we are implicitly making some assumption about function domain extension. It's okay to have gaps due to policy class restrictions, but it's important to understand how large the gap is and in this case, due to the discrete-vs-continuous difference between empirical and generic distributions, the gap is pretty nontrivial to characterize. Please at least add some comments on this limitation in the paper so that readers are aware of such a restriction.

---

> > > ### Author Response · Authors · 2023-05-19
> > > **Response**
> > >
> > > $\textbf{Q1:}$ We would like to clarify that if one follows the standard mean-field proof techniques such as those given in [1], (without artificially lumping different terms together or trying to force a mean-field approximation when there is none), one would eventually end up with the following error.
> > > $$\sum_{x,g}|\boldsymbol{\mu}_t^N(x)\delta(g_t^N=g)-\boldsymbol{\mu}(x)\delta(g_t=g)|$$
> > >
> > > where the notations are the same as in our previous response. In contrast, the standard MFC analysis (without the common state) leads to the following error.
> > >
> > > $$\sum_{x}|\boldsymbol{\mu}_t^N(x)-\boldsymbol{\mu}_t(x)|$$
> > >
> > > Note that while the second term can be shown to be $\mathcal{O}(1/\sqrt{N})$, a similar result cannot be established for the first term. The explanation given in the paper (where $x_t^i, g_t^N$ are lumped together) was an attempt to intuitively clarify why, unlike the second term, the first term cannot be reduced to zero for large $N$. In other words, it is not that our bias toward the global state to behave like a mean-field term is creating all the nuisance, it is just that the standard techniques are inadequate and we are attempting to create an intuition to explain that failure.
> > >
> > > [1] Mondal, W. U., Agarwal, M., Aggarwal, V., & Ukkusuri, S. V. (2022). On the approximation of cooperative heterogeneous multi-agent reinforcement learning (MARL) using mean field control (MFC). Journal of Machine Learning Research, 23(129), 1-46.
> > >
> > > $\textbf{Q2:}$ Thank you for the comment. We have added the following lines in the revised paper (page 5).
> > >
> > > > We would like to point out that although the state and action distributions are treated as continuous variables, in an $N$-agent problem, they can only take a finite number of values in their respective probability simplexes. In the MFC problem, however, these variables can be arbitrary. Therefore, while comparing the $N$-agent and the MFC problem, we are implicitly extending the domain of definition for the reward and the state transition functions.

---

> ### Author Response · Authors · 2023-04-15
> **Response to Reviewer UELm: Part 2**
>
> $\textbf{Q3: Novelty of the analysis of the NPG-based algorithm.}$
>
> $\textbf{Response:}$ We acknowledge that NPG-related analysis is not our primary contribution. We have explicitly mentioned in the revised manuscript that Lemma 1 is a direct application of Theorem 4.9 of (Liu et. al., 2020).  Our primary contributions are characterizing the approximation error between MARL and MFC problems (Theorem 1, 2), and Theorem 3 which combines the approximation results with Lemma 1 to produce convergence guarantees for the $N$-agent problem.
>
>  $\textbf{Q4: The importance of studying mean-field control in general.}$
>
> $\textbf{Response:}$ As mentioned by the reviewer, the field of mean-field MARL (MF-MARL) discussed in [1], [2] attempts to solve an $N$-agent problem without imposing the infinite agent limit. However, this approach is not without limitations. For example, [1] defines the concept of connectedness between two arbitrary states, not between agents. This is reflected in the definition of the policy function, $\pi$ which takes $s_t^i$, (the state of the $i$th agent at time $t$) and, $\mu(s_t^i)$, (the $s_t^i$-th component of the population state distribution) as an input. Clearly, the model of [1] is different from the standard MARL model discussed in our article. On the other hand, [2] analyzes an off-learning procedure in a linear MDP setting that is far more restrictive than our Lipschitz continuous setup. MFC-based learning, thus, is still relevant in the multi-agent literature. Moreover, it is sometimes helpful to study two competing approaches to solve a single problem because innovation in one area might fuel innovation in the other.
>
>  [1] Gu, H., Guo, X., Wei, X., and Xu, R. (2021). Mean-field multi-agent reinforcement learning: A decentralized network approach. arXiv preprint arXiv:2108.02731.
>
>  [2] Chen, M., Li, Y., Wang, E., Yang, Z., Wang, Z., and Zhao, T. (2021). Pessimism meets invariance: Provably efficient offline mean-field multi-agent RL. Advances in Neural Information Processing Systems, 34, 17913-17926.

---

> > ### Comment · Reviewer_UELm · 2023-05-14
> > **Thanks for the comments**
> >
> > Thanks for the confirmation and comments regarding Q3 and sharing your thoughts on Q4. Particularly, for Q4, I agree that these existing works all have certain limitations in the settings but overall, I would really like to see the community to rethink about the stringent assumptions like Lipschitz continuity and contractivity in MFC and its real benefits over directly solving N-player cooperative games as unlike competitive settings, it's naturally an MDP on a larger state-action space and there are so many positive theoretical and numerical results showing that one can easily solve N-player cooperative games even when N is very large, especially when decentralized algorithms are adopted. But in any case, this is beyond the scope of this paper and it's more of a community's joint future endeavor. Thanks again!

---

> > > ### Author Response · Authors · 2023-05-19
> > > **Response**
> > >
> > > Thanks for your constructive comments. We agree that the mean-field approach's benefits need to be carefully evaluated, and more emphasis should be given to solving the $N$-agent problem directly.

---

### Review · Reviewer_BwjS · 2023-03-31

**Summary Of Contributions:**

This paper considers mean field control (MFC) problems in discrete time in which there is a common state. The MFC framework has been developed to approximate control problems with a large number of indistinguishable agents. Here, on top of the individual state for each agent, there is a common state which influences all the agents and, hence, the population distribution. There are two main contributions.

- First, the authors study the connection between this MFC problem and the N-agent problem. They show (Theorem 1) that the optimal values are close and obtain an upper bound of order $\frac{\sqrt{|\mathcal{X}|} + \sqrt{|\mathcal{U}|}}{\sqrt{N}}$, where $\mathcal{X}$ and $\mathcal{U}$ are respectively the state space and the action space, and $N$ is the number of agents.

- Second, they propose a natural policy gradient algorithm for this MFC problem, analyze its convergence (Theorem 3), and illustrate it on an example.

**Audience:**

Yes

**Broader Impact Concerns:**

I don't think there are any broader impact concerns.

**Claims And Evidence:**

Yes

**Requested Changes:**

Critical points:

**Question 1:** Is your setting a special case of MFC with common noise and, if yes, are your results special cases of existing ones?

To be specific, let us write the evolution in terms of transition functions instead of transition kernels. Imagine that $P(x,u, \boldsymbol{\mu},g,\boldsymbol{\nu})$ is the law of $F_x(x,u, \boldsymbol{\mu},g,\boldsymbol{\nu},\epsilon)$ for some transition function $F_x$ and some random variable $\epsilon$, and $P_G(\boldsymbol{\mu},g,\boldsymbol{\nu})$ is the law of $F_G(\boldsymbol{\mu},g,\boldsymbol{\nu},\epsilon^0)$ for some transition function $F_G$ and some random variable $\epsilon^0$, in such a way that

$$x_{t+1} = F_x(x_t, u_t, \boldsymbol{\mu}_t,g_t,\boldsymbol{\nu}_t, \epsilon_t),$$

and

$$g_{t+1} = F_G(\boldsymbol{\mu}_t,g_t,\boldsymbol{\nu}_t,\epsilon^0_t).$$

Now, define
$
X_t = (x_t, g_t),
$
whose transition is given by:

$$
X_{t+1} = F(X_t, u_t, \mathbb{P}^0_{(X_t,u_t)}, \epsilon_t, \epsilon^0_t)
$$

where $\mathbb{P}^0_{(X_t,u_t)}$ denotes the conditional joint state-action distribution given the common noise $\epsilon^0$, and the transition function $F$ is given by:

$$
F((x,g), u, \boldsymbol{\theta},\epsilon,\epsilon^0) = \Big( F_x(x,u, \boldsymbol{\theta}_x,g,\boldsymbol{\theta}_u, \epsilon), F_G(\boldsymbol{\theta}_x,g,\boldsymbol{\theta}_u, \epsilon^0)\Big),
$$

with $P$ and $P_G$ defined as in your work, and $\boldsymbol{\theta}_x$ and $\boldsymbol{\theta}_u$ denote the marginals of the joint distribution on $x$ and $u$ respectively.

With this new state $X=(x,g)$, it seems that the MFC problem fits in the framework of e.g.:
- Motte, M. and Pham, H., 2022. Mean-field Markov decision processes with common noise and open-loop controls. The Annals of Applied Probability, 32(2), pp.1421-1458.
- Carmona, R., Laurière, M. and Tan, Z., 2019. Model-free mean-field reinforcement learning: mean-field MDP and mean-field Q-learning. arXiv preprint arXiv:1910.12802.

**Question 2**: How does the output of Algorithm 1 _really_ relates to the original problem? In particular:

- In the original problem, the optimization is over all policies and not stationary ones. On Page 10, you claim that it is sufficient to consider stationary policies by invoking Puterman's book, but could you clarify which result exactly you use? It does not seem to me that the MFC setting easily fits in the one covered by Puterman's results.

- How does the result of Theorem 3 reflects the fact that you only allow a specific class of parameterized policies instead of all possible policies? On page 10 again, you claim that you "characterize the elements of $\Pi$ by a $d$-dimensional vector $\Phi$". However, since policies take the state distribution as an input, it does not seem that we can characterize all such policies using a finite dimensional vector. So we can expect a gap between the original problem (with optimization over $\Pi$), and the problem with optimization over a class of parameterized policies.

- The Algorithm 1 relies on samples, so the policies $\Phi_j$ are stochastic. Does the bound in Theorem 3 hold for any realization of the samples? Or should it be understood in expectation or in an other probabilistic sense? (Please also check below for further questions that could help improve the statements of Lemma 1 and Theorem 3.)


Other questions (less critical or just to improve the paper):

- Page 2: “comprising of N number of agents”

- Page 3: Are the state and action spaces assumed to be finite?

- Page 3: Equation (6): Please clarify how the policy is used. In particular, are all the agents using the same policy, and if yes, why?

- Page 3: “admittable”: “admissible”? How is defined the set $\Pi$ of “allowable policies”?

- Page 4: “for the agents to (probabilistically) choose actions based on its current local state”: “their current”?

- Page 5: For Assumption 1, when you say that “Such assumptions are common in the mean-field literature”, could you please specify in which part of the books (Carmona et al., 2018) is the same assumption used? The books are quite long and they deal mostly with the continuous setting, which is not the same as yours.

- Page 6: Theorem 1: Here, it is written “max” over policies $\pi$, which implicitly means that there is a policy achieving the maximum value. Could you please clarify why such a policy exists, for $V_N$ and $V_\infty$?

- Page 6: “the function r and P”: “the functions r and P”

- Page 7: “this can be compactly written as follows”: It is not clear that (15) is more compact that (14).

- Page 7: after (15): “we denote this conditional joint probability as”: This notation is a bit strange because it is supposed to represent the conditional probability of a sequence of $g$ but it is actually evaluated at $g_t$. So it looks more like the probability of $g_t$ given the past. Please clarify.

- Page 7: after (16), “we can now re-write ...”: Here and at a few other places, it is not very clear which symbol you are defining because the equality symbol with a triangle can go both ways. Please clarify each instance and define the symbols appropriately.

- Page 7: after (17): What do you mean “the summation operation over $g_{1:t}$”? Do you sum over all possible trajectories?

- Page 8, step 2: In the equality in line 1 of this step: Please clarify the definition of the right-hand side with $\tilde{r}^{MF}$ and $\pi_{t:t}$. In the equality in line 2: Why is there an expectation around $\tilde{r}^{MF}$? Just above (17), the same expression did not have any expectation, so it is not clear to me whether $\tilde{r}^{MF}$ is deterministic or stochastic. This seems to be related to the symbol defined in (31) but this is in the Appendix.

- Page 8: I am not sure how inequality (19) is obtained. Could you please explain further?

- Page 8: “with a (k, t)-dependent Lipschitz parameter”: Do you need this parameter to satisfy some boundedness or growth condition? Otherwise, it is not clear what happens when you sum over time steps up to infinity in order to obtain the full value function. As far as I understand, the bound in Lemma 8 for instance involves $S_P^{t-k}$, and I do not see how you can sum over $t$ up to infinity in Step 6 of the proof (page 9) without having the constant in the big $\mathcal{O}$ going to infinity. I imagine that the condition $\gamma S_P$ can be helpful but this is not very clear from your proof. (Actually I think this clearer when reading Appendix A.4, but this point should also be clarified in the main text.)

- Page 9: “reward of this representative”: missing word?

- Page 10: “Without loss of generality, we assume that the optimal policy sequence is stationary (Puterman, 2014).” Could you please provide the exact result that you use in Puterman? Notice that here the state space has a continuous component due to the distribution, and the set of policies that you use is not of the classical form  (functions of the state taking values in the action space).

- Page 10: “We characterize the elements of $\Pi$ ...”: Could you please clarify what you mean? The set of policies $\Pi$ is a set of functions which take an element of $\Delta{\mathcal{X}}$ as one of its inputs. So it seems that the set $\Pi$ is infinite-dimensional and it is not clear how you can have a bijection to a set of finite-dimensional vectors. I can imagine that the set of functions parameterized by $\Phi$ can approximate the set of all policies $\Pi$ up to some precision (bounded away from $0$ as long as $d$ is finite). Could you please provide more details on this point?

- Page 10, line below (21): “the deterministic quantities”: It seems that $\mu_t$ should be stochastic due to the effect of $g_t$. Please clarify.

- Page 10, two lines below (21): I am not sure why the subscript $\Phi$ in this line looks bold while the one in (21) and the one in the line below (22) do not. Please double-check all the instances of $\Phi$, $\Phi_j$, etc.

- Page 10, three lines below (22): “over the set of feasible policies”: I don't think this is correct since here the optimization is only over the set of parameterized policies.

- Page 10: Could you please provide some intuition for equations (24) and (25)? It seems that (25) gives a kind of occupancy measure but with a discount and I am not sure why this is a relevant quantity for the problem. Did it appear (implicitly) in the original formulation of the MFC problem?

- Page 10: Below (26) you explain that the sampling procedure is provided in Algorithm 2. Could you please clarify (perhaps in the appendix) how Algorithm 2 indeed generates sample according to this distribution? Does it generate samples exactly according to this distribution or is there an approximation?

- Page 10 and 11: The assumptions 4 to 7 are quite abstract. Could you please provide at least one example that fits in your assumptions? It seems that Assumptions 5 and 6 only impose constraints on the class of parameterized policies, while Assumptions 4 and 7 also involve the MFC model (in particular the transitions).

- Page 11, Lemma 1: I have mainly four questions. (1) In the statement, you first fix the sequence of outputs of the algorithm and then say “for some $\eta, \alpha, J, L$” but these parameters are used in the algorithm, so the output of the algorithm depends on these parameters. I would suggest to first fix the parameter and then take the sequence output by the algorithm. (2) Furthermore, it is not clear whether the initial values $\Phi_0, \mu_0,g_0$ need to be fixed before the algorithm's parameters or if there are parameters that will work for any initial values. Please rewrite this statement with a clear order of the quantifiers. (3) Also, what does the “constant” in the big $\mathcal{O}$ depend on? (4) Last, if I understand correctly, the sequence of $\Phi_j$ is stochastic in the sense that it depends on the samples that used for the NPG updates. So does the inequality hold only “with high probability”? Or does it hold for any sequence of $\Phi_j$, independently of the samples used in the algorithm?

- Page 12, Theorem 3: I have the same questions as for Lemma 1. Furthermore, is there a gap between $V_N^*$ as defined here and $\max_{\pi} V_N$ as it appears in Theorem 2? I would imagine that this is the case because the set of parameterized policies is a (strict) subset of the set of all policies. Furthermore, I imagine that the gap between the two can be quite large if the set of parameterized policies has to satisfy Assumptions 4 to 7. If this is true, then it is a bit misleading to claim that Algorithm 1 solves the original problem. Could you please comment on this point?

- Page 16: Lemmas 2 to 5: The quantifiers over the policies and the distributions are missing.

- Page 17, below (31), “indicates a summation operation over”: The summation should be over integers so I am not sure what you mean here.

**Strengths And Weaknesses:**

- Strengths:
    - The paper is generally well written.
    - The proposed MFC setting with common state seems relevant for applications.

- Weaknesses:
    - Some notations and concepts need to be clarified.
    - It seems that the MFC setting with common state can fit in the more general setting of MFC with common noise, which has been studied in a few works already, so the link between the two settings (and the existing results) need to be clarified.
    - There seem to be several gaps that are not even discussed between the initial setting and the algorithm so it is not clear to what extent the convergence result of Theorem 3 really addresses the initial problem.

---

> ### Author Response · Authors · 2023-04-15
> **Response to Reviewer BwjS: Part 1**
>
> $\textbf{Q1: The connection between our setting and the framework of MFC with common noise.}$
>
> $\textbf{Response:}$ We would like to clarify that although our framework is similar to that analyzed in [1], [2], there are some subtle differences.
>
> Firstly, the setting of open-loop control as stated in [1], [2] can be described as a simplified version of our framework. This is because an open-loop policy is defined as a sequence of actions rather than a sequence of state-to-action maps. Our paper is more aligned with closed-loop policies. Secondly, the state-transition dynamics in [2] is written as:
> 	\begin{align*}
> 	X_{t+1}^i = F\left(X_t^i, U_t^i, \dfrac{1}{N}\sum_{j=1}^N\delta(X_t^j, U_t^j), \epsilon_{t+1}^i\right)
> 	\end{align*}
> 	where $\\{X_t^i, U_t^i\\}$ denote the state and action of the $i$th agent at time $t$, and $\\{\epsilon_{t+1}^i\\}_{i=1}^N$ are independent and identically distributed (iid) noise.
>
> Note that the assumption of iid noise ensures that the state-transition probabilities of each agent are conditionally independent (conditioned on current states, and actions) and identical. This assumption allows us to use the law of large numbers and prove  MARL-MFC error guarantees. However, if we define the state of the $i$th agent at time $t$ as $(X_t^i, g_t)$ (as the reviewer suggested), then the evolution of $\\{(X_t^i, g_t)\\}_{i=1}^N$ will no longer be conditionally independent (the common state, $g_t$ will introduce correlation) and the proof techniques that are commonly applied to prove guarantees can no longer be used. The novelty of our paper is in inventing new techniques to establish MARL-MFC error guarantees even when the common state $g_t$ disrupts the conditional independence structure.
>
> [1] Motte, M. and Pham, H., 2022. Mean-field Markov decision processes with common noise and open-loop controls. The Annals of Applied Probability, 32(2), pp.1421-1458.
>
> [2] Carmona, R., Laurière, M. and Tan, Z., 2019. Model-free mean-field reinforcement learning: mean-field MDP and mean-field Q-learning. arXiv preprint arXiv:1910.12802.
>
> $\textbf{Q2: Sufficiency of stationary policy.}$
>
> $\textbf{Response:}$ We refer to Theorem 6.2.12(c) of Puterman's book which dictates that if the state space is Polish and the action space is finite, then there exists an optimal stationary policy. In our case, the state space is $\mathcal{X}\times \Delta(\mathcal{X})\times \mathcal{G}$ where $\mathcal{X}$, $\mathcal{G}$ are finite and $\Delta(\mathcal{X})$ denotes the probability simplex over $\mathcal{X}$. As $\Delta(\mathcal{X})$ is a closed subset of $\mathbb{R}^{|\mathcal{X}|}$, our state space is Polish. Moreover, the action space, $\mathcal{U}$ is also assumed to be finite. Therefore, the existence of a stationary policy is guaranteed. We have added the theorem number along with the citation of the book in the revised manuscript.
>
> $\textbf{Q3: The gap between parameterized policies and all policies.}$
>
> $\textbf{Response:}$ We agree with the reviewer that not all policies can be parameterized by a finite-dimensional parameter. The fact that the parameterized policy set may not contain all policies is manifested in the parameter $\epsilon_{\mathrm{bias}}$ which is used in Lemma 1 and Theorem 3. The following excerpt from the paper explains the importance of the said term (page 12).
>
> > The term $\epsilon_{\mathrm{bias}}$ introduced in Lemma 1 is termed as the expressivity error of the policy class $\Pi$ parameterized by the $\mathrm{d}-$dimensional parameter. For dense neural network-based policies, $\epsilon_{\mathrm{bias}}$ appears to be small [1].
>
>  [1] Yanli Liu, Kaiqing Zhang, Tamer Basar, and Wotao Yin. An improved analysis of (variance-reduced) policy gradient and natural policy gradient methods. Advances in Neural Information Processing Systems, 33:7624–7636, 2020.
>
> $\textbf{Q4: In which sense are the errors in Lemma 1 and Theorem 3 defined?}$.
>
> $\textbf{Response:}$ We thank the reviewer for pointing out the incompleteness of the definition of the error. In the revised manuscript, we have defined the errors in Lemma 1 and Theorem 3 as the difference between the expected value obtained from the algorithm and the optimal value.
>
> $\textbf{Q5: Grammar correction: ``comprising of N number of agents".}$
>
> $\textbf{Response:}$ We thank the reviewer for pointing out the mistake. We have changed it to ''comprising $N$ number of agents''.
>
> $\textbf{Q6: Are the state space and the action space finite?}$
>
> $\textbf{Response:}$ Both the state space and the action space are presumed to be of finite size. We have explicitly mentioned this in the revised manuscript. The following sentence is added (section 1.1).
>
> > We consider a network comprising N number of agents, each associated with a local state space of size $|\mathcal{X}|$, a non-decomposable global state space of size $|\mathcal{G}|$ and an action space of size $|\mathcal{U}|$ (all are assumed to be	of finite size).

---

> > ### Comment · Reviewer_BwjS · 2023-05-03
> > **About Q1 and Q3**
> >
> > Thank you for your answers. I would like to mention the following points:
> >
> > **Related to the previous Q1:** In these two references [1] and [2], the evolution includes a common noise. The dynamics is not as you wrote in your answer because it includes an extra $\epsilon^0$, which is precisely here to take into account correlations between the agents' states. Please check for instance the first equation on page 5 of [2], namely:
> >
> > \[
> > X^i_{n+1} = F(X^i_n, \alpha^i_n, \frac{1}{N}\sum_{j=1}^N\delta_{X^j_n,\alpha^j_n}, \epsilon^i_{n+1},\epsilon^0_{n+1}).
> > \]
> >
> > [1] Motte, M. and Pham, H., 2022. Mean-field Markov decision processes with common noise and open-loop controls. The Annals of Applied Probability, 32(2), pp.1421-1458.
> >
> > [2] Carmona, R., Laurière, M. and Tan, Z., 2019. Model-free mean-field reinforcement learning: mean-field MDP and mean-field Q-learning. arXiv preprint arXiv:1910.12802.
> >
> > **Related to the previous Q3:** I appreciate the explanations, but I did see where this point is clearly explained in the revision.

---

> > > ### Author Response · Authors · 2023-05-03
> > > **Response about Q1 and Q3**
> > >
> > > $\textbf{Q1:}$ Thank you for pointing out the correlation term, $\epsilon^0_{n+1}$. We agree that with the introduction of $\epsilon^0_{n+1}$, our state transition model becomes similar to that reported in [1], [2]. However, [1] considers only open-loop policies which are different from the policies considered in our paper. On the other hand, although [2] considers closed-loop policies, it does not bound the approximation error between an $N$-agent and an infinite-agent system as a function of $N$, and the sizes of state and action spaces. We hope it clarifies the novelty of our paper.
> > >
> > > [1] Motte, M. and Pham, H., 2022. Mean-field Markov decision processes with common noise and open-loop controls. The Annals of Applied Probability, 32(2), pp.1421-1458.
> > >
> > > [2] Carmona, R., Laurière, M. and Tan, Z., 2019. Model-free mean-field reinforcement learning: mean-field MDP and mean-field Q-learning. arXiv preprint arXiv:1910.12802.
> > >
> > > $\textbf{Q3:}$ In the revised manuscript, we have now highlighted the relevant paragraph in blue (page 12).

---

> ### Author Response · Authors · 2023-04-15
> **Response to Reviewer BwjS: Part 2**
>
> $\textbf{Q7: Do all the agents use the same policy?}$
>
> $\textbf{Response:}$ It is assumed that all agents execute the same policy. This is a consequence of the presumption that all agents are identical (i.e., they possess the same reward and transition function) which is a common assumption in the mean-field literature [1] (section 6), [2]. We have explicitly added the following sentence in the revised manuscript to emphasize this point.
>
> > Here we implicitly assume that all agents execute the same policy. This is primarily because the agents are presumed to have the same reward function and state-transition function [1], [2].
>
> [1] Gu, H., Guo, X., Wei, X., \& Xu, R. (2021). Mean-field controls with Q-learning for cooperative MARL: convergence and complexity analysis. SIAM Journal on Mathematics of Data Science, 3(4), 1168-1196.
>
> [2] Pasztor, Barna, Ilija Bogunovic, and Andreas Krause. ``Efficient model-based multi-agent mean-field reinforcement learning." arXiv preprint arXiv:2107.04050 (2021).
>
> $\textbf{Q8: How is the set of allowed policies defined?}$
>
> $\textbf{Response:}$ Performing theoretical analysis/maximization on the set of all policies may not be feasible for many reinforcement learning problems. This is especially true if the state space is either large or infinite. In such a scenario, it is a common practice to narrow down the focus to a subset of policies, often referred to as the set of admissible policies, $\Pi$. On one hand, the choice of $\Pi$ should be narrow enough to yield theoretical guarantees. On the other hand, it should be wide enough to make the results useful for a large number of application scenarios. In our paper, we show that the error approximation guarantees (Theorem 1, 2) hold if $\Pi$ satisfies Assumption 2. However, for Theorem 3 to be true, it is additionally required that $\Pi$ be parameterized by a $\mathrm{d}-$dimensional parameter, and satisfy assumptions $4-7$. The fact that $\Pi$ may not contain all policies is manifested in the parameter $\epsilon_{\mathrm{bias}}$ (used in Lemma 1 and Theorem 3) which, in general, is bounded away from zero.
>
> $\textbf{Q9: Minor correction: ``their current".}$
>
> $\textbf{Response:}$ Thank you for pointing this out. We have corrected it in the revised manuscript.
>
> $\textbf{Q10: Supporting citations for Assumption 1.}$
>
> $\textbf{Response:}$ We agree with the reviewer that [1] may not be appropriate for corroborating  Assumption 1 since [1] primarily deals with continuous spaces while our paper focuses on discrete spaces. In the revised manuscript, we have replaced [1] with a more appropriate citation [2].
>
> [1] René Carmona, François Delarue, et al. Probabilistic Theory of Mean Field Games with Applications I-II. Springer, 2018.
>
> [2] Pasztor, Barna, Ilija Bogunovic, and Andreas Krause. ``Efficient model-based multi-agent mean-field reinforcement learning." arXiv preprint arXiv:2107.04050 (2021).
>
> $\textbf{Q11: Appropriateness of $\max$ operation.}$
>
> $\textbf{Response:}$ Our analysis does not require the existence of a maximum. In the revised manuscript, we have therefore changed the $\max$ operation to the $\sup$ operation.
>
> $\textbf{Q12: Grammar correction: ``the functions r and P"}$.
>
> $\textbf{Response:}$ We have corrected the text accordingly.
>
> $\textbf{Q13: It is not clear that (15) is more compact than (14).}$
>
> $\textbf{Response:}$ In the revised manuscript, we have changed the text to ``this can be alternatively written as follows."
>
> $\textbf{Q14: Clarification on $(15)$.}$
>
> $\textbf{Response:}$ The equation $(15)$ computes the joint probability of the sequence $g_{1:t}\triangleq \\{g_1, \cdots, g_t\\}$ conditioned on initial states $\boldsymbol{\mu}\_0, g\_0$, and the past sequence of policies $\pi\_{0:t-1}\triangleq \\{\pi_0, \cdots, \pi_{t-1}\\}$. Instead of writing the joint probability as a function of the whole sequence $g_{1:t}$, we define a function that depends on $g_{1:t-1}$ and takes $g_t$ as an input. Although these two approaches are mathematically equivalent, the second one is notationally more convenient and therefore has been used in the paper.
>
> $\textbf{Q15: Clarification related to the $\triangleq$ symbol.}$
>
> $\textbf{Response:}$ We thank the reviewer for pointing this out. In the revised manuscript, we have consistently placed the symbol to be defined on the left-hand side of the $\triangleq$ symbol.
>
> $\textbf{Q16: Clarification regarding summation over $g_{1:t}$.}$
>
> $\textbf{Response:}$ Summation over $g_{1:t}$ means summation over all elements of the set $\mathcal{G}^t$ which is $t$-times Cartesian product of $\mathcal{G}$. This has been clarified in the text. We quote the following excerpt for reference.
>
> > where $\sum_{1:t}$ is the summation operation over $g_{1:t}\in\mathcal{G}^t$, $t\geq 1$.

---

> ### Author Response · Authors · 2023-04-15
> **Response to Reviewer BwjS: Part 3**
>
> $\textbf{Q17: Clarification regarding the first equality of Step 2 in Proof Outline (Section 6.2).}$
>
> $\textbf{Response:}$ The first equality is a straightforward application of the definition of the function $\tilde{r}^{\mathrm{MF}}$ provided in section 6.1. We agree that the output of the function, $\tilde{r}^{\mathrm{MF}}$ is deterministic for a given input. Therefore, the expectation around $\tilde{r}^{\mathrm{MF}}$ accounts for the stochasticity of its input variables.
>
> $\textbf{Q18: Regarding the derivation of $(19)$.}$
>
> $\textbf{Response:}$ The derivation of $(19)$ uses the idea of telescoping series. This has been explained in the revised manuscript. We have added an intermediate step in the derivation of $(19)$ and provided the following explanation. (Sec 6.2).
>
> > Equality (a) is essentially a telescoping series.
>
> $\textbf{Q19: Clarification regarding $(k,t)$-dependent Lipschitz parameter.}$
>
> $\textbf{Response:}$ The reviewer is right to point out that a sum over terms with leading parameter $S_P^{t-k}$ may diverge. Fortunately, we need to compute the sum of $\gamma$-discounted terms with leading parameters $\gamma^t S_P^{t-k}$. For sufficiently small $\gamma$, this sum converges. The restriction on $\gamma$, as pointed out by the reviewer, therefore, plays a vital role in establishing the error guarantees. We have added the following explanation in the revised manuscript (section 6.2, step 6).
>
> > We would like to point out here that the leading coefficient of $\Delta R_t$ turns out to be an exponential function of $t$. In order for the sum to converge, one must have $\gamma$ to be sufficiently small.
>
> $\textbf{Q20: Typo: ``reward of representative".}$
>
> $\textbf{Response:}$ We have corrected it to ``reward of representative agent".
>
> $\textbf{Q21: Clarification on the presumption of stationary policy.}$
>
> $\textbf{Response:}$ Please see our response to Q2.
>
> $\textbf{Q22: Clarification on parameterized policies.}$
>
> $\textbf{Response:}$ The set of admissible policies, $\Pi$ is an arbitrary collection of policy functions that we want to restrict our attention to. Such a definition is necessary since performing maximization over the set of all policies may not be feasible. Theorem 1 and 2 require $\Pi$ to satisfy Assumption 2. Theorem 3 additionally requires $\Pi$ to be parameterized by a $\mathrm{d}$-dimensional parameter and satisfy assumptions $4-7$. The fact that $\Pi$ may not contain all policies is manifested in the parameter $\epsilon_{\mathrm{bias}}$ (used in Lemma 1 and Theorem 3) which, in general, is bounded away from zero.
>
> $\textbf{Q23: Correction regarding ``deterministic quantities".}$
>
> $\textbf{Response:}$ We have corrected the mistake in the revised manuscript.
>
> $\textbf{Q24: Correction regarding the subscript of $\Phi$.}$
>
> $\textbf{Response:}$ We have corrected the mistake in the revised manuscript.
>
> $\textbf{Q25: Clarification regarding ``over the set of admissible policies".}$
>
> $\textbf{Response:}$ We would like to clarify that the optimization is not performed over the set of all policies but rather over the set of admissible policies, $\Pi$. In section 7, $\Pi$ is assumed to be parameterized by a $\mathrm{d}$-dimensional parameter.
>
> $\textbf{Q26: Intuition regarding $(24)$ and $(25)$.}$
>
> $\textbf{Response:}$ The terms defined by $(24)$ and $(25)$ are commonly used in the design and analysis of policy gradient algorithms. Specifically, the term defined by $(25)$ is called the occupancy measure and intuitively it counts the total number of $\gamma$-discounted visits to a given state-action pair from an initial state-action pair. On the other hand, the term defined in $(24)$ is used to compute the natural gradient for updating the policy parameters. This can be thought of as the projection error of the gradient onto the advantage values. A detailed discussion of the intuition behind these terms is available in [1].
>
> [1] Alekh Agarwal, Sham M Kakade, Jason D Lee, and Gaurav Mahajan. On the theory of policy gradient methods: Optimality, approximation, and distribution shift. Journal of	Machine Learning Research, 22(98):1–76, 2021.
>
> $\textbf{Q27: Clarification on the sampling subroutine (Algorithm 2).}$
>
> $\textbf{Response:}$ Our Algorithm 1 is an adaptation of the algorithm described in [1] to the mean-field environment. Its sampling subroutine, however, is adopted from [2] (Algorithm 3). Hence, theoretical guarantees regarding the sampling procedure are available in [2].
>
> [1] Yanli Liu, Kaiqing Zhang, Tamer Basar, and Wotao Yin. An improved analysis of (variance-reduced) policy gradient and natural policy gradient methods. Advances in Neural Information Processing Systems, 33:7624–7636, 2020.
>
> [2] Alekh Agarwal, Sham M Kakade, Jason D Lee, and Gaurav Mahajan. On the theory of policy gradient methods: Optimality, approximation, and distribution shift. Journal of	Machine Learning Research, 22(98):1–76, 2021.

---

> ### Author Response · Authors · 2023-04-15
> **Response to Reviewer BwjS: Part 4**
>
> $\textbf{Q28: Existence of a policy class satisfying assumptions $4-7$.}$
>
> $\textbf{Response:}$ The class of Gaussian policies with linearly parameterized mean satisfies the said assumptions [1].
>
> [1] Yanli Liu, Kaiqing Zhang, Tamer Basar, and Wotao Yin. An improved analysis of (variance-reduced) policy gradient and natural policy gradient methods. Advances in Neural Information Processing Systems, 33:7624–7636, 2020.
>
> $\textbf{Q29: Clarifications regarding Lemma 1.}$
>
> $\textbf{Response:}$ Taking cues from Theorem 4.9 of [1], in the revised manuscript, we have described how the parameters related to the algorithm are determined by the parameters described in assumptions $4-7$ (see Lemma 1). It is clarified that the algorithmic parameters are pre-decided and not dependent on the initial condition. We also stated that the constant terms hiding in $\mathcal{O}(\cdot)$ are solely dependent on the parameters described in assumptions $4-7$. Finally, the error expression is corrected and is written as a difference between the expected value obtained from the algorithm and the optimal value.
>
> [1] Yanli Liu, Kaiqing Zhang, Tamer Basar, and Wotao Yin. An improved analysis of (variance-reduced) policy gradient and natural policy gradient methods. Advances in Neural Information Processing Systems, 33:7624–7636, 2020.
>
> $\textbf{Q30: On the gap between $V_N^*$ and $\max_{\boldsymbol{\pi}}V_{\boldsymbol{\pi}}$}$.
>
> $\textbf{Response:}$ Ideally, we would like to perform the maximization $\max_{\boldsymbol{\pi}}V_{\boldsymbol{\pi}}$ over the set of all policies (the $\textit{original}$ problem). However, as discussed before, such a task turns out to be formidable when the state space is either large or infinite. Therefore, our best hope is to solve the maximization over a subset of admissible policies, denoted as $\Pi$. Our approximation results (Theorem $1,2$) hold if $\Pi$ satisfies Assumption 2 whereas Theorem 3 additionally requires $\Pi$ to be parameterized by a $d$-dimensional parameter and obey Assumptions $4-7$. Thus, our algorithm approximately solves the $N$-agent problem if we allow the policies to be chosen only from the restricted set, $\Pi$. Outside of this policy set, we cannot comment on the optimality gap.
>
> $\textbf{Q31: On the missing quantifiers.}$
>
> $\textbf{Response:}$ We have removed the quantifiers to emphasize that the inequalities depicted in Lemma $2-5$ are true $\forall\pi\in\Pi$, and $\forall \boldsymbol{\mu}, \bar{\boldsymbol{\mu}}\in\Delta(\mathcal{X})$.
>
> $\textbf{Q32: Clarifying the summation operation.}$
>
> $\textbf{Response:}$ The summation is defined over the elements of the set, $\mathcal{G}^r$. Please see our response to Q16 for more details.

---

> > ### Comment · Reviewer_BwjS · 2023-05-03
> > **About Q28**
> >
> > Thank you for your answers. I would like to mention the following points:
> >
> > **Related to the previous Q28:** I am sorry but (1) I still do not understand how this provides and example and (2) I did not see where this is mentioned in the revision. Aren't such Gaussian policies for continuous action spaces while you say you consider finite action spaces? The paper [1] of Liu et al. you mention actually refers for instance to the paper of Papini et al. [2], which considers continuous action spaces (see e.g. their appendix C). So I do not see how this is related to the problem you study in the paper under review. A detailed example to justify the applicability of your result would be required.
> >
> > [1] Yanli Liu, Kaiqing Zhang, Tamer Basar, and Wotao Yin. An improved analysis of (variance-reduced) policy gradient and natural policy gradient methods. Advances in Neural Information Processing Systems, 33:7624–7636, 2020.
> >
> > [2] Matteo Papini, Damiano Binaghi, Giuseppe Canonaco, Matteo Pirotta, and Marcello Restelli. Stochastic variance-reduced policy gradient. In Proceedings of the 35th International Confer- ence on Machine Learning, pages 4026–4035, 2018.

---

> > > ### Author Response · Authors · 2023-05-03
> > > **Response about Q28**
> > >
> > > In the revised manuscript, we have now added Remark 1 (page 11) which shows an example of a class of policies that satisfies the said assumptions. Please see the following excerpt.
> > >
> > > > $\textbf{Remark 1.}$ Note that Assumption 7 is trivially satisfied with $\epsilon\_{\mathrm{bias}}=2M\_R/(1-\gamma)$. Assume that, $\mathcal{U}=\\{1, 2\\}$, and a restricted class of parameterized policies is defined as follows.
> > > $$\pi\_{\Phi}(x, \boldsymbol{\mu}, g) = \left[\dfrac{\exp(\Phi)}{1+\exp(\Phi)}~\dfrac{1}{1+\exp(\Phi)}\right]^T$$
> > >
> > > > where $(x, \boldsymbol{\mu}, g)\in \mathcal{X}\times \Delta(\mathcal{X})\times \mathcal{G}$, and $\Phi\in [-\xi, \xi]$ for some constant, $\xi>0$. One can check that, for the class of policies stated above, assumptions $4-6$ are satisfied with $\chi=\exp(2\xi)/(1+\exp(\xi))^4$, $G=1$, and $M=1/4$.

---

### Author Response · Authors · 2023-04-15
**Thanks to the Reviewers**

We thank the reviewers for their constructive and timely reviews. In the revised manuscript, the changes made in response to the reviews have been highlighted in blue.

---

### Decision · Action_Editors · 2023-05-23

**Recommendation:** Accept with minor revision

**Comment:**

This paper extends the scope of mean-field control (MFC) to a natural setting of multi-agent RL where each agent has a non-decomposable global state in addition to the local states. As is standard in MFC, local states are assumed to be conditionally independent of each other, but in this case the global state is observed by all the agents, which can condition their policy. The main contribution is then that MFC can still be used as a good approximation framework accompanied by a natural policy gradient algorithm with bounds on the approximation error (wrt. the optimal policy).

The reviewers had some mixed thoughts about this paper, and the overall points are summarized:

1. There are some limitations that are inherited by MFC required by the proof (e.g. Lipshitz continuity), so the proof has similar restrictions in applicability to the MFC domain. (Both UELm and dGHi)

2. Clarifications remain (both UELm and BwjS). In particular, some connections to previous work (identified by BwjS).

3. Remaining concerns can be addresses as future work (UELm).

4. Concerns about novelty (all reviewers) and scope (BwjS).


Despite outstanding concerns:

- I believe the limitations are a minor problem, since they are inherited by MFC, and the approximation from MFC is what allows the scalability in the first place.
- Novelty (and in particular, technical novelty mentioned by dGHi) is not a requirement for acceptance to TMLR
- Two of three reviewers lean toward accept, one sentiment was particularly well-expressed: "it does touch a useful topic with practical relevance" (dGHi). This point, in addition to the topic being one addressing scaling of MARL, means it will be of interest to at least a few individuals in TMLR's audience.

The authors' verbose replies were appreciated by the reviewers. Still, there are clarifications that remain. In particular, in the latest communication with BjwS, they admit that the similarities (and differences) to past work [1], [2] in Questions 1 & 3, but I do not see any text on this added to the paper. The authors should add (perhaps in an appendix) a discussion of how their work relates to [1], [2] and cite these relevant previous works. (The authors state that they address this on Page 12, but I still don't see any references nor discussion to past work nor how their framework is related to [1], [2])

[1] Motte, M. and Pham, H., 2022. Mean-field Markov decision processes with common noise and open-loop controls. The Annals of Applied Probability, 32(2), pp.1421-1458.

[2] Carmona, R., Laurière, M. and Tan, Z., 2019. Model-free mean-field reinforcement learning: mean-field MDP and mean-field Q-learning. arXiv preprint arXiv:1910.12802.


**Audience:**

This is a result concerning multiagent reinforcement learning (MARL), which will interest at least a few individuals in the audience of TMLR.

That said, the audience is rather small being a small subset (mean-field MARL) of an already small subset (MARL) of RL.

**Claims And Evidence:**

The claims are supported by accurate and clear evidence (see detailed reply in the comment section below)

---

> ### Author Response · Authors · 2023-05-25
> **Response**
>
> Thank you for accepting the paper. We have submitted the deanonymized version of the paper and added the following paragraph that clarifies the connection with previous works on MFC with common noise (section 1.2).
>
> > We would like to point out that our framework is closely aligned with the framework of MFC with common noise (Motte and Pham, 2022; Carmona et al., 2019). However, (Motte and Pham, 2022) only considers open-loop policies which are essentially sequences of actions, rather than state-to-action maps (also known as closed-loop policies). On the other hand, although (Carmona et al., 2019) do consider closed-loop policies, they do not show the convergence between MARL and MFC as a function of the number of agents.